_vms@cref

# A SIMPLE MEAN FIELD MODEL OF FEATURE LEARNING

## ABSTRACT

Feature learning (FL), where neural networks adapt their internal representations during training, remains poorly understood. Using methods from statistical physics, we derive a tractable, self-consistent mean-field (MF) theory for the Bayesian posterior of two-layer non-linear networks trained with stochastic gradient Langevin dynamics (SGLD). At infinite width, this theory reduces to kernel ridge regression, but at finite width it predicts a symmetry breaking phase transition where networks abruptly align with target functions. While the basic MF theory provides theoretical insight into the emergence of FL in the finite-width regime, semi-quantitatively predicting the onset of FL with noise or sample size, it substantially underestimates the improvements in generalisation after the transition. We trace this discrepancy to a key mechanism absent from the plain MF description: *self-reinforcing input feature selection*. Incorporating this mechanism into the MF theory allows us to quantitatively match the learning curves of SGLD-trained networks and provides mechanistic insight into FL.

## 1 INTRODUCTION

The ability of deep neural networks to automatically learn useful features during training is widely regarded as a central factor in their success (Bengio et al., 2014; LeCun et al., 2015). The best understood theories of generalisation work in the infinite-width limit (Jacot et al., 2018; Lee et al., 2018). These yield valuable insights but cannot account for key finite-width phenomena. In particular, finite networks exhibit feature learning (FL) effects such as improved sample complexity (e.g., sparse parity (Damian et al., 2022; Daniely & Malach, 2020) and (multi-)index functions (Bietti et al., 2022)) and task-aligned changes in hidden representations (Papyan et al., 2020; Chizat et al., 2019; Corti et al., 2025; Nam et al., 2024). This gap highlights the need for theoretical frameworks that capture the mechanisms underlying FL.

### 1.1 RELATED WORK

Attempts to bridge this theoretical gap fall into two main categories. *(1) Dynamical theories* describe the evolution of network properties during training via integro-differential equations (Bordelon & Pehlevan, 2022; 2023; Mei et al., 2018; Montanari & Urbani, 2025; Celentano et al., 2025; Lauditi et al., 2025; Shi et al., 2022). These approaches, often centered on the transition from lazy to rich regimes, can yield accurate empirical predictions but are mathematically complex. *(2) Static theories* analyze ensembles of neural networks trained with stochastic gradient Langevin dynamics (SGLD), a process equivalent to sampling from the Bayesian posterior that simplifies the analysis of the final state after training (Welling & Teh, 2011; Teh et al., 2015). By focusing on the stationary distribution of the posterior, these theories capture feature learning (FL) through data-dependent kernel adaptations, often in asymptotic limits of data and width (Pacelli et al., 2023; Baglioni et al., 2024; Fischer et al., 2024; Rubin et al., 2025; Ringel et al., 2025) or via kernel renormalization (Howard et al., 2025; Aiudi et al., 2025). Removed sentence.

Here, we offer a complementary perspective, prioritizing simplicity and mechanistic interpretability. We begin by employing standard methods from statistical physics to derive a self-consistent mean-field (MF) theory for the posterior of a two-layer non-linear network trained with SGLD (building on Rubin et al. (2024)). This MF model can be viewed as a minimal extension beyond the fixed-kernel NNGP limit (see Figure 1 **b**)). It predicts the onset of FL with increasing dataset size $P$

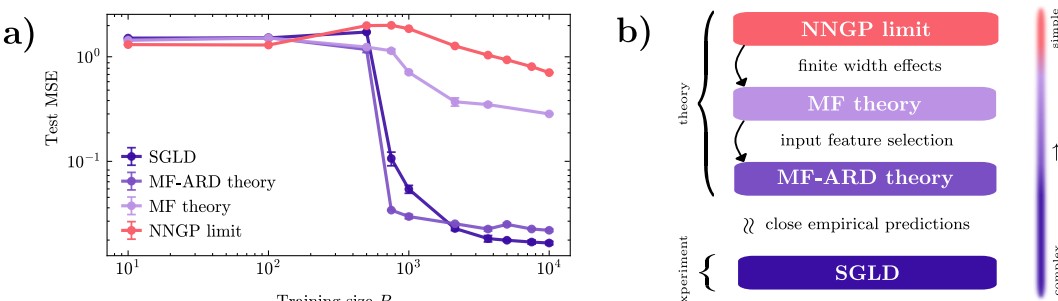

Figure 1: **a)** Generalisation error vs. training set size $P$ for 2-layer $N = 512$ width RELU networks trained with SGLD on $k$-sparse parity compared to the predicted error of the 3 theories presented in the paper (training details in Section F). **b)** Hierarchy of theories presented in this paper.

or decreasing noise strength $\kappa$ as a *symmetry breaking phase transition* where the initial isotropy of the weights is broken towards task-relevant directions given by the data. The symmetry is broken in the input coordinates, unlike symmetry breaking between neurons observed in van Meegen & Sompolinsky. While this MF theory predicts a mechanism for the onset of FL, it substantially underestimates the generalization gains of SGLD-trained networks after FL kicks in (see Figure 1 **a**). We trace this discrepancy to a missing mechanism eliminated by our simple MF approximation: *input feature selection* (IFS), where networks dynamically amplify weights for relevant input dimensions, specializing subsets of neurons while leaving others inactive (see Figure 3 **b**)). The homogeneity assumption in the basic MF model cannot capture these effects.

Our key contribution is to show that incorporating this IFS mechanism requires only a minimal, principled modification to the MF theory: endowing the weight prior with a learnable, coordinate-wise variance. The resulting model, which we call MF-ARD (Automatic Relevance Determination), preserves the simplicity and tractability of the MF framework while capturing the essence of FL.

Our **contributions** are:

- **Interpretable MF theory:** We derive a simple, self-consistent MF theory for the posterior of a non-linear two-layer NN. We show that this model captures the onset of FL as a phase transition but fails to predict the full learning curve of SGLD-trained networks.

- **Identification of a core FL mechanism:** We identify IFS as the key missing mechanism in standard MF theory and introduce MF-ARD, which captures this mechanism while preserving the simplicity of the MF form. We prove (Theorem 4.1) that this ARD extension eliminates the $\mathcal{O}(d)$ penalty in input dimension $d$ inherent in standard MF theory, providing a mechanistic understanding of how FL can overcome the curse of dimensionality.

- **Quantitative prediction of generalisation error:** We demonstrate that MF-ARD quantitatively predicts the generalisation error of SGLD-trained networks across varying dataset sizes and noise levels (see Figure 1 **a)**, Figure 5).

## 2 THEORY: A HIERARCHY OF MODELS (SGLD → MF → NNGP)

SGLD is the limit of full-batch GD with weight decay and injected Gaussian noise. It links optimisation and Bayesian inference by viewing parameter trajectories as samples from a posterior distribution (Welling & Teh, 2011; Teh et al., 2015). Given a neural network described by $f_{\boldsymbol{\theta}}(\boldsymbol{x}_\mu)$, a dataset $\mathcal{D} = \{(\boldsymbol{x}_\mu, \boldsymbol{y}_\mu)\}_{\mu=1}^P$, drawn from an input distribution $q(\boldsymbol{x}, \boldsymbol{y})$, the SGLD update equation is:

$$\Delta\theta_{i,t} := -\eta \left[ \gamma_i\, \theta_{i,t} + \nabla_{\theta_i} \left( \frac{1}{P} \sum_{\mu=1}^P \left( f_{\boldsymbol{\theta}}(\mathbf{x}_\mu) - \boldsymbol{y}_\mu \right)^2 \right) \right] + \sqrt{2T\eta}\, \xi_{i,t}, \quad \xi_{i,t} \sim \mathcal{N}(0,1), \quad (1)$$

with full-batch gradients on parameters $\theta_i$, batch noise replaced by Gaussian noise $\xi$, learning rate $\eta$, weight decay $\gamma_i$, and noise strength set by $T$. The stationary posterior $p_{\text{GD}}$ of Equation (1) as

$\eta \to 0$ for an $L$-layer network (widths $N_l$) is

$$-\ln p_{\text{GD}}(\boldsymbol{\theta}|\mathcal{D}) \propto \underbrace{\sum_{l=1}^{L} \frac{1}{2\sigma_i^2} \sum_{i=1}^{N_l} \|\boldsymbol{\theta}_i\|^2}_{-\ln p_{\text{prior}}} + \underbrace{\frac{1}{2\kappa^2 P} \sum_{\mu=1}^{P} (f_{\boldsymbol{\theta}}(\boldsymbol{x}_\mu) - \boldsymbol{y}_\mu)^2}_{-\ln p_{\text{L}}}, \tag{2}$$

where $\boldsymbol{\theta}_i$ are weight matrices, and $\sigma_i^2 = T/\gamma_i$ is related to noise $\kappa^2 = T/2$. Training with SGLD can be viewed as sampling from the posterior in Equation (2) (see Section A.1 for more details).

## 2.1 SGLD posterior: Fully interacting theory

In this paper, we focus on two-layer fully-connected networks with input dimension $d$, hidden width $N$, input weights $\boldsymbol{w}_i \in \mathbb{R}^d$, output weights $a_i$, output dimension 1, and nonlinearity $\phi$:

$$f_{\boldsymbol{\theta}}(\boldsymbol{x}) = \frac{1}{N^\gamma} \sum_{i=1}^{N} a_i \phi(\boldsymbol{w}_i^\top \boldsymbol{x}), \qquad w_{ij} \sim \mathcal{N}\left(0, \frac{\sigma_w^2}{d}\right), \quad a_i \sim \mathcal{N}(0, \sigma_a^2). \tag{3}$$

The initial parameters are set with Gaussians. The scale factor $N^{-\gamma}$ on the output layer enables interpolation between $\gamma = 1/2$ (NTK scaling) and $\gamma = 1$ (mean field scaling, not to be confused with MF theory, see e.g. Mei et al. (2019) for details). Using the explicit form of $f_{\boldsymbol{\theta}}$ in Equation (3) to rewrite Equation (2) by expanding the square inside $-\ln p_{\text{L}}$ yields (see Section A.2 for the algebra):

---

**SGLD-posterior**

$$-\ln p_{\text{GD}} = \frac{1}{2\sigma_a^2} \sum_{i=1}^{N} a_i^2 + \frac{d}{2\sigma_w^2} \sum_{i=1}^{N} \|\boldsymbol{w}_i\|^2 + \frac{1}{2\kappa^2 N^{2\gamma}} \sum_{i=1}^{N} a_i^2 \Sigma(\mathbf{w}_i)$$

$$+ \frac{1}{2\kappa^2 N^{2\gamma}} \sum_{i \neq i'} a_i a_{i'} G(\boldsymbol{w}_i, \boldsymbol{w}_{i'}) - \frac{1}{\kappa^2 N^\gamma} \sum_{i=1}^{N} a_i J_{\mathcal{Y}}(\mathbf{w}_i) + \text{const.}, \tag{4}$$

where $p_{\text{GD}} \to p_{\text{GD}}(\{\boldsymbol{w}_i\}_{i=1}^{N}, \boldsymbol{a}|\mathcal{D})$ with $\boldsymbol{w}_i \in \mathbb{R}^d$, and output weights $\boldsymbol{a} \in \mathbb{R}^N$.

- - - - - - - - - - - - - - - - - - - - - - - - - - - - - - - - - - - - - - - - - - - - - - - - - - - -

**Interpretation**
> $\Sigma(\boldsymbol{w}) = \frac{1}{P} \sum_\mu \phi(\boldsymbol{w}^\top \boldsymbol{x}_\mu)^2$ : Self-energy preventing infinite activations, penalizing large weights.
> $J_{\mathcal{Y}}(\boldsymbol{w}) = \frac{1}{P} \sum_\mu [\phi(\boldsymbol{w}^\top \boldsymbol{x}_\mu) y(\boldsymbol{x}_\mu)]$ : Neuron–data alignment drives learning (breaks symmetry).
> $G(\boldsymbol{w}_i, \boldsymbol{w}_{i'}) := \frac{1}{P} \sum_{\mu=1}^{P} \phi(\boldsymbol{w}_i^\top \boldsymbol{x}_\mu) \phi(\boldsymbol{w}_{i'} \boldsymbol{x}_\mu)^\top$ : Neuron-neuron interaction.

---

**FL mechanism as complex neuron-neuron interaction** The following competing forces shape the posterior: The self-energy $\Sigma$ acts as a regularizer, keeping neurons from becoming too large. The data coupling term $J_{\mathcal{Y}}$ is the learning signal: it rewards neurons when they align with the target function. Finally, the interaction kernel $G$ captures how neurons influence each other. As $P$ increases (or $\kappa$ decreases), the likelihood tilts the posterior away from the isotropic Gaussian prior, creating anisotropy along task-relevant directions and cooperative alignment across neurons via $G$. This can yield a transition to a non-Gaussian posterior concentrated on low-rank, task-aligned structures.

This posterior is not analytically tractable due to the neuron-neuron coupling via $G$: every neuron's optimal weight is a function of every other neuron (see Figure 2 for an illustration). From a Bayesian perspective, this means we cannot write the posterior distribution as a simple product of independent terms; it is instead a complex, high-dimensional distribution.

## 2.2 MF Theory: Removing off-diagonal couplings

One of the simplest ways to make the posterior in Equation (4) tractable is inspired by statistical physics and called self-consistent mean-field (MF) theory. The main idea is to replace the highly-correlated posterior over the entire weight matrix, $p(\boldsymbol{W}|\mathcal{D})$, with a fully factorized approximation where each of the $N$ neurons is independent: $p(\boldsymbol{W}|\mathcal{D}) \approx \prod_{i=1}^{N} p_{\text{MF}}(\boldsymbol{w}_i)$ (see Figure 2 for an illustration). Instead of interacting with all other neurons, each interacts with the average behavior

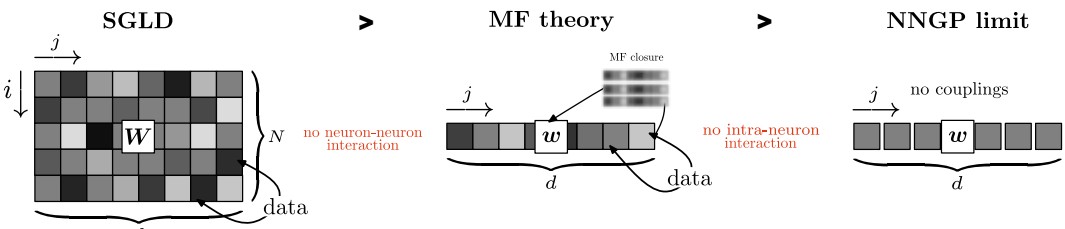

Figure 2: **SGLD Posterior:** The model is a fully interacting system where every neuron (row $\boldsymbol{w}_i$) is coupled to every other neuron ($\boldsymbol{w}_{i'}$) through the interaction kernel $G$. This results in a complex, high-dimensional posterior that is computationally intractable. **MF theory:** The pairwise interactions are replaced by an effective field that represents the average influence of all other neurons. Each neuron is now treated as an independent sample from a single, shared, data-dependent distribution, $p_{\mathrm{MF}}(\boldsymbol{w})$. **NNGP Limit:** All interactions are removed. Each neuron is an independent and identically distributed sample drawn from the fixed prior distribution.

of all other neurons. Take any neuron $\boldsymbol{w}_i$, replace the interaction term $\sum_{i' \neq i} a_j G(\boldsymbol{w}_i, \boldsymbol{w}_i')$, which couples neuron $\boldsymbol{w}_i$ to other neurons $\boldsymbol{w}_{i'}$, with their collective average effect, the 'mean field' $\langle f \rangle$:

$$\sum_{i' \neq i} a_{i'} \, G(\boldsymbol{w}_i, \boldsymbol{w}_{i'}) = \frac{1}{P} \sum_{\mu} \phi(\boldsymbol{w}_i^\top \boldsymbol{x}_\mu) \overbrace{\sum_{i' \neq i} a_{i'} \, \phi(\boldsymbol{w}_{i'}^\top \boldsymbol{x}_\mu)}^{\text{MF closure}} \xrightarrow{\text{MF}} N^\gamma \langle f(\boldsymbol{x}) \rangle. \tag{5}$$

For the approximation to be valid, it must be self-consistent: the average behavior of a neuron drawn from our approximate posterior $p_{\mathrm{MF}}(\boldsymbol{w})$ must exactly reproduce the mean field that we assumed in the first place, giving rise to the following self-consistency equation: $\langle f(\boldsymbol{x}) \rangle = N^{1-\gamma} \cdot \mathbb{E}_{(\boldsymbol{w},a) \sim p(\boldsymbol{w},a)} \big[ a \, \phi(\boldsymbol{w}^\top \boldsymbol{x}) \big]$.

The posterior is best understood in terms of an orthonormal basis $\{\chi_A\}$ w.r.t. the input distribution $q(\boldsymbol{x}, \boldsymbol{y})$, where $A$ indexes the basis. Expanding the MF $\langle f(\mathbf{x}) \rangle = \sum_A m_A \chi_A(\mathbf{x})$ in said basis, where the *feature coefficients* are $m_A = \mathbb{E}_{\boldsymbol{x}} \big[ \langle f(\boldsymbol{x}) \rangle \, \chi_A(\boldsymbol{x}) \big]$, gives the following theory:

**MF theory (fixed point equations)**

$$-\ln p_{\mathrm{MF}} = \frac{a^2}{2\sigma_a^2} + \frac{d}{2\sigma_w^2} \sum_{j=1}^d w_j^2 + \frac{a^2}{2\kappa^2 N^{2\gamma}} \Sigma(\boldsymbol{w}) - \frac{a}{\kappa^2 N^\gamma} \left( J_{\mathcal{Y}}(\boldsymbol{w}) - \sum_A m_A J_A(\boldsymbol{w}) \right) \tag{6}$$

$$m_A = N^{1-\gamma} \langle a J_A(\boldsymbol{w}) \rangle_{p_{\mathrm{MF}}} \quad \forall \, A \tag{7}$$

where $p_{\mathrm{MF}} \to p(\boldsymbol{w}, a | \{m_A\}, \mathcal{D})$, $\boldsymbol{w} \in \mathbb{R}^d$, $a \in \mathbb{R}$ and $J_A(\boldsymbol{w}) = \mathbb{E}_{\boldsymbol{x}}[\phi(\boldsymbol{w}^\top \boldsymbol{x}) \chi_A(\boldsymbol{x})]$.

**Interpretation**
> $m_A$: Feature coefficients measure how strongly the average neuron aligns with a certain basis function.
> $G \to \sum_A m_A J_A$: Coupling of the single neuron to the effective field from all the other neurons.

The core simplification of the MF theory is that *each neuron is treated as an independent sample drawn from the same distribution, which is a function of the prior and the data*. This strongly simplifies the problem. We replace the intractable problem over the entire interacting $N \times d$ weight matrix with a much simpler, self-consistent problem for a single $d$-dimensional weight vector.

These fixed-point equations can easily be solved by iterating to self-consistency, enabling the computation of all statistics $\Sigma(\boldsymbol{w}), J_A(\boldsymbol{w}), J_{\mathcal{Y}}(\boldsymbol{w})$ directly from the *finite training dataset* of size $P$. In this way, we naturally capture the effects of training with limited samples (see Section E for details).

## 2.3 NNGP limit: No interactions

The MF model can be simplified even further in the infinite-width limit with $\gamma = 1/2$ (NTK-scaling). For this scaling the limit is well-behaved, the self-energy term in Equation (6) vanishes as well as any $m_A$-dependent tilt, resulting in $p(\boldsymbol{w})$ collapsing to its prior (see Section D.2 for a *proof*).

**NNGP limit (fixed point equations)**

$$-\ln p_\infty = \frac{d}{2\sigma_w^2}\|\boldsymbol{w}\|^2 \tag{8}$$

$$m_A^\infty = \frac{\sigma_a^2}{\kappa^2}\left(\langle J_\mathcal{Y}(\boldsymbol{w})J_A(\boldsymbol{w})\rangle_{p_\infty} - \sum_B m_B^\infty \langle J_B(\boldsymbol{w})J_A(\boldsymbol{w})\rangle_{p_\infty}\right) \tag{9}$$

where $p_\infty \to p_\infty(\boldsymbol{w})$, $\boldsymbol{w} \in \mathbb{R}^d$.

- - - - - - - - - - - - - - - - - - - - - - - - - - - - - - - - - - - - - - - - - - - -

**Kernel picture** We can define the kernel: $K_{AB} := \mathbb{E}_{\boldsymbol{w}\sim p_\infty}\left[J_A(\boldsymbol{w})\,J_B(\boldsymbol{w})\right]$ reducing the solution above to kernel ridge regression:

$$\boldsymbol{m}^\infty = K\left(K + \tau \mathbb{1}\right)^{-1}\boldsymbol{y}, \qquad \tau = \kappa^2/\sigma_a^2, \quad y_A = \mathbb{E}_{\boldsymbol{x}}[y(\boldsymbol{x})\chi_A(\boldsymbol{x})]. \tag{10}$$

This NNGP limit represents the most restrictive limit in our approximation hierarchy. While MF theory eliminates inter-neuron interactions ($w_{ij} \leftrightarrow w_{i'j}$), the NNGP limit additionally removes intra-neuron coordinate coupling ($w_{ij} \leftrightarrow w_{ij'}$), see Figure 2 for an illustration. Every weight is sampled from the same distribution (the prior).

**No FL mechanism** In the infinite-width limit, there is *no* FL in the sense above. With no data-dependent term ($J_\mathcal{Y}$) to break symmetry, all neurons remain frozen at their prior. The feature coefficients $m_A^\infty$ are now determined purely by the fixed kernel $K_{AB} = \mathbb{E}_{\boldsymbol{w}\sim p_\infty}[J_A(\boldsymbol{w})J_B(\boldsymbol{w})]$ (see Section A.5 how this connects to the usual kernel formulation of the NNGP limit).

## 3 PROBLEM: SIMPLE MF DOES NOT CAPTURE A CENTRAL FL MECHANISM

In the following section, we will argue that FL in 2-layer finite-width networks is a two-stage process:

(i) **Onset (phase transition):** The signal from the data is strong enough to overcome the isotropic prior/self-energy, so the feature coefficients $m_A = \mathbb{E}_{\boldsymbol{x}}[\langle f(\boldsymbol{x})\rangle_p \chi_A(\boldsymbol{x})]$ turn nonzero.

(ii) **Specialization through self-reinforcing input feature selection (IFS):** After the symmetry breaking, the neurons aligning with the target receive disproportionately larger updates, producing heavy-tailed weight marginals on task-relevant coordinates and neuron-wise sparsification.

Plain MF captures *(i)* but largely misses *(ii)*, which is why for a target $y(\boldsymbol{x}) = \chi_S(\boldsymbol{x})$, it underestimates the growth of $m_S$ and the post-transition generalisation (e.g., Figure 1 **a)**, Figure 3 **a)**).

### 3.1 STAGE 1: THE ONSET OF FEATURE LEARNING AS A PHASE TRANSITION

The simplicity of the MF theory allows us to interpret FL as a *self-consistent symmetry breaking* in the posterior $p_{\mathrm{MF}}$, where learning emerges from a competition between two opposing forces. Below a critical ($P_c, \kappa_c$) the only stable fixed point solution of Equations (6) and (7) is $m_A = 0\ \forall A$ and $p_{\mathrm{MF}}$ corresponds to the Gaussian prior. The Gaussian weight prior and the self-energy term $\Sigma(\boldsymbol{w})$ act as regularizing forces, favouring an isotropic, high-entropy state that penalizes large weights and encourages neurons to remain small, unaligned and near their initialization. Upon crossing the critical values ($P_c, \kappa_c$) when the signal from the data (controlled by dataset size $P$ and noise $\kappa$) becomes strong enough, the $m_A = 0$ fixed point becomes unstable (see Theorem D.8 for a *proof* of the phase transition in $\kappa$ and an explicit formula for $\kappa_c$). $p_{\mathrm{MF}}$ becomes non-Gaussian, as the neuron-data coupling term ($J_\mathcal{Y}$), which acts like an external field, rewards neurons whose activations $\phi(\boldsymbol{w}^\top \boldsymbol{x})$ correlate with the target function $y(\boldsymbol{x})$, breaking the prior's symmetry and pulling the weights toward a task-aligned configuration. For the case where the target function is a single basis function $y(\boldsymbol{x}) = \chi_S(\boldsymbol{x})$, the relevant order parameter is the feature coefficient $m_S$ of the target function. In this case, equation 7 becomes (see Section D.6 for the derivation):

$$m_S = \frac{N^{1-2\gamma}}{\kappa^2}(1-m_S)\underbrace{\left\langle \frac{J_S(\boldsymbol{w})^2}{\sigma_a^{-2} + \dfrac{\Sigma(\boldsymbol{w})}{\kappa^2 N^{2\gamma}}} \right\rangle_{\boldsymbol{w}\sim p(\boldsymbol{w}|m_S)}}_{\omega_0}. \tag{11}$$

The phase transition occurs when $\omega_0 > 1$, as shown in Figure 3**a**): for the SGLD-trained network and MF theory, the order parameter $m_S = 0$ until the critical dataset size $P_c$, when it increases abruptly. The NNGP model, lacking this non-linear feedback, shows only a smooth, continuous rise. In Section A.4 we link this FL phase transition to the emergence of an outlier eigenvalue in the Gram kernel $G(\boldsymbol{w}_i, \boldsymbol{w}_{i'})$.

However, Figure 1 **a**) reveals the limitation: after the transition, the SGLD-trained network's generalisation error decreases rapidly, while the MF model's error decreases much more slowly. This discrepancy reveals a second powerful mechanism at play that MF theory misses.

### 3.2 STAGE 2: SPECIALIZATION VIA IFS (WHAT SIMPLE MF MISSES)

What enables the SGLD-trained network to improve so rapidly post-transition? The answer lies in a self-reinforcing dynamic that is lost in the MF approximation.

---

**Hypothesis: *Input feature selection* as central FL mechanism**
Consider the weight gradient of the SLGD posterior:

$$\nabla_{\boldsymbol{w}_i}(-\ln p_{\text{GD}}) = \frac{d}{\sigma_w^2}\,\boldsymbol{w}_i + \frac{a_i}{\kappa^2 P\, N^\gamma}\sum_{\mu=1}^{P} r_\mu\,\phi'(\mathbf{w}_i^\top \boldsymbol{x}_\mu)\,\boldsymbol{x}_\mu, \qquad r_\mu := f_{\boldsymbol{\theta}}(\boldsymbol{x}_\mu) - y_\mu. \quad (12)$$

All neurons $\boldsymbol{w}_i$ observe the same residual $r_\mu$, but differ in their alignment $\phi'(\mathbf{w}_i^\top \boldsymbol{x}_\mu)\,\mathbf{x}_\mu$. Some neurons $\boldsymbol{w}_{i'}$ achieve better alignment at initialization, thus receiving proportionally larger gradient updates, strengthening this alignment further. Simultaneously, the improved prediction from $\boldsymbol{w}_{i'}$ reduces the residual $r_\mu$ for all neurons, weakening the gradient signal for less-aligned neurons whose dynamics become dominated by weight decay. This creates the self-reinforcing *input feature selection*: weak initial signals amplify into strong coordinate-wise symmetry breaking. While this is a dynamical effect, it leaves its traces in the static posterior, in particular in the heavy-tailed structure of $p(\boldsymbol{w}_i)$ (see Figure 4 **a**) and in Figure 3 **b**)) where only a few neurons in the SGLD-trained NN pick up a strong target correlation.

---

A consequence of IFS is *sparsification*: only a small number of neurons strongly align with the target while others remain near initialization. We quantify this through neuron specialization $\tilde{m}_S = \langle a J_S(\boldsymbol{w})\rangle_{p_{\text{GD}}}$, measuring how strongly individual neurons couple to the target function (see Figure 4 **b**)). Before the phase transition $P = 100 < P_c$, $p(\tilde{m})$ is sharply peaked at zero. After the transition $P = 1000 > P_c$, $p(\tilde{m})$ develops pronounced tails, most neurons remain unspecialized while a few become strongly task-aligned (Also, see the plot of $\boldsymbol{W}\boldsymbol{W}^\top$ in Figure 6 where task-relevant coordinates ($j = 0, 1, 2, 3$) have much higher norm.).

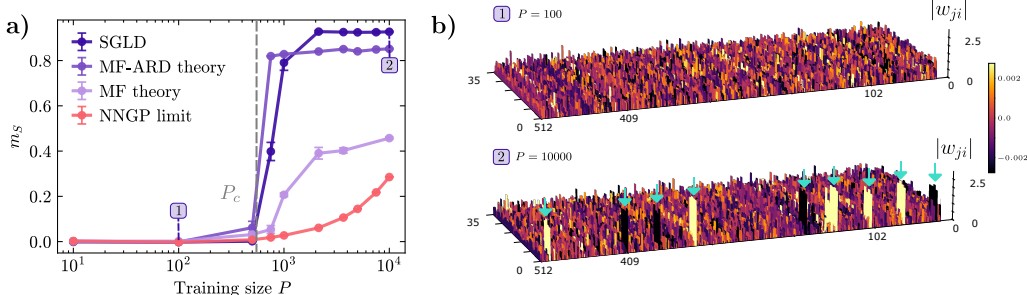

Figure 3: $k$-sparse parity target $y(\boldsymbol{x}) = \chi_S(\boldsymbol{x}) = \Pi_{j\in S} x_j$ with index $S = \{0, 1, 2, 3\}$ in $d = 35$ with a ReLU network ($N = 512$, see Section A.3 and Section F for more details.). **a**) $m_S$ **vs.** $P$: SGLD, MF and MF-ARD exhibit a phase transition at $P_c$, NNGP grows smoothly (no FL). **b**) **Weight matrix plotted for the SGLD-trained NN:** After the phase transition, only a very small set of neurons (9 out of 512) pick up a high correlation with the target ($m_S$ is color coded) and only for these neurons there is a strong symmetry breaking where the norm of the first $j = 0, 1, 2, 3$ coordinates is much larger than for the rest $d - k$ coordinates (coded as the height in the plot).

### 3.3 WHY PLAIN MF MISSES SPECIALIZATION

MF replaces the neuron–neuron interaction by a single effective field. As a result, *all* neurons are sampled from the *same* single-neuron distribution $p(\boldsymbol{w}, a)$ and are pushed to align in the *same average* way. This homogeneity suppresses the greedy dynamics that drive IFS and sparsification: The marginals $p(\boldsymbol{w}_j)$ for MF theory in Figure 4 **b)** show very weak tails compared to SGLD (Figure 4 **a)**) $\Rightarrow$ weak IFS. Similarly, the lack of heavy-tailed specialization $\Rightarrow$ weak sparsification. Hence, the simplification of removing neuron-neuron interaction weakens IFS. We hypothesize that this explains the persistent generalization error gap we observed relative to SGLD in many settings.

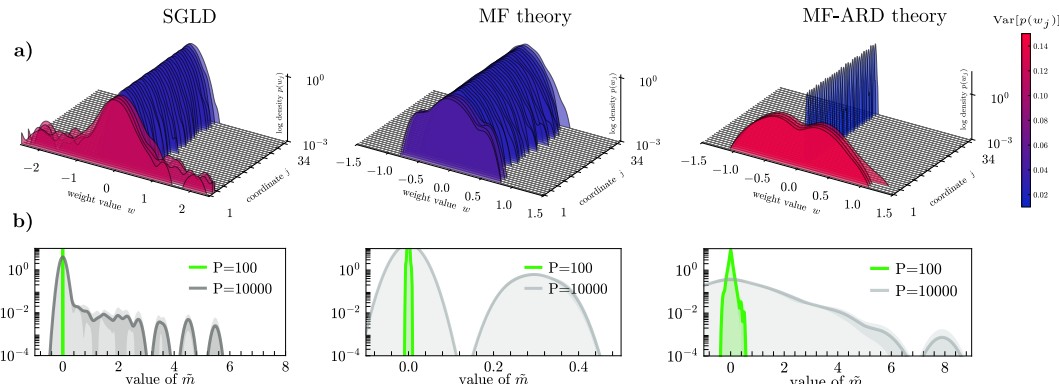

Figure 4: Setup as in Figure 3. **a) Coordinate-wise weight marginals** $p(w_j)$**:** For SGLD, $p(w_j)$ has high variance and strong non-Gaussianity on input relevant coordinates $j = 0, 1, 2, 3$ (symmetry breaking with IFS). Plain MF also shows symmetry breaking for coordinates $j = 0, 1, 2, 3$, but much weaker, while MF-ARD restores the strong anisotropy. **b) Distribution of specialization** $\tilde{m} = a J_S(\boldsymbol{w})$ **before vs. after phase transition:** SGLD develops heavy tails after the transition (sparsification), MF remains narrow, MF-ARD produces the heavy-tailed specialization. Together, MF-ARD qualitatively recovers the two mechanisms, IFS and sparsification, that MF misses.

## 4 A MINIMAL MF MODEL OF FEATURE LEARNING

This is new: How can we capture IFS within the tractable MF framework given that it emerges from neuron-neuron interactions that MF explicitly removes? In the full SGLD dynamics, the gradient $\nabla_w(-\ln p_{\mathrm{GD}})$ creates anisotropy: the likelihood term couples coordinates, driving weights along relevant dimensions $j$ further than irrelevant ones. Standard MF theory misses this because it assumes an isotropic prior, resulting in an identical restoring force for every coordinate $j = 1, \ldots, d$. To reintroduce this crucial coordinate dependence, we apply a linear response approximation to the stationary state. We posit that the complex interaction gradients can be modeled as an effective quadratic potential with a coordinate-specific stiffness $\tilde{\rho}_j$:

$$\mathbb{E}\left[w_j \nabla_{w_j}(-\ln p_{\mathrm{L}})\right] \approx \tilde{\rho}_j \mathbb{E}[w_j^2]. \tag{13}$$

Since the effective stiffness $\tilde{\rho}_j$ varies across input dimensions in real SGLD (high for noise, low for signal), we promote it to a learnable order parameter within the MF theory.

### 4.1 MODEL - ARD AS THE SMALLEST CHANGE THAT ENABLES IFS

In Theorem D.1, we rigorously derive this update by enforcing a physical consistency condition: in the limit of data-free dynamics (where the likelihood gradient vanishes and the data-induced stiffness $\tilde{\rho}_j \to 0$), the system must revert to the 'free' SGLD dynamics governed solely by the prior. Under this constraint, we prove that the total effective stiffness $\rho_j$ must satisfy the following fixed-point conditions.

end of the new text

**MF-ARD theory (fixed point equations)**

$$-\ln p_{\text{ARD}} = \frac{a^2}{2\sigma_a^2} + \frac{1}{2}\sum_{j=1}^{d}\rho_j w_j^2 + \frac{a^2}{2\kappa^2 N^{2\gamma}}\Sigma(\boldsymbol{w}) - \frac{a}{\kappa^2 N^\gamma}\left(J_{\mathcal{Y}}(\boldsymbol{w}) - \sum_A m_A J_A(\boldsymbol{w})\right) \quad (14)$$

$$m_A = N^{1-\gamma}\langle a J_A(\boldsymbol{w})\rangle_{p_{\text{ARD}}} \ \forall A \quad \text{and} \quad \rho_j = \frac{\alpha_0 + \frac{N}{2}}{\frac{\alpha_0}{d} + \frac{N}{2}\langle w_j^2\rangle_{p_{\text{ARD}}}} \quad (15)$$

where $p_{\text{ARD}} \to p_{\text{ARD}}(\boldsymbol{w}, a|\{m_A, \rho_i\}, \mathcal{D})$ and $\boldsymbol{w} \in \mathbb{R}^d$, $\boldsymbol{\rho} \in \mathbb{R}^d$, $a \in \mathbb{R}$.

The MF-ARD theory is still a MF theory, as neurons remain independent and interact only through the self-consistent averages $\{m_A\}$ and $\{\rho_i\}$. The crucial difference is the introduction of the new order parameters $\{\rho_j\}$, which allow the model to learn the relevance of each input feature.

**How ARD induces IFS**  We assume a target $y(\boldsymbol{x}) = \chi_S(\boldsymbol{x})$. The map $\langle w_j^2\rangle_{p_{\text{ARD}}} \mapsto \rho_j \mapsto p_{\text{ARD}}(\boldsymbol{w})$ realises IFS: if coordinate $j$ aligns with the target ($j \in S$), the term $J_{\mathcal{Y}}(\boldsymbol{w}) - \sum_A m_A J_A(\boldsymbol{w})$ increases, making $\langle w_{j\in S}^2\rangle_{p_{\text{ARD}}} \uparrow$ larger, which *decreases* $\rho_{j\in S} \downarrow$ via Equation (15), thereby *reducing* shrinkage on $w_{j\in S}$ and further increasing $\langle w_{j\in S}^2\rangle_{p_{\text{ARD}}} \uparrow$. Conversely, for non-aligned coordinates $j \notin S$, $\langle w_j\rangle^2$ remains small, which increases $\rho_j$ and strengthens shrinkage. This positive feedback realises IFS and leads to sparsification of the posterior (see Figure 4 **a)** and **b)**). Accordingly, in $\boldsymbol{w} = (w_1, ..., w_k, w_{k+1}, ..., w_d)$ the first $k$ weights are much larger than the remaining $d - k$. This increases $J_S(\boldsymbol{w})$ while decreasing $J_A(\boldsymbol{w}) \ \forall A \neq S$, driving $m_S$ up strongly after the phase transition while decreasing $m_A \ \forall A \neq S$. This explains the strong generalisation error drop of MF-ARD after the phase transition (unlike MF). Note that MF-ARD is still a *static* theory. The mechanism above reflects how the fixed point map contracts to its solution (via Equations (14) and (15)), not a model of the actual training dynamics. See Section A.7 for a discussion of the infinite width limit of the MF-ARD model.

## 4.2 How MF-ARD beats the curse of dimensionality

Let $y(\boldsymbol{x}) = \chi_S(\boldsymbol{x})$ with $|S| = k$ in $d$ dimensions and $\phi = \text{ReLU}$. Write $\kappa_c^2$ for the critical noise at onset of the phase transition. We assume the infinite data limit and focus on the phase transition in the critical noise as it is analytically more tractable than critical data size.

**Theorem 4.1.** *There exists an outer iteration $t_0 = \mathcal{O}(1)$ of the fixed point algorithm and a constant $\varepsilon_0 > 0$ (independent of $d$ such that we get $\varepsilon_0$-level symmetry breaking towards $S$: $\min_{j\in S}\langle w_j^2\rangle_{p_{\text{ARD}}}^{t_0} - \max_{j\notin S}\langle w_j^2\rangle_{p_{\text{ARD}}}^{t_0} \geq c \cdot \varepsilon_0$ (see Section D.7 for details). Then, the critical noise scales as:*

$$\kappa_c^2 \asymp \begin{cases} \Theta\big(\sqrt{1/(dk)}\big) & \text{(plain MF)}, \\ \Theta\big(\sqrt{1/k}\big) & \text{(MF–ARD)}. \end{cases} \quad (16)$$

*Proof.* See Section D.7 for a proof and exact constants. Here we summarize the scaling. □

$\varepsilon$-symmetry breaking assumes that there is already a very small difference in the variance for weights that are relevant vs. irrelevant for learning the target. Given this asymmetry, MF-ARD can amplify it far more effectively than plain MF. For plain MF, the noise threshold required to transition from high to low generalisation error (equivalently $m_S = 0 \to m_S \neq 0$) scales with the ambient dimension $d$, requiring noise levels $1/\sqrt{d}$ times smaller than MF-ARD. This dimensional dependence is eliminated in MF-ARD. Consequently, the phase boundary scales with the intrinsic problem dimension $k$ rather than the ambient dimension $d$.

## 4.3 Results- numerical prediction of learning curves

We evaluate on $k$-sparse parity, a setting where fixed kernels are known to require super-polynomial samples in $d, k$, while finite-width networks can have lower sample complexity due to FL (Daniely & Malach, 2020; Damian et al., 2022; Barak et al., 2022). Figure 5 shows test-MSE heatmaps over dataset size $P$ and noise $\kappa$ for (a) SGLD-trained networks, (b) plain MF, and (c) MF–ARD. All three display a transition from high to low error as $P$ increases or $\kappa$ decreases. Plain MF detects *when* learning starts but yields a diffuse, shifted boundary and overestimates post-transition error.

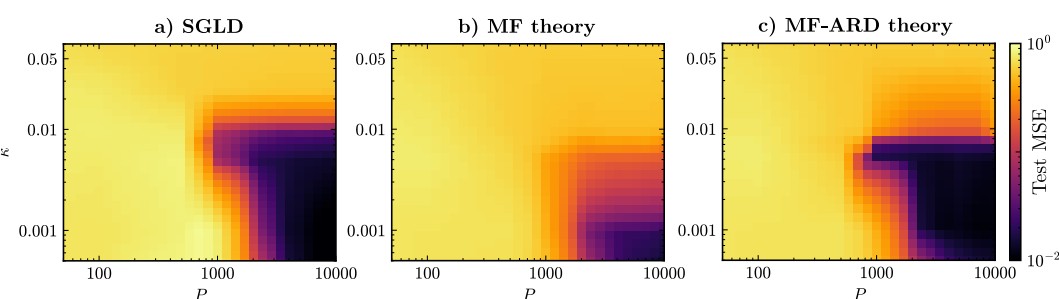

Figure 5: **Generalisation phase diagram (Test MSE) vs. dataset size $P$ and noise $\kappa$ for $k$-sparse parity target** $y(\boldsymbol{x}) = \chi_S(\boldsymbol{x})$ **with** $S = \{0, 1, 2, 3\}$ **in** $d = 35$. **a) SGLD, b) MF, c) MF-ARD:** All panels show a transition from high to low test error as $P$ increases or $\kappa$ decreases. However, the plain MF theory strongly underestimates the sharpness of the transition while the MF-ARD theory largely reproduces the shape/location of this boundary. Training details are in Section F.

MF–ARD, with a single additional set of order parameters $\{\rho_j\}$, closely tracks both the *location* and the *sharpness* of the SGLD phase boundary and reproduces the "helpful noise" regime in which moderate $\kappa$ lowers the critical sample size $P_c$ (the kink around $\kappa = 0.05$).

In Figure 7 in the appendix we present the same phase diagram analysis for a single index-model task (Bietti et al., 2022), where the target function takes the form $y(\boldsymbol{x}) = g(\mathbf{v}^T \boldsymbol{x})$. Here, $\boldsymbol{v}, \boldsymbol{x} \in \mathbb{R}^d$, the input $\boldsymbol{x}$ follows a standard multivariate normal distribution $\mathcal{N}(0, \mathbb{1})$, $v_{j \in S} = 1/\sqrt{k}$, $v_{j \notin S} = 0$ for an index set with $|S| = k$ and $g$ is a non-linear function (Hermitian polynomial in our case).

## 5 CONCLUSION

We present a MF theory for the posterior of SGLD-trained two-layer networks that is simple to interpret, with a clear infinite-width NNGP limit. Extending it via Automatic Relevance Determination (MF-ARD) to include coordinate-dependent weight variances preserves this simplicity while quantitatively matching network performance as a function of data set size and noise on tasks such as $k$-sparse parity and index models, where feature learning is essential.

Our framework characterises feature learning as a two-stage process: a data-driven *phase transition* mechanism that initiates feature learning, followed by a self-reinforcing *input feature selection* mechanism that leads to sparsification and drives improved generalisation. This explains why finite-width networks outperform kernel methods: they reshape their feature space in a self-reinforcing way that amplifies task-relevant signals and dramatically improves sample complexity. Standard MF theories capture only the transition, but MF-ARD captures both mechanisms.

Limitations of our approach include the focus on two-layer networks and tasks with sparse structures for which the ARD mechanism may be best suited. Extending to deeper architectures (e.g., convolutional or attention layers), and settings requiring distributed or smooth representations remains an open challenge (Petrini et al., 2022), as does connecting the static posterior view to training dynamics.

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

# A  ADDITIONAL BACKGROUND

## A.1  EQUIVALENCE OF LANGEVIN DYNAMICS AND BAYESIAN INFERENCE

In this section, we provide a detailed derivation for the equivalence between the stationary distribution of a network trained with Stochastic Gradient Langevin Dynamics (SGLD) and the posterior distribution of a corresponding Bayesian model.

**Network and Priors**  We analyze a two-layer neural network with the functional form:

$$f(\boldsymbol{x}) = \frac{1}{N^\gamma} \sum_{i=1}^{N} a_i \phi(\boldsymbol{w}_i^\top \boldsymbol{x}). \tag{17}$$

The parameters are drawn from independent Gaussian priors, which corresponds to choosing a specific prior in a Bayesian model. The initializations are given by:

- Weights: $w_{ij} \sim \mathcal{N}(0, g_w^2)$ with $g_w^2 = \frac{\sigma_w}{d}$. This implies a prior probability $p(\boldsymbol{w}_i) \propto \exp(-\frac{1}{2g_w^2}\|\boldsymbol{w}_i\|^2)$.

- Amplitudes: $a_i \sim \mathcal{N}(0, g_a^2)$ with $g_a^2 = \sigma_a^2$. This implies a prior probability $p(a_i) \propto \exp(-\frac{1}{2g_a^2}\|a_i\|^2)$.

Let $\boldsymbol{\theta}_i = (\boldsymbol{w}_i, a_i)$ denote the parameters for the $i$-th neuron.

**Stochastic Gradient Langevin Dynamics**  We consider a full-batch Gradient Descent (GD) update for a parameter set $\theta_i$, which includes a weight decay term with coefficient $\gamma$ and injected isotropic Gaussian noise $\xi_t \sim \mathcal{N}(0, I)$. This algorithm is known as Stochastic Gradient Langevin Dynamics (SGLD):

$$\Delta\theta_{i,t} := \theta_{i,t+1} - \theta_{i,t} \tag{18}$$

$$= -\eta\big(\gamma\,\theta_{i,t} + \nabla_{\theta_i}\ell(f_{\boldsymbol{\theta}})\big) + \sqrt{2T\eta}\,\xi_t\,. \tag{19}$$

Here, $\eta$ is the learning rate, $T$ is a scalar temperature that controls the noise magnitude, and $\ell(f_{\boldsymbol{\theta}}) = \frac{1}{P}\sum_{\mu=1}^{P}(y_\mu - f(\boldsymbol{x}_\mu))^2$ is the mean squared error loss over the dataset of size $P$.

In the continuous-time limit ($\eta \to 0$), this discrete update equation corresponds to a Langevin stochastic differential equation. The stationary distribution of this process, reached as $t \to \infty$, is given by the Gibbs-Boltzmann distribution:

$$p(\boldsymbol{\theta}_i) \propto \exp\!\Big(-\frac{1}{T}\Big(\frac{\gamma}{2}\|\theta_i\|^2 + \ell(f_{\boldsymbol{\theta}})\Big)\Big) \tag{20}$$

We can separate the weight decay terms $\gamma$ into two separate parameters for weights and amplitudes, $\gamma_w$ and $\gamma_a$, respectively. The stationary distribution for the parameters of a single neuron $(\boldsymbol{w}_i, a_i)$ is then:

$$p(\boldsymbol{w}_i, a_i) \propto \exp\left(-\frac{1}{T}\left(\frac{\gamma_w}{2}\|\boldsymbol{w}_i\|^2 + \frac{\gamma_a}{2}\|a_i\|^2 + \frac{1}{P}\sum_{\mu=1}^{P}(y_\mu - f(\boldsymbol{x}_\mu))^2\right)\right). \tag{21}$$

**Bayesian Posterior Distribution**  From a Bayesian perspective, we aim to find the posterior distribution of the parameters given the data $\mathcal{D} = \{(\boldsymbol{x}_\mu, y_\mu)\}_{\mu=1}^{P}$. The posterior is given by Bayes' theorem: $p(\boldsymbol{\theta}|\mathcal{D}) \propto p(\mathcal{D}|\boldsymbol{\theta})p(\boldsymbol{\theta})$.

- The prior $p(\boldsymbol{\theta})$ is defined by our choice of initialization: $p(\boldsymbol{\theta}) = \prod_i p(\boldsymbol{w}_i)p(a_i) \propto \exp\left(-\sum_i\left(\frac{1}{2g_w^2}\|\boldsymbol{w}_i\|^2 + \frac{1}{2g_a^2}\|a_i\|^2\right)\right)$.

- The likelihood $p(\mathcal{D}|\boldsymbol{\theta})$ is chosen to be a Gaussian distribution with variance $\kappa^2$, corresponding to the mean squared error loss: $p(\mathcal{D}|\boldsymbol{\theta}) \propto \exp\left(-\frac{1}{2\kappa^2 P}\sum_{\mu=1}^{P}(y_\mu - f(\boldsymbol{x}_\mu))^2\right)$.

Combining these, the log-posterior is proportional to the negative of an energy function $\mathcal{E}(\boldsymbol{\theta}, \mathcal{D})$. The posterior distribution for a single neuron's parameters is:

$$p(\boldsymbol{w}_i, a_i | \mathcal{D}) \propto \exp\left(-\left(\frac{1}{2g_w^2}\|\boldsymbol{w}_i\|^2 + \frac{1}{2g_a^2}\|a_i\|^2 + \frac{1}{2\kappa^2 P}\sum_{\mu=1}^{P}(y_\mu - f(\boldsymbol{x}_\mu))^2\right)\right). \qquad (22)$$

**Identifying the Distributions**   To establish the equivalence, we equate the functional forms of the SGLD stationary distribution and the Bayesian posterior. By comparing the exponents term-by-term, we find the following correspondences:

- **Weights:** $\frac{\gamma_w}{2T}\|\boldsymbol{w}_i\|^2 = \frac{1}{2g_w^2}\|\boldsymbol{w}_i\|^2 \implies \frac{\gamma_w}{T} = \frac{1}{g_w^2}$

- **Amplitudes:** $\frac{\gamma_a}{2T}\|a_i\|^2 = \frac{1}{2g_a^2}\|a_i\|^2 \implies \frac{\gamma_a}{T} = \frac{1}{g_a^2}$

- **Loss Term:** $\frac{1}{TP}\sum_\mu \ell_\mu = \frac{1}{2\kappa^2 P}\sum_\mu \ell_\mu \implies T = 2\kappa^2$

This implies that the SGLD algorithm with temperature $T$ effectively samples from a Bayesian posterior with data noise variance $\kappa^2 = T/2$, and with prior variances $g_w^2 = T/\gamma_w$ and $g_a^2 = T/\gamma_a$.

**Final Update Equations**   By substituting these relations back into the SGLD update rules, we obtain the dynamics for sampling from the desired posterior:

$$\Delta a_{t,i} = -\eta\left(\frac{T}{g_a^2}a_{t,i} + \nabla_{a_i}\left(\frac{1}{P}\sum_\mu \ell_\mu\right)\right) + \sqrt{2T\eta}\,\xi_{t,a} \qquad (23)$$

$$\Delta \boldsymbol{w}_{t,i} = -\eta\left(\frac{T}{g_w^2}\boldsymbol{w}_{t,i} + \nabla_{\boldsymbol{w}_i}\left(\frac{1}{P}\sum_\mu \ell_\mu\right)\right) + \sqrt{2T\eta}\,\xi_{t,w}. \qquad (24)$$

For training, this means we must set the temperature to $T = 2\kappa^2$.

A.2   DERIVATION OF EQUATION 4

We start from the negative log posterior in Eq. equation 2 and the two–layer model in Eq. equation 3:

$$-\ln p_{\mathrm{GD}}(\boldsymbol{\theta} \mid \mathcal{D}) = \underbrace{\sum_l \frac{1}{2\sigma_l^2}\sum_j \|\boldsymbol{\theta}_{l,j}\|^2}_{-\ln p_{\mathrm{prior}}} + \underbrace{\frac{1}{2\kappa^2 P}\sum_{\mu=1}^{P}\big(f_{\boldsymbol{\theta}}(\boldsymbol{x}_\mu) - y_\mu\big)^2}_{-\ln p_{\mathrm{L}}}.$$

For our two–layer network, $f_{\boldsymbol{\theta}}(\boldsymbol{x}) = \frac{1}{N^\gamma}\sum_{i=1}^{N} a_i \phi(\boldsymbol{w}_i^\top \boldsymbol{x})$. Define the shorthand $\phi_{i\mu} := \phi(\boldsymbol{w}_i^\top \boldsymbol{x}_\mu)$. Then the data term expands as

$$-\ln p_{\mathrm{L}} = \frac{1}{2\kappa^2 P}\sum_{\mu=1}^{P}\left(\frac{1}{N^\gamma}\sum_{i=1}^{N} a_i \phi_{i\mu} - y_\mu\right)^2$$

$$= \frac{1}{2\kappa^2 P}\sum_{\mu=1}^{P}\left[\frac{1}{N^{2\gamma}}\sum_{i=1}^{N}\sum_{j=1}^{N} a_i a_j\,\phi_{i\mu}\phi_{j\mu} - \frac{2}{N^\gamma}\sum_{i=1}^{N} a_i\,\phi_{i\mu}y_\mu + y_\mu^2\right]. \qquad (25)$$

Introduce the dataset–averaged quantities

$$\Sigma(\boldsymbol{w}) := \frac{1}{P}\sum_{\mu=1}^{P}\phi(\boldsymbol{w}^\top \boldsymbol{x}_\mu)^2, \qquad\qquad G(\boldsymbol{w}, \boldsymbol{w}') := \frac{1}{P}\sum_{\mu=1}^{P}\phi(\boldsymbol{w}^\top \boldsymbol{x}_\mu)\,\phi(\boldsymbol{w}'^\top \boldsymbol{x}_\mu),$$

$$J_{\mathcal{Y}}(\boldsymbol{w}) := \frac{1}{P}\sum_{\mu=1}^{P}\phi(\boldsymbol{w}^\top \boldsymbol{x}_\mu)\,y_\mu. \qquad (26)$$

Using $\sum_{i,j} = \sum_{i=j} + \sum_{i \neq j}$ in equation 25, we obtain

$$
-\ln p_{\text{L}} = \frac{1}{2\kappa^2 N^{2\gamma}} \left[ \sum_{i=1}^{N} a_i^2 \underbrace{\frac{1}{P} \sum_{\mu} \phi_{i\mu}^2}_{\Sigma(\boldsymbol{w}_i)} + \sum_{i \neq j} a_i a_j \underbrace{\frac{1}{P} \sum_{\mu} \phi_{i\mu} \phi_{j\mu}}_{G(\boldsymbol{w}_i, \boldsymbol{w}_j)} \right]
$$

$$
- \frac{1}{\kappa^2 N^{\gamma}} \sum_{i=1}^{N} a_i \underbrace{\frac{1}{P} \sum_{\mu} \phi_{i\mu} y_{\mu}}_{J_{\mathcal{Y}}(\boldsymbol{w}_i)} + \underbrace{\frac{1}{2\kappa^2 P} \sum_{\mu=1}^{P} y_{\mu}^2}_{\text{const.}}. \tag{27}
$$

The prior part for our parameterization $w_{ij} \sim \mathcal{N}(0, \sigma_w^2/d)$ and $a_i \sim \mathcal{N}(0, \sigma_a^2)$ is

$$
-\ln p_{\text{prior}} = \frac{1}{2\sigma_a^2} \sum_{i=1}^{N} a_i^2 + \frac{d}{2\sigma_w^2} \sum_{i=1}^{N} \|\boldsymbol{w}_i\|^2. \tag{28}
$$

Combining equation 27 and equation 28, and discarding the $\boldsymbol{\theta}$-independent constant, yields Eq. equation 4:

$$
-\ln p_{\text{GD}}(\boldsymbol{W}, \boldsymbol{a} \mid \mathcal{D}) = \frac{1}{2\sigma_a^2} \sum_{i=1}^{N} a_i^2 + \frac{d}{2\sigma_w^2} \sum_{i=1}^{N} \|\boldsymbol{w}_i\|^2 + \frac{1}{2\kappa^2 N^{2\gamma}} \sum_{i=1}^{N} a_i^2 \, \Sigma(\boldsymbol{w}_i)
$$

$$
+ \frac{1}{2\kappa^2 N^{2\gamma}} \sum_{i \neq j} a_i a_j \, G(\boldsymbol{w}_i, \boldsymbol{w}_j) - \frac{1}{\kappa^2 N^{\gamma}} \sum_{i=1}^{N} a_i \, J_{\mathcal{Y}}(\boldsymbol{w}_i) + \text{const.}
$$

This derivation holds for any nonlinearity $\phi$.

### A.3 $k$-SPARSE PARITY TARGET FUNCTION

A $k$-sparse parity target function teacher is a single Walsh basis function on the Boolean hypercube. Let $S \subseteq [d]$ with $|S| = k$. The Walsh function indexed by $S$ is

$$
\chi_S(\boldsymbol{x}) = \prod_{j \in S} x_j, \qquad \boldsymbol{x} \in \{\pm 1\}^d,
$$

(and, if $\boldsymbol{x} \in [-1, 1]^d$, one may use $\chi_S(\boldsymbol{x}) = \prod_{j \in S} \text{sign}(x_j)$). The family $\{\chi_S\}_{S \subseteq [d]}$ forms an orthonormal basis under the uniform product measure, i.e. $\mathbb{E}[\chi_S(\boldsymbol{x}) \chi_T(\boldsymbol{x})] = \mathbb{1}\{S = T\}$. In our experiments the teacher is $y(\boldsymbol{x}) = \chi_S(\boldsymbol{x})$, when $|S| = k$ this is exactly the $k$-parity problem.

### A.4 RELATION TO KERNEL EIGENVALUE OUTLIERS

The transition from $m_S = 0$ to $m_S > 0$ manifests as an outlier eigenvalue in the learned kernel. Recall that $m_S = N^{1-\gamma} \langle a J_S(\boldsymbol{w}) \rangle$ measures the mean alignment between neurons and the target mode, where $J_S(\boldsymbol{w}) = \mathbb{E}[\phi(\boldsymbol{w}^\top \boldsymbol{x}) \chi_S(\boldsymbol{x})]$ quantifies how well a single neuron with weights $\boldsymbol{w}$ correlates with $\chi_S$. When $m_S$ becomes non-zero, it indicates that neurons have collectively aligned their weights to capture the target structure. Their $J_S(\boldsymbol{w}_i)$ values have grown large. This alignment directly impacts the empirical kernel through

$$
\widehat{R}_S^{(a)} = \frac{\chi_S^\top K \chi_S}{\text{tr}(K)} = \frac{\sum_{i=1}^{N} a_i^2 \, J_S(\boldsymbol{w}_i)^2}{\sum_{i=1}^{N} a_i^2 \, \Sigma(\boldsymbol{w}_i)}, \tag{29}
$$

where $K_{\mu\nu} = \frac{1}{N} \sum_{i=1}^{N} \phi(\boldsymbol{w}_i^\top \boldsymbol{x}_\mu) \, \phi(\boldsymbol{w}_i^\top \boldsymbol{x}_\nu)$ is the learned kernel. The numerator $\sum_i a_i^2 J_S(\boldsymbol{w}_i)^2$ grows quadratically with the alignment strengths $J_S(\boldsymbol{w}_i)$, while the denominator (trace) remains roughly constant. In the kernel regime ($m_S = 0$), neurons remain randomly oriented so $J_S(\boldsymbol{w}_i) \sim O(N^{-1/2})$ and the ratio stays $O(1/P)$. However, when FL occurs ($m_S > 0$), the enhanced $J_S(\boldsymbol{w}_i)$ values cause $\chi_S^\top K \chi_S$ to grow substantially, making $\chi_S$ an outlying eigendirection of $K$. This anisotropic deformation, from the isotropic NNGP to a low-rank, plus isotropic structure, provides a direct spectral signature of the FL phase transition.

## A.5 CONNECTION TO THE STANDARD NNGP FORMULATION

The kernel representation in the function basis $\{\chi_A\}$ can be transformed to recover the standard NNGP formulation in input space. We start with the function-basis kernel

$$K_{AB} = \mathbb{E}_{\boldsymbol{w} \sim p_\infty}[J_A(\boldsymbol{w})J_B(\boldsymbol{w})], \tag{30}$$

where $J_A(\boldsymbol{w}) = \mathbb{E}_{\boldsymbol{x}}[\phi(\boldsymbol{w}^\top \boldsymbol{x})\chi_A(\boldsymbol{x})]$ projects the neuron's output onto basis function $\chi_A$. Expanding this definition:

$$K_{AB} = \mathbb{E}_{\boldsymbol{w}}\left[\mathbb{E}_{\boldsymbol{x}}[\phi(\boldsymbol{w}^\top \boldsymbol{x})\chi_A(\boldsymbol{x})] \cdot \mathbb{E}_{\boldsymbol{x}'}[\phi(\boldsymbol{w}^\top \boldsymbol{x}')\chi_B(\boldsymbol{x}')]\right] \tag{31}$$

$$= \mathbb{E}_{\boldsymbol{w}}\mathbb{E}_{\boldsymbol{x},\boldsymbol{x}'}\left[\phi(\boldsymbol{w}^\top \boldsymbol{x})\phi(\boldsymbol{w}^\top \boldsymbol{x}')\chi_A(\boldsymbol{x})\chi_B(\boldsymbol{x}')\right] \tag{32}$$

$$= \mathbb{E}_{\boldsymbol{x},\boldsymbol{x}'}\left[\chi_A(\boldsymbol{x})\chi_B(\boldsymbol{x}') \cdot \underbrace{\sigma_a^2\mathbb{E}_{\boldsymbol{w}}[\phi(\boldsymbol{w}^\top \boldsymbol{x})\phi(\boldsymbol{w}^\top \boldsymbol{x}')]}_{=:K(\boldsymbol{x},\boldsymbol{x}')}\right], \tag{33}$$

where $K(\boldsymbol{x}, \boldsymbol{x}')$ is the standard NNGP kernel in input space. The fixed-point equation $m_A = \frac{\sigma_a^2}{\kappa^2}(\Xi_A - \sum_B K_{AB}m_B)$ with $\Xi_A = \sum_S y_S K_{AS}$ becomes

$$m = K(K + \tau I)^{-1}y, \qquad \tau = \kappa^2/\sigma_a^2. \tag{34}$$

To see this explicitly, consider data points $\{\boldsymbol{x}_\mu\}_{\mu=1}^P$ with labels $y_\mu$. The predictor in the function basis is $f(\boldsymbol{x}) = \sum_A m_A\chi_A(\boldsymbol{x})$, while in the input basis it becomes $f(\boldsymbol{x}_\mu) = \sum_{\nu=1}^P \alpha_\nu K(\boldsymbol{x}_\mu, \boldsymbol{x}_\nu)$ where $\alpha = (K + \tau I)^{-1}y$. The equivalence follows from the change of basis: if $\chi_A(\boldsymbol{x}) = \delta_{\boldsymbol{x},\boldsymbol{x}_A}$ (point evaluation basis), then $K_{AB} = K(\boldsymbol{x}_A, \boldsymbol{x}_B)$ directly recovers the Gram matrix. For general orthogonal bases, the kernel ridge regression solution remains invariant under this transformation.

## A.6 DERIVATION OF THE ARD PRECISION FIXED POINT

Recall the ARD prior on per-coordinate precisions $\boldsymbol{\rho} = (\rho_1, \ldots, \rho_d)$ and weights:

$$p(\boldsymbol{w} \mid \boldsymbol{\rho}) = \prod_{j=1}^d \mathcal{N}(w_j \mid 0, \rho_j^{-1}), \qquad p(\boldsymbol{\rho}) = \prod_{j=1}^d \Gamma(\rho_j \mid \alpha_0, \beta_0), \tag{35}$$

and the single-neuron MF–ARD action (Equation (14) in the main text)

$$-\ln p_{\text{ARD}}(\boldsymbol{w}, a \mid \{m_A\}, \boldsymbol{\rho}, \mathcal{D}) = \frac{a^2}{2\sigma_a^2} + \frac{1}{2}\sum_{j=1}^d \rho_j\, w_j^2 + \frac{a^2}{2\kappa^2 N^{2\gamma}}\Sigma(\boldsymbol{w}) - \frac{a}{\kappa^2 N^\gamma}\left(J_{\mathcal{Y}}(\boldsymbol{w}) - \sum_A m_A J_A(\boldsymbol{w})\right). \tag{36}$$

For a width-$N$ two-layer network, the joint action is the sum over neurons $i = 1, \ldots, N$ of Equation (36), and the (negative) log-evidence (free energy) for fixed $\{m_A\}$ is

$$\mathcal{F}(\boldsymbol{\rho}; \{m_A\}) = -\ln Z(\boldsymbol{\rho}; \{m_A\}) - \ln p(\boldsymbol{\rho}) \quad \text{with} \quad Z(\boldsymbol{\rho}; \{m_A\}) = \int \prod_{i=1}^N d\boldsymbol{w}_i\, da_i\, e^{-\sum_{i=1}^N S_{\text{ARD}}(\boldsymbol{w}_i, a_i)}. \tag{37}$$

Stationarity of the negative log-evidence w.r.t. $\rho_j$ gives the ARD FP:

$$0 = \frac{\partial \mathcal{F}}{\partial \rho_j} = \underbrace{\left\langle \frac{\partial}{\partial \rho_j}\sum_{i=1}^N S_{\text{ARD}}(\boldsymbol{w}_i, a_i)\right\rangle_{p_{\text{ARD}}}}_{\text{energy term}} \underbrace{-\frac{N}{2}\frac{1}{\rho_j}}_{\text{Gaussian normalizer}} \underbrace{-\frac{\partial \ln p(\boldsymbol{\rho})}{\partial \rho_j}}_{\text{prior term}} = \frac{1}{2}\sum_{i=1}^N \langle w_{ij}^2\rangle_{p_{\text{ARD}}} - \frac{N}{2}\frac{1}{\rho_j} - \frac{\alpha_0 - 1}{\rho_j} + \beta_0, \tag{38}$$

where we used $\partial S_{\text{ARD}}/\partial \rho_j = \frac{1}{2}\sum_i w_{ij}^2$ and $\partial \ln \Gamma(\rho_j \mid \alpha_0, \beta_0)/\partial \rho_j = (\alpha_0 - 1)/\rho_j - \beta_0$. By MF symmetry, all neurons are i.i.d. under $p_{\text{ARD}}$, so $\sum_{i=1}^N \langle w_{ij}^2\rangle_{p_{\text{ARD}}} = N\langle w_j^2\rangle_{p_{\text{ARD}}}$, and Equation (38) yields the closed-form FP update

$$\rho_j^\star = \frac{\alpha_0 - 1 + \frac{N}{2}}{\beta_0 + \frac{N}{2}\langle w_j^2\rangle_{p_{\text{ARD}}}} \overset{\text{MAP corr.}}{\approx} \frac{\alpha_0 + \frac{N}{2}}{\beta_0 + \frac{N}{2}\langle w_j^2\rangle_{p_{\text{ARD}}}}, \tag{39}$$

where the MAP conjugacy correction (absorbing the $-1$ in $\alpha_0$) gives the form used in the main text (Equation (15)). With the scale-matching choice $\beta_0 = \alpha_0/d$, Equation (39) becomes

$$\rho_j^\star \;=\; \frac{\alpha_0 + \frac{N}{2}}{\frac{\alpha_0}{d} + \frac{N}{2}\,\langle w_j^2\rangle_{p_{\mathrm{ARD}}}}. \tag{40}$$

## A.7 NNGP LIMIT OF THE MF-ARD MODEL

**NNGP-limit of MF-ARD**

$$\mathcal{S}_\infty^{\mathrm{FL}}(\mathbf{w}|\rho) = \frac{1}{2}\sum_{j=1}^{d}\rho_j w_j^2, \qquad p_{\infty,\rho}(\mathbf{w}) = \frac{1}{\mathcal{Z}}\exp\big(-\mathcal{S}_\infty^{\mathrm{FL}}(\mathbf{w}|\rho)\big), \tag{41}$$

$$m_A^\infty(\rho) = \frac{\sigma_a^2}{\kappa^2}\Big\langle \big(J\mathcal{y}(\mathbf{w}) - \sum_B m_B^\infty(\rho) J_B(\mathbf{w})\big) J_A(\mathbf{w})\Big\rangle_{p_{\infty,\rho}}, \qquad \forall A. \tag{42}$$

$$0 = \frac{1}{2}\mathrm{tr}\big[(A - (Ay)(Ay)^\top)\partial_{\rho_j}K_\rho\big] + \beta_0 - \frac{\alpha_0-1}{\rho_j}, \qquad A = (K_\rho + \tau I)^{-1} \tag{43}$$

- - - - - - - - - - - - - - - - - - - - - - - - - - - - - - - - - - - - - - - - - - - - - - - - - - - - - -

**Kernel picture** We can define the kernel : $K_{\rho,AB} = \mathbb{E}_{\mathbf{w}\sim p_{\infty,\rho}}[J_A(\mathbf{w})J_B(\mathbf{w})]$ reducing the solution above to KRR

$$m^\infty(\rho) = K_\rho(K_\rho + \tau I)^{-1}y, \quad \tau = \kappa^2/\sigma_a^2 \tag{44}$$

**Usual (kernel) form** Let $C_\rho = \mathrm{diag}(\rho)^{-1}$ and $K_\rho(\mathbf{x},\mathbf{x}') = \sigma_a^2\,\mathbb{E}_{\mathbf{w}\sim\mathcal{N}(0,C_\rho)}[\phi(\mathbf{w}^\top\mathbf{x})\phi(\mathbf{w}^\top\mathbf{x}')]$. Then

$$m^\infty(\rho) = K_\rho(K_\rho + \tau I)^{-1}y, \quad K_\rho\chi_A = \lambda_A(\rho)\chi_A \;\Rightarrow\; m_A^\infty(\rho) = \frac{\lambda_A(\rho)}{\lambda_A(\rho) + \tau}\,y_A. \tag{45}$$

This shows that $\rho$ rescales coordinates before the nonlinearity, so $K_\rho = \mathbb{E}_{\mathbf{z}}[J_A(C_\rho^{1/2}\mathbf{z})\,J_B(C_\rho^{1/2}\mathbf{z})]$ is a nonlinear deformation of the isotropic kernel. Any nontrivial change in $\rho$ therefore produces an $O(1)$ change in the Gaussian measure over $\mathbf{w}$, hence an $O(1)$ change in $K$ (it does not vanish with width). ARD improves spectral alignment by increasing $\lambda_A(\rho)$ along task-aligned directions. When $\lambda_A(\rho)$ crosses the scale $\tau$, $m_A^\infty$ exhibits a jump, yielding the leading-order FL effect at infinite width.

## B ADDITIONAL FIGURES

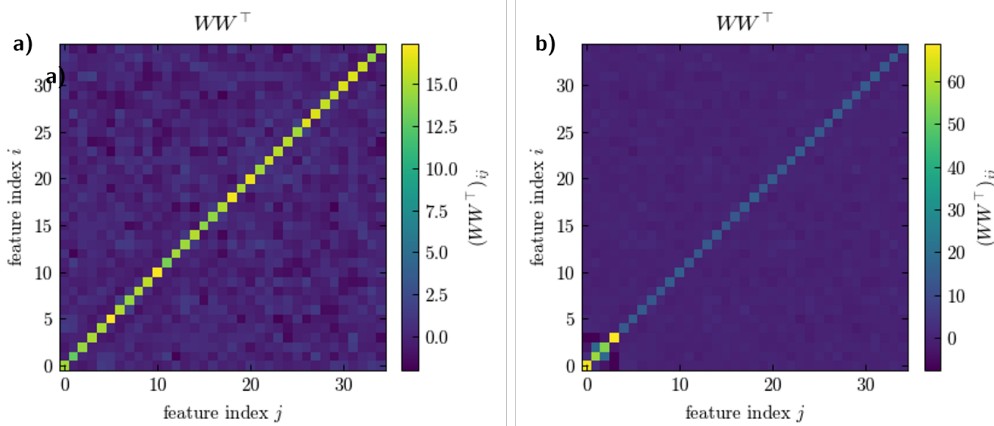

Figure 6: Plot of $\boldsymbol{W}\boldsymbol{W}^\top$ for a ReLU $N = 512$ network, trained with SGLD on the same setting as Figure 3.

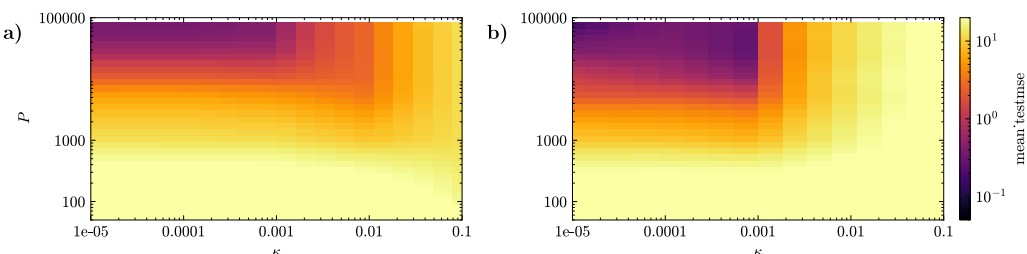

Figure 7: Same plot as Figure 5 **a)** SGLD, **b)** MF-ARD, but for a single-index model with Gaussian inputs $\boldsymbol{x}! \sim! \mathcal{N}(0, \mathbb{1})$ and teacher $y = \text{He}_p(\boldsymbol{w}^\top \boldsymbol{x})$, where $\boldsymbol{w}_j = 1/\sqrt{k}$ on a $k$-sized support and $0$ otherwise, here $d = 18$, $k = 2$, $p = 4$.

## C  RELATED WORK: FURTHER DETAILS

Motivated by the observation that existing theories of deep learning are often neither simple nor mutually consistent, recent work seeks a unified account that (i) retains analytical simplicity, (ii) captures the essential structure of learning curves , and (iii) explains when and how finite networks escape the curse of dimensionality. The objective across these efforts is to develop theories that consistently cover both "lazy" kernel regimes and genuine FL. A recurring result is that a small number of control parameters, most notably an *effective learning rate*, organize the transition between regimes while preserving MF tractability. The emerging conclusion is that appropriately reduced MF or Bayesian descriptions can remain simple yet accurately capture kernel evolution, alignment to target structure, and the finite-width mechanisms by which deep networks surpass their infinite-width kernel limits.

As mentioned in the main text, attempts to bridge the theoretical gap between infinite-width kernel limits and finite-width FL largely fall into two categories. *(1) Dynamical theories* derive integro–differential or MF equations that track how representations, kernels, and losses evolve *during training* (Bordelon & Pehlevan, 2022; 2023; Mei et al., 2018; Montanari & Urbani, 2025; Celentano et al., 2025; Lauditi et al., 2025). *(2) Static theories* analyze the *learned posterior* after training, connecting finite-width learning to data-dependent kernel adaptation (Pacelli et al., 2023; Baglioni et al., 2024; Fischer et al., 2024; Rubin et al., 2025), or to explicit renormalization of the kernel (Howard et al., 2025; Aiudi et al., 2025).

**Dynamical (training-time) theories**  A central line of work develops dynamical MF theory (DMFT) for deep networks, showing that a single control parameter, the effective learning rate, governs whether training remains in the lazy NNGP/NTK regime or enters a FL regime with evolving kernels. Lauditi et al. (2025) formalize this for deep architectures, proving exact analytic solutions in the deep *linear* case and introducing numerical solvers for *nonlinear* activations. Comparisons on synthetic Gaussian data and real networks (including CIFAR tasks) demonstrate close agreement between DMFT predictions and observed training, confirming that the MF reduction captures genuine FL even beyond linear models.

Earlier, Bordelon & Pehlevan (2022) derived DMFT saddle-point equations intended for broad classes of infinite-width networks. These equations are solvable in closed form for deep linear networks, while nonlinear cases require numerical approximation; experiments on deep linear models and wide CNNs trained on two-class CIFAR tasks showed that both loss dynamics and feature-kernel evolution match theory across widths and scaling laws. The same line of work identifies the effective learning rate as the parameter driving faster training, larger kernel movement, and stronger alignment to target functions, with Bordelon & Pehlevan (2023) extending these predictions to *finite-width* networks. Complementing this, Fischer et al. (2024) provide a DMFT description for two-layer nonlinear networks trained by SGD, yielding closed equations for kernels and fields (exact for linear networks; Monte-Carlo DMFT for nonlinear activations). Their experiments on Gaussian-mixture single-index tasks and binary CIFAR subsets again validate the MF predictions and the regime split controlled by effective learning rate. Rubin et al. (2025) generalize the framework to *arbitrary depth*, retaining exact solutions for deep linear networks and approximate

numerical solutions for nonlinear activations; experiments on Gaussian-mixture and single-index tasks and CIFAR subsets show that DMFT tracks real training behavior across regimes.

Orthogonal to DMFT but still dynamical, Mei et al. (2018) show that, in the infinite-width MF limit, SGD on two-layer networks follows a distributional-dynamics PDE in Wasserstein space. With noise or regularization this becomes a free-energy flow that guarantees global convergence. Unlike later DMFT work, they do not analyze NTK limits or a kernel–feature phase transition, instead they establish propagation-of-chaos, convergence of noisy SGD, and stability of fixed points in the noiseless case. Experiments on Gaussian classification and ReLU models show the PDE tracks SGD closely and diagnose when poor activations cause failure. In closely related simplified settings, Montanari & Urbani (2025) analyze *single-index* models with targets such as $h(z) = 0.9z + z^3/6$. Here neurons become exchangeable samples from a common evolving distribution that aligns with the latent direction, the DMFT equations admit exact analytic solutions (as in linear networks), and test error converges to the Bayes limit, with experiments confirming the theory's accuracy. Additional high-dimensional analyses further develop these dynamics (Celentano et al., 2025).

**Static (posterior-time) theories**  A complementary direction studies the learned posterior and reframes FL as *kernel adaptation*. Pacelli et al. (2023) reduce FL to *layerwise rescaling* of the kernel. Because the Bayesian predictor involves a matrix inverse, these rescalings act as mode-specific changes to the effective regularization of every kernel eigenmode, without rotating or performing relative scaling of feature directions. They derive analytic saddle-point equations for a pair of scalar descriptors per layer and, by solving them, obtain test-error predictions. On MNIST and CIFAR-10, where fully connected networks are not believed to do in strong representation learning, this coarse description already matches the observed bias–variance trade-off and the generalization error of SGLD-trained networks. Pushing this simplification further, Baglioni et al. (2024) collapse all finite-width effects into a *single global rescaling* of the infinite-width kernel, the resulting Bayesian effective action closely matches numerical experiments across MNIST and CIFAR-10.

Static renormalization viewpoints make the same idea explicit. Howard et al. (2025) interpret learning as a Wilsonian renormalization of the kernel, while Aiudi et al. (2025) show that architectural inductive bias matters: fully connected networks can be captured by a scalar kernel rescaling, but convolutional networks induce a *spatially indexed local kernel*. Training then learns a matrix $Q_{ij}^*$ that reweights patch–patch correlations, enabling selective emphasis or suppression of local interactions, this richer, weight-sharing–enabled mechanism provides a genuine route to FL that can outperform infinite-width kernel counterparts.

**Summary**  Across both dynamical and static perspectives, a coherent picture emerges. DMFT and related dynamical theories identify an effective learning-rate control parameter that cleanly separates lazy NTK behavior from FL with evolving kernels, with exact solutions in linear (and certain single-index) settings and accurate numerical solvers for nonlinear networks (Bordelon & Pehlevan, 2022; 2023; Lauditi et al., 2025; Fischer et al., 2024; Rubin et al., 2025; Mei et al., 2018; Montanari & Urbani, 2025; Celentano et al., 2025). Static Bayesian and renormalization accounts show that much of finite-width generalization can be captured by low-dimensional kernel *rescalings*, global, layerwise, or spatially local, without tracking full feature rotations (Pacelli et al., 2023; Baglioni et al., 2024; Howard et al., 2025; Aiudi et al., 2025). Together, these results support a simple, consistent theory that reproduces the characteristic shape and offsets of learning curves and clarifies how finite networks can break the apparent curse of dimensionality via structured kernel adaptation and alignment.

Below is new:

## C.1 COMPARISON WITH RECENT MEAN-FIELD WORKS

In this section, we clarify the relationship and fundamental differences between our MF-ARD framework and the recent work by van Meegen & Sompolinsky. While both works utilize mean-field approximations to study the posterior of neural networks, they address fundamentally different phenomena occurring at different levels of the network hierarchy. The confusion often stems from the term "sparsification," which refers to distinct mechanisms in the two papers.

## C.2 COORDINATE-WISE VS. NEURON-WISE SPARSIFICATION

The most critical distinction lies in the domain where symmetry breaking occurs:

- **(Neuron-wise Sparsification):** van Meegen & Sompolinsky investigates how the population of $N$ neurons organizes to represent the target. They find that the posterior distribution over the weight vectors $\{\boldsymbol{w}_1, \ldots, \boldsymbol{w}_N\}$ breaks permutation symmetry, resulting in a solution where only a subset of neurons ($O(1)$ out of $N$) strongly correlate with the target, while others remain agnostic or redundant. This is a textitinter-neuron phenomenon.

- **Our Work (Coordinate-wise Sparsification / IFS):** Our theory models the posterior of a *single representative neuron* $\boldsymbol{w} = [w_1, \ldots, w_d] \in \mathbb{R}^d$. We investigate how the symmetry between input coordinates $j = 1, \ldots, d$ is broken. We find that for tasks with isotropic data but low-dimensional targets, the posterior marginals $p(w_j)$ become heavy-tailed and high-variance only for task-relevant coordinates ($j \in S$), while remaining Gaussian and narrow for irrelevant coordinates ($j \notin S$). This is an *intra-neuron* phenomenon.

Standard mean-field theories (including the base model in van Meegen & Sompolinsky) typically assume that the weight distribution is isotropic over input coordinates unless the prior enforces otherwise. Consequently, while van Meegen & Sompolinsky successfully predicts that some neurons become "active" and others "dormant," it does not explicitly model the self-reinforcing mechanism that allows an *active* neuron to selectively amplify specific input coordinates to overcome the curse of dimensionality.

## C.3 UNIVERSALITY OF IFS VS. NONLINEARITY-DEPENDENT CODING SCHEMES

van Meegen & Sompolinsky emphasizes that the nature of the population code depends strongly on the nonlinearity (e.g., ReLU leads to sparse coding, Sigmoid leads to redundant coding).

In contrast, our Input Feature Selection (IFS) mechanism is largely agnostic to the specific choice of nonlinearity, provided the nonlinearity allows for feature learning. As shown in Figure 8 (reproduced from our rebuttal experiments), networks trained with Sigmoid activations exhibit the same coordinate-wise heavy-tailed distributions on relevant input dimensions as ReLU networks. While the population-level organization (sparse vs. redundant neurons) may differ as predicted by van Meegen & Sompolinsky, the fundamental mechanism of discovering the relevant input subspace via IFS remains the same.

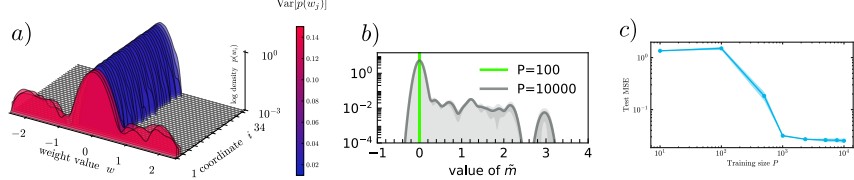

Figure 8: **IFS with Sigmoid Activations.** Similar to the ReLU results in the main text, SGLD training with Sigmoid activations leads to strong coordinate-wise symmetry breaking (high variance on relevant features $j \in S$). This demonstrates that IFS is a fundamental mechanism for subspace discovery, distinct from the neuron-level coding schemes discussed in van Meegen & Sompolinsky.

## C.4 THE NECESSITY OF COORDINATE ANISOTROPY: A CONTROL EXPERIMENT

To rigorously prove that coordinate-wise symmetry breaking (the core of our MF-ARD theory) is the causal mechanism for beating the kernel regime, we performed a control experiment enforcing coordinate homogeneity.

We trained networks on the $k$-sparse parity task using a modified loss function that penalizes anisotropy within neurons, without penalizing sparsity across neurons:

$$\ell_{\text{total}} = \ell_{\text{MSE}} + \lambda_h \sum_{j=1}^{N} \left[ \frac{1}{d} \sum_{i=1}^{d} \left( |w_{ij}| - \left( \frac{1}{d} \sum_{k=1}^{d} |w_{kj}| \right) \right)^2 \right]. \tag{46}$$

**Result:** As $\lambda_h$ increases, the learning curve of the neural network reverts to the NNGP learning curve (see Figure 9). Even if the network is allowed to sparsify at the neuron level (some neurons large, some small), suppressing the variance *between coordinates* $w_{ij}$ prevents the network from learning the target efficiently.

This confirms that the performance gap between Finite-Width NNs and Kernels on isotropic data is driven by the mechanism modeled in this paper (IFS/Coordinate Anisotropy), a mechanism that is distinct from and complementary to the population-level phenomena described in van Meegen & Sompolinsky.

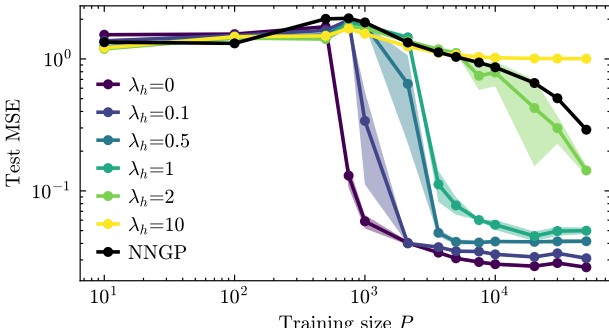

Figure 9: **Suppressing IFS kills performance.** Generalization error for $k$-sparse parity. As we increase the regularization $\lambda_h$ that forces coordinates within a neuron to be homogeneous (preventing IFS), the NN performance degrades back to the Kernel (NNGP) limit. This confirms that coordinate-wise anisotropy is the necessary condition for sample-efficient learning in this setting.

# D  PROOFS

This is new

## D.1  MECHANISM: INPUT FEATURE SELECTION VIA DYNAMIC ANISOTROPY

Standard MF theory suffers from a 'homogeneity gap': it imposes an isotropic prior $\mathcal{N}(\mathbf{0}, \sigma_w^2/d \cdot \mathbf{I})$ that distributes regularization pressure uniformly across all $d$ input dimensions. In contrast, SGLD dynamics are inherently anisotropic. Gradients along task-relevant directions ($j \in S$) accumulate constructively, reducing the effective local curvature of the potential and allowing weights to diffuse away from the origin. Conversely, along noise directions ($j \notin S$), gradients effectively average to zero, leaving the restoring force of the weight decay to dominate.

Our first-principles derivation (Theorem D.1) captures this symmetry breaking by identifying the *effective stiffness* $\rho_j$ as a dynamical variable rather than a fixed hyperparameter. The resulting fixed-point condition, $\rho_j \propto (\beta_0 + \frac{N}{2}\langle w_j^2 \rangle)^{-1}$, induces a self-reinforcing feedback loop that we term *Input Feature Selection* (IFS):

- **Signal Amplification:** For relevant features, the gradient signal increases the second moment $\langle w_j^2 \rangle$. This decreases the effective stiffness $\rho_j$, locally relaxing the regularization penalty and allowing further weight growth.
- **Noise Suppression:** For irrelevant features, the lack of gradient signal keeps $\langle w_j^2 \rangle$ small. This maintains a high stiffness $\rho_j \approx d/\sigma_w^2$, aggressively confining these weights to the origin via an Ornstein-Uhlenbeck-like process.

Crucially, this mechanism enables the network to overcome the curse of dimensionality. By effectively setting $\rho_{j \notin S} \to \infty$, the posterior mass collapses onto the task-relevant subspace, reducing the effective problem dimension from the ambient $d$ to the intrinsic $k \ll d$.

**Theorem D.1** (Derivation of ARD Updates via Regularized SGLD Stationarity). *Consider a one-hidden-layer network with weights $\boldsymbol{w}_i$ for neuron $i \in \{1, \ldots, N\}$. We posit that the update equation for the effective precision $\rho_j$ is the unique minimizer of the effective free energy defined by SGLD stationarity, subject to finite-width corrections and consistency with the free dynamics of the optimizer.*

*We make the following assumptions:*

(A1) **SGLD Dynamics.** *The parameters evolve via the stochastic differential equation governed by the posterior $p_{\mathrm{GD}}(\boldsymbol{\theta}|\mathcal{D})$:*

$$dw_{ij}(t) = -\nabla_{w_{ij}} \left( -\ln p_{\mathrm{GD}}(\boldsymbol{\theta}) \right) dt + \sqrt{2}\, dB_{ij}(t), \tag{47}$$

*where $-\ln p_{\mathrm{GD}} = -\ln p_{prior} - \ln p_L$.*

(A2) **Effective Quadratic Potential (Linear Response).** *In the stationary state, we assume the gradient of the likelihood term (the data interaction) can be approximated by an effective coordinate-wise stiffness $\tilde{\rho}_j$. Specifically, we assume the linear response relation:*

$$\mathbb{E}\left[ w_{ij} \nabla_{w_{ij}} (-\ln p_L) \right] \approx \tilde{\rho}\, \mathbb{E}[w_{ij}^2], \tag{48}$$

*where $Q_j := \mathbb{E}[w_{ij}^2]$ is the stationary second moment and $-\ln p_L = \frac{1}{2\kappa^2 P} \sum_\mu (f_{\boldsymbol{\theta}}(\boldsymbol{x}_\mu) - \boldsymbol{y}_\mu)^2$.*

(A3) **Consistency with Free Dynamics.** *The effective free energy $\mathcal{F}(\rho_j)$ includes a One-Loop entropy correction $(-\frac{N}{2} \ln \rho_j)$ accounting for finite-width fluctuations. Crucially, we impose a boundary condition: in the limit where the data signal vanishes ($\tilde{\rho}_j \to 0$), the fixed point of the effective theory must recover the exact stationary variance of the free dynamics (the Ornstein-Uhlenbeck process governed solely by the prior $p_{prior}$).*

*Result: The optimal effective precision $\rho_j$ satisfies the fixed-point equation:*

$$\rho_j = \frac{\alpha_0 + \frac{N}{2}}{\beta_0 + \frac{N}{2} Q_j}, \tag{49}$$

*where $\alpha_0$ is a stiffness hyperparameter and $\beta_0$ is calibrated to the isotropic initialization variance $\sigma_w^2/d$.*

*Proof.* The proof proceeds by constructing the effective free energy from the dynamical stationarity conditions and solving for its minimizer.

**1. Dynamical Stationarity (The Energy Term).** By Itô's Lemma, the evolution of the squared weight $w_{ij}^2$ is given by $d(w_{ij}^2) = 2w_{ij} dw_{ij} + (dw_{ij})^2$. Substituting the SGLD dynamics (A1):

$$d(w_{ij}^2) = 2w_{ij} \left( -\nabla_{w_{ij}}(-\ln p_{\mathrm{GD}})dt + \sqrt{2}dB_{ij} \right) + 2dt. \tag{50}$$

Taking the expectation with respect to the stationary distribution, the drift vanishes ($\frac{d}{dt}\mathbb{E}[w^2] = 0$), and the stochastic increment averages to zero, yielding the equipartition identity:

$$\mathbb{E}\left[ w_{ij} \nabla_{w_{ij}}(-\ln p_{\mathrm{GD}}) \right] = 1. \tag{51}$$

We decompose the gradients into prior and likelihood components. Using the Gaussian prior form from Eq. 4, $-\ln p_{\mathrm{prior}} = \sum_{i,k} \frac{d}{2\sigma_w^2} w_{ik}^2$:

$$\mathbb{E}\left[ w_{ij} \left( \frac{d}{\sigma_w^2} w_{ij} + \nabla_{w_{ij}}(-\ln p_{\mathrm{L}}) \right) \right] = 1. \tag{52}$$

Applying the Linear Response assumption (A2) to the likelihood term:

$$\frac{d}{\sigma_w^2} \mathbb{E}[w_{ij}^2] + \tilde{\rho}_j \mathbb{E}[w_{ij}^2] = 1 \quad \Longrightarrow \quad Q_j \left( \frac{d}{\sigma_w^2} + \tilde{\rho}_j \right) = 1. \tag{53}$$

This identifies the effective precision contribution from the potential as $\rho_{\mathrm{eff}} = \frac{d}{\sigma_w^2} + \tilde{\rho}_j = Q_j^{-1}$.

Summed over $N$ neurons, the energetic cost associated with precision $\rho_j$ is $E_{\mathrm{weights}}(\rho_j) = \frac{N}{2} \rho_j Q_j$.

**2. Consistency with Free Dynamics (The Regularization).** To determine the entropic and regularization terms, we consider the "free dynamics" limit where the likelihood is flat ($p_L = \text{const}$, $\tilde{\rho}_j = 0$). In this case, the SGLD equation decouples into a standard Ornstein-Uhlenbeck process driven only by the prior:

$$dw_{ij}(t) = -\frac{d}{\sigma_w^2} w_{ij} dt + \sqrt{2} dB_{ij}(t). \tag{54}$$

The exact stationary solution of this process is a Gaussian with variance $\sigma_w^2/d$. Any valid effective theory for $\rho_j$ must possess a stable minimum at $\rho_{\text{free}} = d/\sigma_w^2$ in this limit. We construct the free energy $\mathcal{F}(\rho_j)$ by adding the One-Loop entropy correction $S = -\frac{N}{2} \ln \rho_j$ (Assumption A3) and a convex regularization potential $R(\rho_j) = \beta_0 \rho_j - \alpha_0 \ln \rho_j$. The coefficients $\alpha_0, \beta_0$ are calibrated to match the free dynamics.

**3. Free Energy Minimization.** The total effective free energy is:

$$\mathcal{F}(\rho_j) = \frac{N}{2} \rho_j Q_j - \frac{N}{2} \ln \rho_j + \beta_0 \rho_j - \alpha_0 \ln \rho_j. \tag{55}$$

Minimizing with respect to $\rho_j$ ($\partial_{\rho_j} \mathcal{F} = 0$) yields:

$$\frac{N}{2} Q_j - \frac{N}{2\rho_j} + \beta_0 - \frac{\alpha_0}{\rho_j} = 0 \quad \Longrightarrow \quad \rho_j \left( \frac{N}{2} Q_j + \beta_0 \right) = \frac{N}{2} + \alpha_0. \tag{56}$$

Solving for $\rho_j$:

$$\rho_j = \frac{\alpha_0 + \frac{N}{2}}{\beta_0 + \frac{N}{2} Q_j}. \tag{57}$$

Finally, applying the consistency condition: when $\tilde{\rho}_j \to 0$, $Q_j \to \sigma_w^2/d$. The fixed point must be $\rho_{\text{free}} = d/\sigma_w^2$. This requires fixing the ratio $\alpha_0/\beta_0 = d/\sigma_w^2$ (or $\beta_0 = \alpha_0 \sigma_w^2/d$), which ensures the ARD prior matches the isotropic initialization in expectation. □

### D.2 KERNEL LIMIT

**Theorem D.2.** *With $\gamma = 1/2$ the infinite width limit has the following FP equations:*

$$m_A^\infty = \frac{\sigma_a^2}{\kappa^2} \left( \mathbb{E}_{\boldsymbol{w} \sim p_\infty}[J_A(\boldsymbol{w}) J_{\mathcal{Y}}(\boldsymbol{w})] - \sum_B \mathbb{E}_{\boldsymbol{w} \sim p_\infty}[J_A(\boldsymbol{w}) J_B(\boldsymbol{w})] m_B^\infty \right). \tag{58}$$

*Proof.* The proof proceeds by taking the $N \to \infty$ limit of the self-consistency equations derived from the cavity method's free energy.

**1. Self-Consistency from Free Energy.** We begin with the MF free energy functional derived from the cavity method:

$$\mathcal{F}(\{m_A\}) = \frac{1}{2\kappa^2} \sum_A (y_A - m_A)^2 - N \ln \mathcal{Z}_1(\{m_A\}) \tag{59}$$

where $\mathcal{Z}_1(\{m_A\}) = \int d\boldsymbol{w} \, e^{-\bar{S}_{\text{eff}}^{(\infty)}(\boldsymbol{w}; \{m_A\})}$ is the single-neuron partition function. The effective action for a single weight vector $\boldsymbol{w}$, after integrating out the amplitude $a$, is:

$$\bar{S}_{\text{eff}}^{(\infty)}(\boldsymbol{w}; \{m_A\}) = \frac{d}{2\sigma_w^2} \|\boldsymbol{w}\|^2 + \frac{1}{2} \ln \left( \frac{1}{\sigma_a^2} + \frac{\Sigma(\boldsymbol{w})}{N^{2\gamma}\kappa^2} \right) - \frac{(J_{\mathcal{Y}}(\boldsymbol{w}) - \sum_A m_A J_A(\boldsymbol{w}))^2}{2\kappa^4 N^{2\gamma} \left( \frac{1}{\sigma_a^2} + \frac{\Sigma(\boldsymbol{w})}{N^{2\gamma}\kappa^2} \right)} \tag{60}$$

The equilibrium state is found by the stationarity condition $\partial \mathcal{F}/\partial m_A = 0$, which yields the self-consistency equation:

$$m_A = N^{1-\gamma} \langle \mu(\boldsymbol{w}) J_A(\boldsymbol{w}) \rangle_{p(\boldsymbol{w}|\{m_B\})} \tag{61}$$

where $p(\boldsymbol{w}|\{m_B\}) = \frac{1}{\mathcal{Z}_1} e^{-\bar{S}_{\text{eff}}^{(\infty)}}$ is the posterior distribution on a single neuron's weights, and $\mu(\boldsymbol{w})$ is the posterior mean of its amplitude $a$:

$$\mu(\boldsymbol{w}) = \frac{\frac{1}{\kappa^2 N^\gamma}\left(J_{\mathcal{Y}}(\boldsymbol{w}) - \sum_B m_B J_B(\boldsymbol{w})\right)}{\frac{1}{\sigma_a^2} + \frac{\Sigma(\boldsymbol{w})}{N^{2\gamma}\kappa^2}} \tag{62}$$

**2. The Infinite-Width Limit with Critical Scaling.** To obtain a non-trivial limit as $N \to \infty$, we must set the scaling to the critical value $\gamma = 1/2$. For $\gamma > 1/2$, the prefactor $N^{1-\gamma} \to 0$, decoupling the neurons and leading to $m_A \to 0$. For $\gamma < 1/2$, the interaction term diverges. With $\gamma = 1/2$, the terms of order $1/N$ in the effective action $\bar{S}_{\text{eff}}^{(\infty)}$ vanish. Specifically:

$$\frac{\Sigma(\boldsymbol{w})}{N^{2\gamma}\kappa^2} = \frac{\Sigma(\boldsymbol{w})}{N\kappa^2} \xrightarrow{N\to\infty} 0 \tag{63}$$

As a result, the data-dependent and $m_A$-dependent terms in the exponent of $p(\boldsymbol{w}|\{m_A\})$ vanish. The distribution over weights collapses to its prior:

$$p(\boldsymbol{w}|\{m_A\}) \xrightarrow{N\to\infty} p_\infty(\boldsymbol{w}) \propto \exp\left(-\frac{d}{2\sigma_w^2}\|\boldsymbol{w}\|^2\right) \tag{64}$$

Simultaneously, the denominator in $\mu(\boldsymbol{w})$ simplifies to $1/\sigma_a^2$. The expression for the posterior mean amplitude becomes:

$$\mu(\boldsymbol{w}) \xrightarrow{N\to\infty} \frac{\sigma_a^2}{\kappa^2 N^{1/2}}\left(J_{\mathcal{Y}}(\boldsymbol{w}) - \sum_B m_B J_B(\boldsymbol{w})\right) \tag{65}$$

**3. Deriving the Linear System.** We now substitute these limiting forms back into the self-consistency equation, using $m_A^\infty$ to denote the solution in this limit:

$$m_A^\infty = N^{1-1/2}\left\langle \left[\frac{\sigma_a^2}{\kappa^2 N^{1/2}}\left(J_{\mathcal{Y}}(\boldsymbol{w}) - \sum_B m_B^\infty J_B(\boldsymbol{w})\right)\right] J_A(\boldsymbol{w})\right\rangle_{p_\infty(\boldsymbol{w})} \tag{66}$$

$$m_A^\infty = \frac{\sigma_a^2}{\kappa^2}\left\langle \left(J_{\mathcal{Y}}(\boldsymbol{w}) - \sum_B m_B^\infty J_B(\boldsymbol{w})\right) J_A(\boldsymbol{w})\right\rangle_{p_\infty(\boldsymbol{w})} \tag{67}$$

$$m_A^\infty = \frac{\sigma_a^2}{\kappa^2}\left(\mathbb{E}_{\boldsymbol{w}\sim p_\infty}[J_A(\boldsymbol{w})J_{\mathcal{Y}}(\boldsymbol{w})] - \sum_B \mathbb{E}_{\boldsymbol{w}\sim p_\infty}[J_A(\boldsymbol{w})J_B(\boldsymbol{w})]m_B^\infty\right) \tag{68}$$

Let us define the NNGP kernel matrix $K$ with elements $K_{AB} := \mathbb{E}_{\boldsymbol{w}\sim p_\infty}[J_A(\boldsymbol{w})J_B(\boldsymbol{w})]$ and the data-kernel coupling vector $\Xi$ with elements $\Xi_A := \mathbb{E}_{\boldsymbol{w}\sim p_\infty}[J_A(\boldsymbol{w})J_{\mathcal{Y}}(\boldsymbol{w})] = \sum_S y_S K_{AS}$. The equation becomes a linear system for the vector $m^\infty$:

$$m^\infty = \frac{\sigma_a^2}{\kappa^2}(\Xi - Km^\infty) = \frac{\sigma_a^2}{\kappa^2}(Ky - Km^\infty) \tag{69}$$

**4. Solution as Kernel Ridge Regression.** Rearranging the linear system, we get:

$$\left(I + \frac{\sigma_a^2}{\kappa^2}K\right)m^\infty = \frac{\sigma_a^2}{\kappa^2}Ky \tag{70}$$

$$\left(\frac{\kappa^2}{2\sigma_a^2}I + \frac{1}{2}K\right)m^\infty = \frac{1}{2}Ky \tag{71}$$

Let the ridge be $\tau = \kappa^2/(2\sigma_a^2)$. The equation is $(\tau I + K)m^\infty = Ky$. Solving for $m^\infty$ gives the final KRR solution:

$$m^\infty = K(K + \tau I)^{-1}y \tag{72}$$

This completes the proof. The solution depends only on the NNGP kernel $K$, which is fixed by the network architecture and priors, not the training data labels. This demonstrates the absence of FL in this specific infinite-width limit. $\qquad\square$

## D.3 INTEGRATING OUT $a$

As the action is quadratic in $a$ we can integrate it out.

**Theorem D.3.** *The distribution is given by*

$$S_{\text{MF}}(\boldsymbol{w}, \{m_A\}) = const. + \frac{1}{2} \ln \left( \frac{1}{\sigma_a^2} + \frac{\Sigma(\boldsymbol{w})}{N^{2\gamma}\kappa^2} \right) - \frac{(J_{\mathcal{Y}}(\mathbf{w}) - \sum_A m_A J_A(\mathbf{w}))^2}{2N^{2\gamma}\kappa^4 \left( \frac{1}{\sigma_a^2} + \frac{\Sigma(\boldsymbol{w})}{N^{2\gamma}\kappa^2} \right)} \tag{73}$$

$$+ \frac{d}{2\sigma_w^2} \|\boldsymbol{w}\|^2 \tag{74}$$

$$p(a|\boldsymbol{w}) = \mathcal{N}(\mu(\boldsymbol{w}), \sigma^2(\boldsymbol{w})) \tag{75}$$

$$\sigma(\boldsymbol{w})^2 = \frac{1}{2\alpha} = \left( \frac{1}{\sigma_a^2} + \frac{\Sigma(\boldsymbol{w})}{N^{2\gamma}\kappa^2} \right)^{-1} \tag{76}$$

$$\mu(\boldsymbol{w}) = \sigma^2 \beta = \frac{\beta}{2\alpha} = \frac{\frac{1}{N^{\gamma}\kappa^2} (J_{\mathcal{Y}}(\mathbf{w}) - \sum_A m_A J_A(\mathbf{w}))}{\left( \frac{1}{\sigma_a^2} + \frac{\Sigma(\boldsymbol{w})}{N^{2\gamma}\kappa^2} \right)} \tag{77}$$

*Proof.* We start with the standard Gaussian integral

$$\int da d\boldsymbol{w} e^{-[\alpha \cdot a^2 - \beta \cdot a + c]} = \int d\boldsymbol{w} \sqrt{\frac{\pi}{\alpha}} e^{\frac{\beta^2}{4\alpha} - c} \tag{78}$$

We have

$$\mathcal{S}_{\text{MF}}(\boldsymbol{w}, a, \{m_A\}) = \frac{1}{2\sigma_a^2} \sum_{i=1}^N a_i^2 + \frac{d}{2\sigma_w^2} \sum_{i=1}^d \|\boldsymbol{w}_i\|^2 + \frac{a^2}{2\kappa^2 N^{2\gamma}} \Sigma(\boldsymbol{w}) \tag{79}$$

$$- \frac{a}{\kappa^2 N^{\gamma}} \left( J_{\mathcal{Y}}(\mathbf{w}) - \sum_A m_A J_A(\mathbf{w}) \right) \tag{80}$$

For 1 neuron we get

$$\mathcal{S}_{\text{MF}}(\boldsymbol{w}, a, \{m_A\}) = a^2 \left( \frac{1}{2\sigma_a^2} + \frac{\Sigma(\boldsymbol{w})}{2\kappa^2 N^{2\gamma}} \right) - \frac{a}{\kappa^2 N^{\gamma}} \left( J_{\mathcal{Y}}(\mathbf{w}) - \sum_A m_A J_A(\mathbf{w}) \right) + \frac{d}{2\sigma_w^2} \|\boldsymbol{w}\|^2 \tag{81}$$

with

$$\alpha := \frac{1}{2\sigma_a^2} + \frac{\Sigma(w)}{2N^{2\gamma}\kappa^2}, \tag{82}$$

$$\beta := \frac{J_{\mathcal{Y}}(w) - \sum_A m_A J_A(w)}{N^{\gamma}\kappa^2}. \tag{83}$$

$$c = \frac{d}{2\sigma_w^2} \|\boldsymbol{w}\|^2 \tag{84}$$

and $\alpha(a - \frac{\beta}{2\alpha})^2 - \frac{\beta^2}{4\alpha} + c$ comparing to $\frac{-(a-\mu)^2}{2\sigma^2}$ we get $\alpha = \frac{1}{2\sigma^2}$ and $\mu = \frac{\beta}{2\alpha}$ for the Gaussian identification. This gives

$$\int d\boldsymbol{w} \sqrt{\frac{\pi}{\alpha}} e^{\frac{\beta^2}{4\alpha} - c} = \int d\boldsymbol{w} e^{-[\frac{1}{2}\ln(\pi) - \frac{1}{2}\ln(\alpha) + \frac{\beta^2}{4\alpha} - c]} \tag{85}$$

where the exponent is identified as the effective action. $\qquad \square$

## D.4 MF- FP EQUATION

**Lemma D.4.** *Consider a single target mode $y(\boldsymbol{x}) = \chi_S(\boldsymbol{x})$ and assume other overlaps vanish. Using the self–consistency $m_A = N^{1-\gamma} \langle a\, J_A(\boldsymbol{w}) \rangle$ and the conditional mean $\mu(\boldsymbol{w})$ above, we obtain*

$$m_S = N^{1-\gamma} \Big\langle \mu(\boldsymbol{w})\, J_S(\boldsymbol{w}) \Big\rangle = \frac{N^{1-2\gamma}}{\kappa^2}(1-m_S) \left\langle \frac{J_S(\boldsymbol{w})^2}{\sigma_a^{-2} + \dfrac{\Sigma(\boldsymbol{w})}{\kappa^2 N^{2\gamma}}} \right\rangle_{\boldsymbol{w}\sim p(\boldsymbol{w}|m_S)}. \tag{86}$$

*and in the $\kappa \to 0$ limit it is*

$$m_S \approx (1-m_S)\, N \left\langle \frac{J_S(\boldsymbol{w})^2}{\Sigma(\boldsymbol{w})} \right\rangle_{\boldsymbol{w}\sim p(\boldsymbol{w}|m_S)} \tag{87}$$

*Proof.* Consider a single target mode $y(\boldsymbol{x}) = \chi_S(\boldsymbol{x})$ and assume other overlaps vanish. Using the self–consistency $m_A = N^{1-\gamma} \langle a\, J_A(\boldsymbol{w}) \rangle$ and the conditional mean $\mu(\boldsymbol{w})$ above, we obtain

$$m_S = N^{1-\gamma} \Big\langle \mu(\boldsymbol{w})\, J_S(\boldsymbol{w}) \Big\rangle = \frac{N^{1-2\gamma}}{\kappa^2}(1-m_S) \left\langle \frac{J_S(\boldsymbol{w})^2}{\sigma_a^{-2} + \dfrac{\Sigma(\boldsymbol{w})}{\kappa^2 N^{2\gamma}}} \right\rangle_{\boldsymbol{w}\sim p(\boldsymbol{w}|m_S)}.$$

In the small–noise regime $\kappa \to 0$ (with $\Sigma(\boldsymbol{w}) > 0$), the denominator simplifies as $\sigma_a^{-2} + \frac{\Sigma(\boldsymbol{w})}{\kappa^2 N^{2\gamma}} \approx \frac{\Sigma(\boldsymbol{w})}{\kappa^2 N^{2\gamma}}$, so that

$$m_S \approx (1-m_S)\, N \left\langle \frac{J_S(\boldsymbol{w})^2}{\Sigma(\boldsymbol{w})} \right\rangle_{\boldsymbol{w}\sim p(\boldsymbol{w}|m_S)} \implies m_S \approx \frac{N\Big\langle J_S(\boldsymbol{w})^2/\Sigma(\boldsymbol{w}) \Big\rangle}{1 + N\Big\langle J_S(\boldsymbol{w})^2/\Sigma(\boldsymbol{w}) \Big\rangle}$$

By Cauchy–Schwarz, $J_S(\boldsymbol{w})^2 \le \Sigma(\boldsymbol{w})$, so $0 \le m_S < 1$ unless all mass concentrates on perfectly aligned $\boldsymbol{w}$. $\square$

## D.5 SOLUTION TO THE FIXED POINT EQUATION

**Lemma D.5.** *Consider $\phi = \mathrm{ReLU}$. Let the inputs $x_j \in \{\pm 1\}$ be i.i.d. and $y(\boldsymbol{x}) = \chi_S(\mathbf{x}) = \prod_{j\in S} x_j$ with $S = \{0,1,...,k-1\}$. We define $R_k(\boldsymbol{w}) = \frac{J_S(\boldsymbol{w})^2}{\Sigma(\boldsymbol{w})}$. It holds that*

$$\boldsymbol{w}^* = \max_{\boldsymbol{w}} R_k(\boldsymbol{w}) \tag{88}$$

*is given by $\boldsymbol{w}^* = (\overbrace{\alpha,\dots,\alpha}^{k}, 0, ..., 0)$.*

*Proof.* Decompose $w_S = \alpha \frac{\mathbb{1}_S}{\sqrt{k}} + u$, $\quad u \perp \mathbb{1}_S$ and keep arbitrary $\boldsymbol{w}^C$. Because the data distribution is invariant to permutations of the $k$ coordinates in $S$ the only $S$-dependent statistic that survives in $\chi_S$-weighted expectations is the sum $s(\boldsymbol{x}) = \sum_{j\in S} x_j$. Any component orthogonal to $\mathbb{1}_S$ averages out in $J_S$ but increases $\Sigma(\boldsymbol{w})$ (by convexity of $x \mapsto \phi(x)^2$). Likewise, weights in $S^C$ contribute variance to $\Sigma$ but contribute nothing to $J_S$ (they are independent of $\chi_S$. and average to zero under the sign symmetry). Thus the maximizer of $R_k(\boldsymbol{w})$ lives in the span of $\mathbb{1}_S$ and $\boldsymbol{w}^C$. $\square$

**Lemma D.6.** *Consider $\phi = \mathrm{ReLU}$. Let the inputs $x_j \in \{\pm 1\}$ be i.i.d. and $y(\boldsymbol{x}) = \chi_S(\mathbf{x}) = \prod_{j\in S} x_j$ with $|S| = k$. We assume $\boldsymbol{w}^*$ from Theorem D.5. Then, the ratio of the squared neuron-target coupling $J_S(\mathbf{w})^2$ to the neuron's self-energy $\Sigma(\mathbf{w})$ is a constant $R_k$ independent of the scale $\alpha$.*

*Proof.* Let $r \sim \mathrm{Binomial}(k, 1/2)$ be the number of components $x_j = -1$ for $j \in S$. The inner product is $s(\mathbf{x}) = \mathbf{w}^\top \mathbf{x} = \alpha \sum_{j\in S} x_j = \alpha(k - 2r)$, and the target function is $\chi_S(\mathbf{x}) = (-1)^r$.

The neuron-target coupling $J_S(\mathbf{w})$ is the expectation $\mathbb{E}[\phi(\mathbf{w}^\top \mathbf{x}) \chi_S(\mathbf{x})]$.

$$J_S(\mathbf{w}) = \mathbb{E}\left[\alpha \cdot [k - 2r]_+ \cdot (-1)^r\right] \tag{89}$$

$$= \alpha \sum_{r=0}^{k} P(r) \cdot [k - 2r]_+ \cdot (-1)^r \tag{90}$$

$$= \alpha \cdot 2^{-k} \sum_{r=0}^{\lfloor (k-1)/2 \rfloor} \binom{k}{r} (k - 2r)(-1)^r \tag{91}$$

where $[z]_+ = \max(0, z)$, and the sum is restricted to the terms where $k - 2r > 0$.

The neuron's self-energy $\Sigma(\mathbf{w})$ is the expectation $\mathbb{E}[\phi(\mathbf{w}^\top \mathbf{x})^2]$.

$$\Sigma(\mathbf{w}) = \mathbb{E}\left[(\alpha \cdot [k - 2r]_+)^2\right] \tag{92}$$

$$= \alpha^2 \sum_{r=0}^{k} P(r) \cdot [k - 2r]_+^2 \tag{93}$$

$$= \alpha^2 \cdot 2^{-k} \sum_{r=0}^{\lfloor (k-1)/2 \rfloor} \binom{k}{r} (k - 2r)^2 \tag{94}$$

We define the scale-independent constants $D_k$ and $C_k$:

$$D_k := 2^{-k} \sum_{r=0}^{\lfloor (k-1)/2 \rfloor} \binom{k}{r} (k - 2r)(-1)^r \tag{95}$$

$$C_k := 2^{-k} \sum_{r=0}^{\lfloor (k-1)/2 \rfloor} \binom{k}{r} (k - 2r)^2 \tag{96}$$

such that $J_S(\mathbf{w}) = \alpha D_k$ and $\Sigma(\mathbf{w}) = \alpha^2 C_k$. The ratio is then

$$R_k := \frac{J_S(\mathbf{w})^2}{\Sigma(\mathbf{w})} = \frac{(\alpha D_k)^2}{\alpha^2 C_k} = \frac{D_k^2}{C_k}$$

which is independent of $\alpha$, thus proving the proposition.

$\square$

### D.6 EXACT SOLUTION OF THE FP EQUATION

**Theorem D.7.** *Using the setup from Theorem D.6, especially the proposed $\boldsymbol{w}^*$, the FP is given by*

$$m_S = (1 - m_S) N R_k \left(1 - \frac{\sigma_a^{-2}}{A^\star(m_S)}\right), \quad A^\star(m_S) = \frac{-C_k + \sqrt{C_k^2 + 4\left(\frac{dk}{\sigma_w^2}\right) N^{2\gamma} (1 - m_S)^2 D_k^2 \sigma_a^{-2}}}{2\left(\frac{dk}{\sigma_w^2}\right) \kappa^2 N^{2\gamma}} \tag{97}$$

*Proof.* From Theorem D.4, we know:

$$m_S = \frac{N^{1-2\gamma}}{\kappa^2} (1 - m_S) \left\langle \frac{J_S(\mathbf{w})^2}{\sigma_a^{-2} + \Sigma(\mathbf{w})/(\kappa^2 N^{2\gamma})} \right\rangle_{\mathbf{w}|m_S}$$

Assuming the posterior distribution of weights is sharply peaked around the optimal direction given by the ansatz in Theorem D.6, the expectation $\langle \cdot \rangle_{\mathbf{w}|m_S}$ reduces to an evaluation at the optimal weight scale $\alpha^\star$. The value of $\alpha^\star$ is determined by minimizing the single-neuron effective action $\mathcal{S}_{\text{eff}}(\alpha)$:

$$\mathcal{S}_{\text{eff}}(\alpha) = \underbrace{\frac{dk}{2\sigma_w^2} \alpha^2}_{\text{Weight Prior}} + \underbrace{\frac{1}{2} \ln A(\alpha)}_{\text{Normalizer}} - \underbrace{\frac{1}{2} \frac{[(1 - m_S) J_S(\alpha)]^2 / (\kappa^2 N^{2\gamma})}{A(\alpha)}}_{\text{Data Gain}}$$

where $A(\alpha) = \sigma_a^{-2} + \Sigma(\alpha)/(\kappa^2 N^{2\gamma}) = \sigma_a^{-2} + \alpha^2 C_k/(\kappa^2 N^{2\gamma})$.

Setting $\frac{\partial \mathcal{S}_{\text{eff}}}{\partial \alpha} = 0$ yields a quadratic equation for the optimal value $A^\star = A(\alpha^\star)$:

$$\left(\frac{dk}{\sigma_w^2}\right) \kappa^2 N^{2\gamma} \left(A^\star\right)^2 + C_k\, A^\star - \frac{(1-m_S)^2 D_k^2 \sigma_a^{-2}}{\kappa^2} = 0$$

The positive root of this equation gives the solution for $A^\star(m_S)$:

$$A^\star(m_S) = \frac{-C_k + \sqrt{C_k^2 + 4\left(\frac{dk}{\sigma_w^2}\right) N^{2\gamma}(1-m_S)^2 D_k^2\, \sigma_a^{-2}}}{2\left(\frac{dk}{\sigma_w^2}\right) \kappa^2 N^{2\gamma}}$$

We now substitute this back into the self-consistency equation. Using the definitions of $J_S$, $\Sigma$, and $A^\star$, we have $(\alpha^\star)^2 = \frac{\kappa^2 N^{2\gamma}}{C_k}(A^\star - \sigma_a^{-2})$.

$$m_S = (1-m_S)\frac{N^{1-2\gamma}}{\kappa^2}\frac{J_S(\alpha^\star)^2}{A^\star} \tag{98}$$

$$= (1-m_S)\frac{N^{1-2\gamma}}{\kappa^2}\frac{(\alpha^\star)^2 D_k^2}{A^\star} \tag{99}$$

$$= (1-m_S)\frac{N^{1-2\gamma}}{\kappa^2}\frac{1}{A^\star}\left[\frac{\kappa^2 N^{2\gamma}}{C_k}(A^\star - \sigma_a^{-2})\right] D_k^2 \tag{100}$$

$$= (1-m_S)N\frac{D_k^2}{C_k}\frac{A^\star - \sigma_a^{-2}}{A^\star} \tag{101}$$

$$= (1-m_S)NR_k\left(1 - \frac{\sigma_a^{-2}}{A^\star(m_S)}\right) \tag{102}$$

This gives the final fixed-point equation for the order parameter $m_S$. $\qquad\square$

**Theorem D.8.** *Consider the fixed–point equation in the infinite $P$-limit*

$$m_S = (1-m_S)\,NR_k\left(1 - \frac{\sigma_a^{-2}}{A^\star(m_S)}\right), \tag{103}$$

*with $A^\star(m_S)$ given implicitly as the positive root of*

$$\left(\frac{dk}{\sigma_w^2}\right)\kappa^2 N^{2\gamma}\left(A^\star\right)^2 + C_k\,A^\star - (1-m_S)^2 D_k^2\,\sigma_a^{-2} = 0, \tag{104}$$

*and define $C := \frac{dk}{\sigma_w^2}$. Then the* critical noise *level*

$$\kappa_c^2 = \frac{\sqrt{C_k^2 + 4\,C\,N^{2\gamma}D_k^2\sigma_a^{-2}} - C_k}{2\,C\,\sigma_a^{-2}\,N^{2\gamma}} \tag{105}$$

*marks a phase transition:*

*(i) If $\kappa^2 \geq \kappa_c^2$, then $A^\star(0) \leq \sigma_a^{-2}$ and $m_S = 0$ is a fixed point; in particular for $\kappa^2 = \kappa_c^2$ we have $A^\star(0) = \sigma_a^{-2}$ and the only solution is $m_S = 0$.*

*(ii) If $\kappa^2 < \kappa_c^2$, then $A^\star(0) > \sigma_a^{-2}$ and there exists a unique nontrivial solution $m_S \in (0,1)$, thus the system exhibits symmetry breaking $m_S : 0 \to m_S > 0$ as $\kappa$ crosses $\kappa_c$ from above.*

*Proof.* Set $m_S = 0$ in equation 104 to get

$$C\kappa^2 N^{2\gamma}\left(A^\star\right)^2 + C_k A^\star - D_k^2\sigma_a^{-2} = 0. \tag{106}$$

At the onset of FL the trivial fixed point $m_S = 0$ changes stability precisely when the right–hand side of the FP map ceases to vanish at $m_S = 0$, i.e. when

$$NR_k\left(1 - \frac{\sigma_a^{-2}}{A^\star(0)}\right) = 0 \quad\Longleftrightarrow\quad A^\star(0) = \sigma_a^{-2}. \tag{107}$$

Plugging $A^\star = \sigma_a^{-2}$ into the quadratic and solving for $\kappa^2$ gives

$$C\,\kappa_c^2 N^{2\gamma}\sigma_a^{-4} + C_k\sigma_a^{-2} - D_k^2\sigma_a^{-2} = 0, \tag{108}$$

which, after rearrangement, yields

$$\kappa_c^2 = \frac{\sqrt{C_k^2 + 4\,C\,N^{2\gamma}D_k^2\sigma_a^{-2}} - C_k}{2\,C\,\sigma_a^{-2}\,N^{2\gamma}}, \tag{109}$$

the stated expression.

For $\kappa^2 > \kappa_c^2$ the quadratic gives $A^\star(0) \leq \sigma_a^{-2}$, hence $1 - \sigma_a^{-2}/A^\star(0) \leq 0$ and the FP map evaluates to 0 at $m_S = 0$, so $m_S = 0$ is a (and in fact the only) solution. For $\kappa^2 < \kappa_c^2$ we have $A^\star(0) > \sigma_a^{-2}$ so the FP map at $m_S = 0$ is strictly positive, continuity and the fact that the map is strictly decreasing in $m_S$ (because $(1 - m_S)$ and $A^\star(m_S)$ both decrease with $m_S$) imply a unique intersection with the diagonal in $(0, 1)$, i.e. a unique $m_S \in (0, 1)$ solves the FP equation. $\qquad\square$

### D.7 How MF-ARD beats the curse of dimensionality

We use the following notation. Let $S \subset [d]$ with $|S| = k$ denote the (unknown) support. For a weight coordinate $w_j$ at a given outer iterate, write

$$v_j := \langle w_j^2 \rangle_p, \qquad v_j'(\rho) := \mathbb{E}_{p_\rho}[w_j^2] \tag{110}$$

for the current and next second moments, respectively. For a vector of ARD precisions $\rho \in \mathbb{R}_+^d$ define the explicit ARD map

$$\rho_j(v_j) = \frac{\alpha_0 + \frac{N}{2}}{\frac{\alpha_0}{d} + \frac{N}{2} v_j} =: \frac{A}{B(v_j)}, \qquad A := \alpha_0 + \frac{N}{2}, \ B(v) := \frac{\alpha_0}{d} + \frac{N}{2} v. \tag{111}$$

We write $w_{-j} := (w_1, \ldots, w_{j-1}, w_{j+1}, \ldots, w_d)$ for all coordinates except $j$ and write $p_\rho$ for the posterior $p_{\mathrm{ARD}}$ to make the $\rho$ dependence explicit.

**Assumption** Here, we will state the assumption that is needed to prove the theorem: $\varepsilon$ **symmetry breaking towards** $S$. There exists an outer iterate $t_0 = \mathcal{O}(1)$ and a constant $\varepsilon_0 > 0$ (independent of $d$) such that

$$\min_{j \in S} v_j^{t_0} - \max_{j \notin S} v_j^{t_0} \geq c\, \varepsilon_0, \qquad v_j^t := \langle w_j^2 \rangle_p \text{ at outer time } t. \tag{112}$$

We need to establish the following global bound.

**Lemma D.9.** *Fix $j \in [d]$ and $w_{-j}$. Assume $\|x\|_\infty \leq 1$ (e.g. $x \in \{\pm 1\}^d$) and $\phi(z) = \max(0, z)$, as well as $m_S \leq 1$ Let*

$$g(\boldsymbol{w}) = \frac{1}{2} \ln\left(\sigma_a^{-2} + \frac{\Sigma(\boldsymbol{w})}{\kappa^2 N^{2\gamma}}\right) - \frac{\left(J_{\mathcal{Y}}(\boldsymbol{w}) - m_S J_S(\boldsymbol{w})\right)^2}{2\kappa^4 N^{2\gamma}\left(\sigma_a^{-2} + \frac{\Sigma(\boldsymbol{w})}{\kappa^2 N^{2\gamma}}\right)}. \tag{113}$$

*Then there exists $L_\star > 0$, independent of $d$ and $w_{-j}$, such that the map $t \mapsto g(w_{-j}, t)$ satisfies*

$$\left|\partial_t^2 g(w_{-j}, t)\right| \leq L_\star \qquad \text{for a.e. } t \in \mathbb{R}. \tag{114}$$

*Consequently, for all $t \in \mathbb{R}$,*

$$g(w_{-j}, t) \geq g(w_{-j}, 0) - \frac{L_\star}{2} t^2. \tag{115}$$

*Proof.* Fix $j \in [d]$ and $w_{-j}$. Write $t := w_j$ and, for each input $x$, set $z(t, x) := \boldsymbol{w}^\top x = t\, x_j + c_x$ with $c_x := w_{-j}^\top x_{-j}$. Throughout, $\phi(z) = \max(0, z)$ and $\|x\|_\infty \leq 1$.

**1. Derivative of $\Sigma$ and $J_A$**

For $\Sigma(\boldsymbol{w}) = \mathbb{E}_x[\phi(z)^2] = \mathbb{E}_x[z(t, x)^2\, \mathbf{1}\{z(t, x) > 0\}]$ we have, for a.e. $t$,

$$\partial_t \Sigma = 2\, \mathbb{E}_x[z \mathbf{1}_{\{z>0\}}\, x_j], \qquad \partial_t^2 \Sigma = 2\, \mathbb{E}_x[x_j^2\, \mathbf{1}_{\{z>0\}}] \leq 2, \tag{116}$$

because $|x_j| \leq 1$. By Cauchy–Schwarz,

$$|\partial_t \Sigma| \leq 2\left(\mathbb{E}_x[z^2 \mathbf{1}_{\{z>0\}}]\right)^{1/2}\left(\mathbb{E}_x[x_j^2 \mathbf{1}_{\{z>0\}}]\right)^{1/2} \leq 2\sqrt{\Sigma}. \tag{117}$$

For $J_A(\boldsymbol{w}) = \mathbb{E}_x[\phi(z)\, \chi_A(x)]$ (with $|\chi_A| \leq 1$), we get for a.e. $t$:

$$\partial_t J_A = \mathbb{E}_x[\mathbf{1}_{\{z>0\}}\, x_j\, \chi_A(x)], \quad \partial_t^2 J_A = 0 \text{ (a.e.)}, \qquad |\partial_t J_A| \leq \mathbb{E}|x_j| \leq 1, \tag{118}$$

and by Cauchy–Schwarz again

$$|J_A| \leq \left(\mathbb{E}_x[\phi(z)^2]\right)^{1/2}\left(\mathbb{E}_x[\chi_A(x)^2]\right)^{1/2} = \sqrt{\Sigma}. \tag{119}$$

**2. Decompose $g$ and bound each second derivative**

Let

$$a := \sigma_a^{-2}, \qquad c := \frac{1}{\kappa^2 N^{2\gamma}}, \qquad J_\Delta := J_{\mathcal{Y}} - m_S J_S, \qquad q := J_\Delta^2, \tag{120}$$

so

$$g(\boldsymbol{w}) = \underbrace{\tfrac{1}{2} \log \left( a + c\Sigma \right)}_{=:g_1(\Sigma)} - \underbrace{\frac{q}{2\kappa^4 N^{2\gamma} \left( a + c\Sigma \right)}}_{=:g_2(\Sigma, J_\Delta)}. \tag{121}$$

*Term 1:* Set $h_1(s) := \tfrac{1}{2} \log(a + cs)$, so $h_1'(s) = \frac{c}{2(a+cs)}$ and $h_1''(s) = -\frac{c^2}{2(a+cs)^2}$. By the chain rule,

$$\partial_t^2 g_1 = h_1''(\Sigma) \left( \partial_t \Sigma \right)^2 + h_1'(\Sigma) \partial_t^2 \Sigma. \tag{122}$$

Using equation 117 and $\partial_t^2 \Sigma \leq 2$,

$$\left| h_1''(\Sigma) \left( \partial_t \Sigma \right)^2 \right| \leq \frac{c^2}{2(a + c\Sigma)^2} \cdot 4\Sigma = 2c^2 \frac{\Sigma}{(a + c\Sigma)^2} \leq \frac{c\,\sigma_a^2}{2}, \tag{123}$$

where the last inequality follows by maximizing $u \mapsto \frac{u}{(a+cu)^2}$ at $u = a/c$. Also $\left| h_1'(\Sigma) \partial_t^2 \Sigma \right| \leq \frac{c}{2(a+c\Sigma)} \cdot 2 \leq c\,\sigma_a^2$. Hence

$$\left| \partial_t^2 g_1 \right| \leq \frac{3}{2}\, c\,\sigma_a^2. \tag{124}$$

*Term 2:* Write $g_2 = -\alpha \frac{q}{a+c\Sigma}$ with $\alpha := \frac{1}{2\kappa^4 N^{2\gamma}}$. Differentiating twice and grouping terms gives (for a.e. $t$):

$$\partial_t^2 g_2 = -\alpha \left[ \frac{q''}{a + c\Sigma} - \frac{2\,q'\,c\,\Sigma'}{(a + c\Sigma)^2} - \frac{q\,c\,\Sigma''}{(a + c\Sigma)^2} + \frac{2\,q\,(c\Sigma')^2}{(a + c\Sigma)^3} \right],$$

where primes denote $\partial_t$. We now bound $q, q', q''$ with step 1 and using equation 119 and $|\partial_t J_A| \leq 1$.

- $q$: Using $|J_\Delta| \leq |J_{\mathcal{Y}}| + |m_S|\,|J_S| \leq (1 + |m_S|)\sqrt{\Sigma} =: C_\Delta \sqrt{\Sigma}$ we get $q = J_\Delta^2 \leq C_\Delta^2 \Sigma$.

- $|q'|$: We get

$$|q'| = 2|J_\Delta||\partial_t J_\Delta| \leq 2C_\Delta \sqrt{\Sigma} \cdot \left( |\partial_t J_{\mathcal{Y}}| + |m_S||\partial_t J_S| \right) \leq 2C_\Delta \sqrt{\Sigma}(1 + |m_S|)$$
$$\leq 2C_\Delta^2 \sqrt{\Sigma}$$

- $|q''|$: We get $q'' = 2(\partial_t J_\Delta)^2 \leq 2C_\Delta^2$

Together with $\Sigma' \leq 2\sqrt{\Sigma}$ and $\Sigma'' \leq 2$, we get

$$\left| \frac{q''}{a+c\Sigma} \right| \leq \frac{2C_\Delta^2}{a}, \tag{125}$$

$$\left| \frac{2\,q'\,c\,\Sigma'}{(a+c\Sigma)^2} \right| \leq \frac{2 \cdot 2C_\Delta^2 \sqrt{\Sigma} \cdot c \cdot 2\sqrt{\Sigma}}{(a + c\Sigma)^2} = 8C_\Delta^2 c \frac{\Sigma}{(a + c\Sigma)^2} \leq \frac{2C_\Delta^2}{a}, \tag{126}$$

$$\left| \frac{q\,c\,\Sigma''}{(a+c\Sigma)^2} \right| \leq \frac{C_\Delta^2 \Sigma \cdot c \cdot 2}{(a + c\Sigma)^2} \leq \frac{C_\Delta^2}{a}, \tag{127}$$

$$\left| \frac{2\,q\,(c\Sigma')^2}{(a+c\Sigma)^3} \right| \leq \frac{2\,C_\Delta^2 \Sigma \cdot c^2 \cdot 4\Sigma}{(a + c\Sigma)^3} = 8C_\Delta^2 c^2 \frac{\Sigma^2}{(a + c\Sigma)^3} \leq \frac{32C_\Delta^2}{27\,a}, \tag{128}$$

where in the last two lines we used that $u \mapsto \frac{u}{(a+cu)^2}$ and $u \mapsto \frac{u^2}{(a+cu)^3}$ are maximized at $u = \frac{a}{c}$ and $u = \frac{2a}{c}$, respectively, with finite maxima depending only on $a, c$. Therefore $\left| \partial_t^2 g_2 \right| \leq \alpha K_2$ for a constant $K_2$ depending only on $(a, c, m_S)$, and independent of $d$, $w_{-j}$ and $t$.

**3: Uniform bound on $\partial_t^2 g$**

Combining equation 124 and the bound for $\partial_t^2 g_2$, there exists

$$L_\star \;:=\; \frac{3}{2}\, c\, \sigma_a^2 \;+\; \alpha\, K_2 \tag{129}$$

such that $|\partial_t^2 g(w_{-j}, t)| \leq L_\star$ for a.e. $t \in \mathbb{R}$. This proves the first claim.

**4: Standard taylor expansion**

For any twice–differentiable $h$ with $|h''| \leq L_\star$ a.e., the 1D Taylor inequality gives

$$h(t) \;\geq\; h(0) \;+\; h'(0)\, t \;-\; \tfrac{L_\star}{2}\, t^2 \qquad (\forall t \in \mathbb{R}).$$

Applying this to $h(t) = g(w_{-j}, t)$ yields

$$g(w_{-j}, t) \;\geq\; g(w_{-j}, 0) \;+\; \partial_t g(w_{-j}, 0)\, t \;-\; \frac{L_\star}{2}\, t^2.$$

In the parity setting and for $j \notin S$ (off–support), symmetry implies $\partial_t g(w_{-j}, 0) = 0$, giving the stated global quadratic lower bound $g(w_{-j}, t) \geq g(w_{-j}, 0) - \frac{L_\star}{2} t^2$. $\qquad\square$

This lemma provides a precise bound on the conditional second moment of off-support $w_j$. The key insight is that when the precision parameter $\rho_j$ is large enough to dominate the coupling term, the second moment behaves essentially like that of a Gaussian with precision $\rho_j$, up to small corrections.

**Lemma D.10.** *Let $j \notin S$ be an off-support coordinate and condition on $w_{-j}$. Given Lemma D.9, there exists $L_\star > 0$ (independent of $d$ and $w_{-j}$) such that the coupling function $t \mapsto g(w_{-j}, t)$ satisfies:*

$$|\partial_t^2 g(w_{-j}, t)| \leq L_\star \quad \text{for a.e. } t \in \mathbb{R}.$$

*Assume furthermore the **off-support symmetry condition**:*

$$\partial_j g(w_{-j}, 0) = 0.$$

*Consider the conditional distribution for $w_j$:*

$$p(w_j \mid w_{-j}, \rho) \propto \exp\big(-U_j(w_j)\big),$$

*where the potential is given by:*

$$U_j(w_j) = \frac{1}{2}\rho_j w_j^2 + g(w_j, w_{-j}).$$

*If $\rho_j > L_\star$, then for every fixed $\theta \in (0, 1]$, there exist constants $C, c > 0$ (depending only on $L_\star$ and $\theta$, but independent of $d$ and $w_{-j}$) such that:*

$$\mathbb{E}\big[w_j^2 \mid w_{-j}\big] \leq \frac{e^{L_\star \theta^2}}{\rho_j - L_\star} + C\, e^{-c\,(\rho_j - L_\star)\,\theta^2}.$$

*In particular, for any $\varepsilon > 0$, one can choose $\theta \in (0, 1]$ small enough so that $e^{L_\star \theta^2} \leq 1 + \varepsilon$, yielding:*

$$\mathbb{E}[w_j^2 \mid w_{-j}] \leq \frac{1 + \varepsilon}{\rho_j - L_\star} + C\, e^{-c\,(\rho_j - L_\star)}.$$

*Hence, if $\rho_j = \Omega(d)$, then:*

$$\mathbb{E}[w_j^2 \mid w_{-j}] \leq \frac{1 + o_d(1)}{\rho_j}$$

*with the $o_d(1)$ term uniform in $w_{-j}$.*

*Proof.* The proof uses a careful splitting argument to handle the near-Gaussian behavior in the core region while controlling exponential tails. The key is to leverage the smoothness bound to create uniform upper and lower envelopes for the potential function.

Fix $j \notin S$ and condition on $w_{-j}$. For notational simplicity, write $w := w_j$, $\rho := \rho_j$, and define:

$$U(w) = \frac{1}{2}\rho w^2 + g(w),$$

where $g(w) := g(w, w_{-j})$ is the coupling term.

**Step 1: Establishing potential envelopes**

By Lemma D.9, we have $|g''(t)| \leq L_\star$ for almost every $t$, and the off-support symmetry gives $g'(0) = 0$.

Using Taylor expansion with integral remainder, for all $w \in \mathbb{R}$:

$$-\frac{L_\star}{2}w^2 \leq g(w) - g(0) \leq \frac{L_\star}{2}w^2. \tag{130}$$

Define the effective precision $m := \rho - L_\star > 0$. From equation 130, we obtain two crucial envelopes:

**Global lower envelope:**

$$U(w) \geq U(0) + \frac{m}{2}w^2 \quad \text{for all } w \in \mathbb{R}. \tag{131}$$

**Local upper envelope:** For any $\theta > 0$ and $|w| \leq \theta$:

$$U(w) \leq U(0) + \frac{\rho}{2}w^2 + \frac{L_\star}{2}w^2 \leq U(0) + \frac{m}{2}w^2 + L_\star\theta^2. \tag{132}$$

The global lower envelope ensures integrability, while the local upper envelope allows us to approximate the distribution by a Gaussian in the core region.

**Step 2: Setting up the moment computation**

Let $\mu$ be the conditional measure with density proportional to $e^{-U(w)}$. Define the normalization and second moment integrals:

$$Z := \int_{\mathbb{R}} e^{-U(w)}\, dw, \qquad N := \int_{\mathbb{R}} w^2 e^{-U(w)}\, dw,$$

so that $\mathbb{E}_\mu[w^2] = N/Z$.

We partition $\mathbb{R}$ into the **inside region** $I := \{|w| \leq \theta\}$ and the **outside region** $O := \{|w| > \theta\}$, writing $Z = Z_I + Z_O$ and $N = N_I + N_O$.

**Step 3: Bounding the inside region contribution**

Using the local upper envelope equation 132:

$$Z_I = \int_{|w| \leq \theta} e^{-U(w)}\, dw \geq e^{-U(0)}e^{-L_\star\theta^2} \int_{|w| \leq \theta} e^{-\frac{m}{2}w^2}\, dw.$$

Using the global lower envelope equation 131:

$$N_I = \int_{|w| \leq \theta} w^2 e^{-U(w)}\, dw \leq e^{-U(0)} \int_{|w| \leq \theta} w^2 e^{-\frac{m}{2}w^2}\, dw.$$

Combining these bounds:

$$\frac{N_I}{Z} \leq \frac{N_I}{Z_I} \leq e^{L_\star\theta^2} \cdot \frac{\int_{|w| \leq \theta} w^2 e^{-\frac{m}{2}w^2}\, dw}{\int_{|w| \leq \theta} e^{-\frac{m}{2}w^2}\, dw} \leq \frac{e^{L_\star\theta^2}}{m}. \tag{133}$$

The final inequality follows because the ratio represents the truncated second moment of a zero-mean Gaussian with precision $m$, which is bounded by the untruncated value $1/m$.

**Step 4: Controlling the outside region contribution**

For the tail contribution, we use the global lower envelope equation 131:

$$N_O = \int_{|w|>\theta} w^2 e^{-U(w)} \, dw \le e^{-U(0)} \int_{|w|>\theta} w^2 e^{-\frac{m}{2}w^2} \, dw.$$

A standard Gaussian tail bound (obtained by integration by parts) gives, for $a > 0$ and $\theta > 0$:

$$\int_\theta^\infty t^2 e^{-at^2} \, dt \le \left( \frac{\theta}{2a} + \frac{1}{4a^2\theta} \right) e^{-a\theta^2}.$$

Applying this with $a = m/2$ and doubling for both tails:

$$\int_{|w|>\theta} w^2 e^{-\frac{m}{2}w^2} \, dw \le \frac{2}{m} \left( \theta + \frac{1}{m\theta} \right) e^{-\frac{m}{2}\theta^2}. \tag{134}$$

Using the lower bound on $Z_I$ from Step 3 and equation 134:

$$\frac{N_O}{Z} \le \frac{N_O}{Z_I} \le \frac{e^{L_\star \theta^2}}{c_0} \cdot \frac{2}{m} \left( \theta + \frac{1}{m\theta} \right) e^{-\frac{m}{2}\theta^2} \sqrt{m} \le C_1 e^{-c_1 m\theta^2},$$

where $c_0, C_1, c_1 > 0$ are absolute constants depending only on $L_\star$ and $\theta$. The polynomial factors in $m$ are absorbed into the constant since the exponential term dominates for $m > 0$.

**Step 5: Combining the contributions**

From equation 133 and the outside region bound:

$$\mathbb{E}_\mu[w^2] = \frac{N}{Z} = \frac{N_I}{Z} + \frac{N_O}{Z} \le \frac{e^{L_\star \theta^2}}{m} + C e^{-cm\theta^2},$$

which establishes the main inequality with $m = \rho_j - L_\star$.

For the refined bound, choosing $\theta > 0$ small enough so that $e^{L_\star \theta^2} \le 1 + \varepsilon$ gives the result.

Finally, if $\rho_j = \Omega(d)$, then $m = \rho_j - L_\star = \Omega(d)$, so the exponential tail term is $e^{-\Omega(d)}$ uniformly in $w_{-j}$, and:

$$\frac{1+\varepsilon}{\rho_j - L_\star} = \frac{1+\varepsilon}{\rho_j \left( 1 - \frac{L_\star}{\rho_j} \right)} = \frac{1 + o_d(1)}{\rho_j}.$$

$\square$

The next lemma establishes that off-support $w_j$ contract towards equilibrium values of order $O(1/d)$ when initialized in a suitable bootstrap region. The key insight is that by controlling the initialization within $O(1/d)$, we can ensure the dynamics remain stable and converge.

**Lemma D.11.** *Let $r > 0$ be a radius parameter and let $L_\star$ be the global smoothness constant from Lemma D.9. We define the following key quantities:*

$$A := \alpha_0 + \frac{N}{2}, \qquad \text{(total prior mass)} \tag{135}$$

$$B(v) := \frac{\alpha_0}{d} + \frac{N}{2} \cdot v, \qquad \text{(effective local prior)} \tag{136}$$

*For any threshold $K_\star > 0$ and dimension $d \in \mathbb{N}$, we introduce the bootstrap parameters:*

$$U_d(K_\star) := \frac{\alpha_0 + \frac{N}{2}K_\star}{d}, \qquad \text{(maximum local prior)} \tag{137}$$

$$a_d(r, K_\star) := \frac{e^{L_\star r^2} \cdot \alpha_0}{A - L_\star \cdot U_d(K_\star)}, \qquad \text{(affine drift term)} \tag{138}$$

$$b_d(r, K_\star) := \frac{e^{L_\star r^2} \cdot (N/2)}{A - L_\star \cdot U_d(K_\star)}. \qquad \text{(contraction coefficient)} \tag{139}$$

*Note that as $d \to \infty$, we have the asymptotic behavior:*

$$b_d(r, K_\star) \to b_\infty(r) := \frac{e^{L_\star r^2} \cdot (N/2)}{A}.$$

*Choose $r > 0$ sufficiently small so that $b_\infty(r) < 1$. This ensures asymptotic contraction. Then there exist $d_0 \in \mathbb{N}$, a finite threshold $K_\star > 0$ (both independent of $d$), and positive constants $C, c, c' > 0$ such that for all $d \geq d_0$:*

1. **One-step contraction bound:** *If $v_{\text{off}}^t \leq K_\star/d$, then with $\rho_{\text{off}}^t = A/B(v_{\text{off}}^t)$,*

$$v_{\text{off}}^{t+1} \leq \frac{e^{L_\star r^2} \cdot B(v_{\text{off}}^t)}{A - L_\star B(v_{\text{off}}^t)} + C\, e^{-c\,(\rho_{\text{off}}^t - L_\star)\, r^2} \tag{140}$$

$$\leq \frac{a_d(r, K_\star)}{d} + b_d(r, K_\star) \cdot v_{\text{off}}^t + \eta_d, \tag{141}$$

   *where the exponential tail satisfies $\eta_d \leq Ce^{-c'd}$.*

2. **Bootstrap invariance:** *There exists a finite $K_\star$ such that*

$$v_{\text{off}}^t \leq \frac{K_\star}{d} \quad \implies \quad v_{\text{off}}^{t+1} \leq \frac{K_\star}{d} \quad \text{for all } t \geq 0.$$

   *In particular, any initialization with $v_{\text{off}}^0 \leq K_\star/d$ remains within the bootstrap region for all time.*

3. **Equilibrium bound:** *Along any trajectory that remains in the bootstrap region,*

$$v_{\text{off}}^\star \leq \frac{a_d(r, K_\star)}{(1 - b_d(r, K_\star)) \cdot d} + O(e^{-c'd}) = \Theta\left(\frac{1}{d}\right).$$

*Proof.* The proof proceeds in four main steps: establishing a one-step bound, deriving an affine upper bound, proving bootstrap invariance, and analyzing convergence.

Fix $r > 0$ and let $L_\star$ be the smoothness constant from Lemma D.9. Consider an off-support coordinate at outer iteration $t$. We write $v := v_{\text{off}}^t$, $B(v) = \frac{\alpha_0}{d} + \frac{N}{2}v$, and $\rho := \rho_{\text{off}}^t = \frac{A}{B(v)}$ with $A = \alpha_0 + \frac{N}{2}$ (see equation 111).

Throughout, we assume $v \leq K_\star/d$ for some threshold $K_\star > 0$ to be determined.

**Step 1: Establishing the one-step bound**

We begin by applying the second moment bound from Lemma D.10. For any $r > 0$ and whenever $\rho > L_\star$, this yields:

$$v_{\text{off}}^{t+1} = \mathbb{E}[w_j^2] \leq \frac{e^{L_\star r^2}}{\rho - L_\star} + C\, e^{-c\,(\rho - L_\star)\, r^2}. \tag{142}$$

Substituting $\rho = \frac{A}{B(v)}$, the main term becomes:

$$\frac{e^{L_\star r^2}}{\rho - L_\star} = e^{L_\star r^2} \cdot \frac{B(v)}{A - L_\star B(v)}.$$

Since we assume $v \leq K_\star/d$, we have the upper bound:

$$B(v) \leq U_d(K_\star) := \frac{\alpha_0 + \frac{N}{2}K_\star}{d} =: \frac{C_B}{d},$$

which implies $\rho = \frac{A}{B(v)} \geq \frac{A \cdot d}{C_B}$. This lower bound on $\rho$ allows us to control the exponential tail:

$$C\, e^{-c\,(\rho - L_\star)\, r^2} \leq C\, e^{-c'd} =: \eta_d \tag{143}$$

for some $c' = c'(A, C_B, L_\star, r) > 0$ that depends only on the fixed parameters.

**Step 2: Deriving the affine upper bound**

To obtain a tractable recursion, we upper bound the main term using convexity. Define the function:

$$f(u) := \frac{u}{A - L_\star u},$$

which is convex on the interval $[0, A/L_\star)$.

For all $u \in [0, U_d(K_\star)]$, monotonicity of the denominator gives:

$$f(u) = \frac{u}{A - L_\star u} \leq \frac{u}{A - L_\star U_d(K_\star)}.$$

Applying this with $u = B(v)$, we obtain the key affine upper bound:

$$e^{L_\star r^2} \cdot \frac{B(v)}{A - L_\star B(v)} \leq \frac{e^{L_\star r^2}}{A - L_\star U_d(K_\star)} \left( \frac{\alpha_0}{d} + \frac{N}{2} v \right) =: \frac{a_d(r, K_\star)}{d} + b_d(r, K_\star) \cdot v, \quad (144)$$

where we have defined:

$$a_d(r, K_\star) := \frac{e^{L_\star r^2} \cdot \alpha_0}{A - \frac{L_\star C_B}{d}}, \quad (145)$$

$$b_d(r, K_\star) := \frac{e^{L_\star r^2} \cdot (N/2)}{A - \frac{L_\star C_B}{d}}. \quad (146)$$

Note that $a_d(r, K_\star) = \Theta(1)$ and $b_d(r, K_\star) \to b_\infty(r) = \frac{e^{L_\star r^2} \cdot (N/2)}{A}$ as $d \to \infty$.

**Step 3: Establishing bootstrap invariance**

The key is to choose parameters ensuring contraction. First, pick $r > 0$ small enough so that:

$$b_\infty(r) = \frac{e^{L_\star r^2} \cdot (N/2)}{A} < 1.$$

This is possible because $\frac{N/2}{A} < 1$ (since $\alpha_0 > 0$) and $e^{L_\star r^2} \to 1$ as $r \to 0$.

With $r$ fixed, there exists $d_0$ such that for all $d \geq d_0$:

$$b_d(r, K_\star) \leq \frac{b_\infty(r) + 1}{2} < 1.$$

The bound is uniform in $K_\star$ because $U_d(K_\star) = O(1/d)$.

Combining equations equation 142, equation 144, and equation 143, for $d \geq d_0$ and $v \leq K_\star/d$:

$$v_{\text{off}}^{t+1} \leq \frac{a_d(r, K_\star)}{d} + b_d(r, K_\star) \cdot v + \eta_d.$$

Now choose $K_\star$ large enough so that:

$$K_\star \geq \sup_{d \geq d_0} \frac{a_d(r, K_\star)}{1 - b_d(r, K_\star)} < \infty.$$

The supremum is finite because $a_d(r, K_\star) = \Theta(1)$ and $1 - b_d(r, K_\star)$ is bounded away from zero for $d \geq d_0$.

Finally, enlarge $d_0$ if necessary so that $\eta_d \leq \frac{1}{2}(1 - b_d(r, K_\star))\frac{K_\star}{d}$ for all $d \geq d_0$. This ensures bootstrap invariance:

$$v_{\text{off}}^t \leq \frac{K_\star}{d} \quad \Longrightarrow \quad v_{\text{off}}^{t+1} \leq \frac{K_\star}{d}.$$

**Step 4: Convergence analysis**

Within the invariant bootstrap region, the affine recursion:

$$v_{\text{off}}^{t+1} \leq \frac{a_d(r, K_\star)}{d} + b_d(r, K_\star) \cdot v_{\text{off}}^t + \eta_d$$

contracts towards its fixed point. Since $b_d(r, K_\star) < 1$, the equilibrium value satisfies:

$$v_{\text{off}}^\star \leq \frac{a_d(r, K_\star)}{(1 - b_d(r, K_\star)) \cdot d} + O(e^{-c'd}) = \Theta\left(\frac{1}{d}\right).$$

$\square$

**Lemma D.12.** *Assume Theorem D.11 3. For all sufficiently large $d$,*

$$\min_{j \in S} v_j^{t_0} \geq c\varepsilon_0 + \max_{j \notin S} v_j^{t_0} \geq c\varepsilon_0 - O(d^{-1}) \geq \tfrac{c}{2}\varepsilon_0 = \Theta(1).$$

*Consequently, at (and after) time $t_0$, the ARD precisions satisfy $\rho_{\text{on}} = \Theta(1)$ while $\rho_{\text{off}} = \Theta(d)$.*

*Proof.* The first display follows directly from the assumption D.7 and $v_{\text{off}}^{t_0} = O(d^{-1})$ (bootstrap entry). For $j \in S$, $B(v_j) = \frac{\alpha_0}{d} + \frac{N}{2}v_j \geq \frac{N}{2} \cdot \frac{c}{2}\varepsilon_0$ for large $d$, so $\rho_j = A/B(v_j) = \Theta(1)$ uniformly in $d$. For $j \notin S$, by Theorem D.11 3), $v_j = \Theta(d^{-1})$, hence $B(v_j) = \Theta(d^{-1})$ and $\rho_j = \Theta(d)$. $\square$

**Lemma D.13.** *For a neuron with equal weights on $S$:*

$$C_{\text{MF}} = \frac{dk}{\sigma_w^2} \qquad \text{and} \qquad C_{\text{ARD}}(t) = \sum_{j \in S} \rho_j^t = \Theta(k) \text{ for all } t \geq t_0 \text{ under Theorem D.12.}$$

*Proof. MF:* With prior penalty $\frac{d}{2\sigma_w^2}\|\boldsymbol{w}\|^2$, along $\boldsymbol{w} = \alpha\,\mathbf{1}_S$ we get $\frac{dk}{2\sigma_w^2}\alpha^2$, hence $C_{\text{MF}} = \frac{dk}{\sigma_w^2}$. *ARD:* By Theorem D.12, $\rho_j^t = \Theta(1)$ for $j \in S$ and $t \geq t_0$, so $C_{\text{ARD}}(t) = \Theta(k)$. $\square$

**Lemma D.14.** *At the symmetric initialization point $m_S = 0$, the FL threshold satisfies:*

$$\kappa_c^2 = \frac{\sqrt{C_k^2 + 4CN^{2\gamma}D_k^2\sigma_a^{-2}} - C_k}{2C\sigma_a^{-2}N^{2\gamma}}, \tag{147}$$

*where:*

- *$C$ is the quadratic curvature along the equal-weights $S$-direction,*

- *$C_k, D_k$ are geometric constants depending only on the support size $k$,*

- *$N$ is the number of training samples, $\gamma$ is a scaling exponent,*

- *$\sigma_a$ controls the output noise level.*

*For large curvature $C$ (corresponding to strong regularization), the threshold simplifies to:*

$$\kappa_c^2 \sim \frac{|D_k|\sigma_a}{N^\gamma\sqrt{C}} =: \frac{\Lambda_k}{\sqrt{C}}, \tag{148}$$

*where $\Lambda_k := |D_k|\sigma_a/N^\gamma$.*

*Proof.* By Lemma D.5, the optimal weight configuration is $\mathbf{w}^* = \alpha\mathbf{1}_S$. By Lemma D.6, this gives scale-independent constants $C_k = \mathbb{E}[Z_+^2]$ and $D_k = \mathbb{E}[Z_+\chi_S(x)]$ such that:

$$\Sigma(\alpha) = C_k\alpha^2, \qquad J_S(\alpha) = D_k\alpha, \qquad R_k = \frac{D_k^2}{C_k}.$$

From Theorem D.8, the critical threshold occurs when $A^*(0) = \sigma_a^{-2}$, where $A^*$ solves the quadratic equation at $m_S = 0$.

**Step 2: Taylor expansion analysis**

We analyze the one-dimensional effective potential along $\mathbf{w} = \alpha\mathbf{1}_S$:

$$U(\alpha) = \frac{C}{2}\alpha^2 + g(\alpha),$$

where $C$ is the prior curvature (Lemma D.13) and $g(\alpha)$ is the coupling term from Lemma D.9.

At onset ($m_S = 0$), we have $J_\Delta = J_S = D_k\alpha$. The coupling term becomes:

$$g(\alpha) = \frac{1}{2}\log\left(a + c\Sigma(\alpha)\right) - \frac{J_S(\alpha)^2}{2\kappa^4 N^{2\gamma}(a + c\Sigma(\alpha))},$$

with $a = \sigma_a^{-2}$, $c = \frac{1}{\kappa^2 N^{2\gamma}}$, $\Sigma(\alpha) = C_k\alpha^2$, and $J_S(\alpha) = D_k\alpha$.

**Step 3: Computing the Taylor expansion**

Expanding around $\alpha = 0$:

For the logarithmic term:

$$\frac{1}{2}\log\left(a + cC_k\alpha^2\right) = \frac{1}{2}\log a + \frac{cC_k}{2a}\alpha^2 + O(\alpha^4).$$

For the rational term:

$$\frac{D_k^2\alpha^2}{2\kappa^4 N^{2\gamma}(a + cC_k\alpha^2)} = \frac{D_k^2\alpha^2}{2\kappa^4 N^{2\gamma}a} + O(\alpha^4) = \frac{\sigma_a^2 D_k^2}{2\kappa^4 N^{2\gamma}}\alpha^2 + O(\alpha^4).$$

Therefore:

$$U(\alpha) = \text{const} + \frac{1}{2}\left[C + \frac{\sigma_a^2 C_k}{\kappa^2 N^{2\gamma}} - \frac{\sigma_a^2 D_k^2}{\kappa^4 N^{2\gamma}}\right]\alpha^2 + O(\alpha^4).$$

**Step 4: Onset condition and threshold**

FL onset occurs when the quadratic coefficient vanishes:

$$C + \frac{\sigma_a^2 C_k}{\kappa^2 N^{2\gamma}} - \frac{\sigma_a^2 D_k^2}{\kappa^4 N^{2\gamma}} = 0.$$

Multiplying by $\kappa^4 N^{2\gamma}/\sigma_a^2$ and rearranging:

$$CN^{2\gamma}\sigma_a^{-2}\kappa^4 + C_k\kappa^2 - D_k^2 = 0.$$

Solving this quadratic in $\kappa^2$ gives:

$$\kappa_c^2 = \frac{\sqrt{C_k^2 + 4CN^{2\gamma}D_k^2\sigma_a^{-2}} - C_k}{2C\sigma_a^{-2}N^{2\gamma}}.$$

**Step 5: Asymptotic approximation**

For large $C$, the square root is dominated by the term $4CN^{2\gamma}D_k^2\sigma_a^{-2}$, so:

$$\sqrt{C_k^2 + 4CN^{2\gamma}D_k^2\sigma_a^{-2}} \approx 2|D_k|\sigma_a^{-1}\sqrt{CN^{2\gamma}}.$$

This yields:

$$\kappa_c^2 \sim \frac{2|D_k|\sigma_a^{-1}\sqrt{CN^{2\gamma}}}{2C\sigma_a^{-2}N^{2\gamma}} = \frac{|D_k|\sigma_a}{N^\gamma\sqrt{C}} = \frac{\Lambda_k}{\sqrt{C}}. \tag{149}$$

$\square$

Now the proof of Theorem 4.1 follows by the lemmata above.

*Proof. MF:* Combine $C_{\text{MF}} = \frac{dk}{\sigma_w^2}$ (Theorem D.13) with equation 149 to get $\kappa_{c,\text{MF}}^2 \asymp \Lambda_k\sqrt{\sigma_w^2/(dk)} = \Theta(\sqrt{1/(dk)})$. *MF–ARD:* By Theorem D.12, for $t \geq t_0$ we have $C_{\text{ARD}}(t) = \Theta(k)$, hence $\kappa_{c,\text{ARD}}^2 \asymp \Lambda_k/\sqrt{k} = \Theta(\sqrt{1/k})$ from equation 149. $\square$

# E    ALGORITHMS

## E.1    FP ALGORITHM

**Model.**    Given $(X, y)$ with $X \in \{\pm 1\}^{P \times d}$ and $y \in \mathbb{R}^{P \times 1}$, we approximate the predictor by a particle ensemble,

$$f(\boldsymbol{x}) = s_f \sum_{b=1}^{B} a_b \, \phi(\boldsymbol{w}_b^\top \boldsymbol{x}), \qquad s_f = \frac{N^{1-\gamma}}{B}. \tag{150}$$

We *draw a single dataset of size $P$ once at initialization and keep it fixed for the entire run*. On the training set let $f_p = f(\boldsymbol{x}_p)$ and $r_p = y_p - f_p$. The Langevin temperature is fixed by the likelihood noise as $T = 2 \kappa^2$ (Section A.1). This choice makes SGLD asymptotically sample from the Bayesian posterior. Because all objectives and gradients are computed as empirical averages over this fixed sample (via $1/P$ factors), the dynamics naturally exhibit finite-$P$ fluctuations.

**Sufficient statistics**    We define the following low-rank statistics per particle $b$, with $z_{pb} = \boldsymbol{x}_p^\top \boldsymbol{w}_b$:

$$C_{1,b} = \sum_{p=1}^{P} \phi(z_{pb}) \, r_p, \qquad C_{2,b} = \sum_{p=1}^{P} \phi(z_{pb})^2, \tag{151}$$

$$G_b^{\text{data}} = -\frac{2}{P} \sum_{p=1}^{P} \Big( r_p - s_f a_b \, \phi(z_{pb}) \Big) \, \phi'(z_{pb}) \, (s_f a_b) \, x_p \; \in \mathbb{R}^d. \tag{152}$$

These quantities are the only per-pass summaries we need to form gradients for $a_b$ and $w_b$.

**Prior and SGLD potential**    We impose a diagonal (ARD) Gaussian prior on the weights and an i.i.d. Gaussian prior on amplitudes:

$$E_{\text{prior}} = \tfrac{1}{2} \sum_{b=1}^{B} \rho^\top (\boldsymbol{w}_b \odot \boldsymbol{w}_b) + \tfrac{1}{2\sigma_a^2} \sum_{b=1}^{B} a_b^2, \quad \mathcal{L}_{\text{data}} = \frac{1}{P} \sum_{p=1}^{P} (y_p - f_p)^2, \tag{153}$$

and update parameters by Langevin dynamics on the potential

$$U(W, a) = T E_{\text{prior}} + \mathcal{L}_{\text{data}}. \tag{154}$$

**Gradients used by SGLD**    From the streamed statistics we get closed-form gradients:

$$\nabla_{a_b} U = \frac{T}{\sigma_a^2} a_b - \frac{2 s_f}{P} C_{1,b} + \frac{2 s_f^2}{P} C_{2,b} a_b, \qquad \nabla_{w_b} U = G_b^{\text{data}} + T (\rho \odot \boldsymbol{w}_b). \tag{155}$$

These are the only quantities used inside the inner SGLD loop.

**SGLD updates**    We apply Euler–Maruyama steps with isotropic Gaussian noise:

$$w_b \leftarrow w_b - \eta \, \nabla_{w_b} U + \sqrt{2T\eta} \, \xi_{w_b}, \qquad a_b \leftarrow a_b - \eta \, \nabla_{a_b} U + \sqrt{2T\eta} \, \xi_{a_b}, \tag{156}$$

where $\xi_{w_b} \sim \mathcal{N}(0, I_d)$ and $\xi_{a_b} \sim \mathcal{N}(0, 1)$. We use polynomial decay on $\eta$ always matching the SGLD-trained full NNs.

**ARD update**    The ARD update is:

$$\alpha_{\text{post}} = \alpha_0 + \frac{B}{2}, \qquad \beta_{\text{post}} = \beta_0 + \tfrac{1}{2} \sum_{b=1}^{B} \|w_b\|_2^2, \qquad \rho \leftarrow (1 - \lambda) \rho + \lambda \frac{\alpha_{\text{post}}}{\beta_{\text{post}}}. \tag{157}$$

**Fixed-point view and the $K$ inner steps**    The outer loop implements a fixed-point iteration on the network $f$. Writing $r = y - f$, define the map $\mathcal{G}_K$ as: (i) run $K$ inner SGLD steps on $(\boldsymbol{w}_b, a)$ using the current residual $r$; (ii) recompute $f^{\text{new}}(x_p) = s_f \sum_b a_b \phi(w_b^\top x_p)$. As $K \to \infty$ and $\eta \to 0$ the inner Markov chain approaches its stationary law, and the iteration solves the MF FP equations described in the theory sections.

---

**Algorithm 1** Simple streaming SGLD for $(a, w)$ with optional ARD

---

**Require:** $(X, y)$; $B, N, \gamma, \sigma_a, \sigma_w, \kappa$; activation $\phi$; steps $T_{\text{out}}$, inner steps $K$, step size $\eta$; optional
    ARD $(\alpha_0, \beta_0, \lambda)$
**Ensure:** Final particles $\{(w_b, a_b)\}_{b=1}^{B}$ and predictor $f(x) = s_f \sum_b a_b \phi(w_b^\top x)$
 1: **Init:** $w_b \sim \mathcal{N}(0, \sigma_w^2 I_d/d)$, $a_b \sim \mathcal{N}(0, \sigma_a^2)$; set $\rho \leftarrow d/\sigma_w^2$; set $T \leftarrow 2\kappa^2$
 2: **for** $t = 1..T_{\text{out}}$ **do**
 3:     compute $f_p = s_f \sum_b a_b \phi(x_p^\top w_b)$ and residuals $r_p = y_p - f_p$
 4:     **for** $k = 1..K$ **do**
 5:         compute $\{C_{1,b}, C_{2,b}, G_b^{\text{data}}\}$ via formulas above
 6:         Form $\nabla_{a_b} U, \nabla_{w_b} U$; update $w_b \leftarrow w_b - \eta \nabla_{w_b} U + \sqrt{2T\eta}\, \xi_{w_b}$, $a_b \leftarrow a_b - \eta \nabla_{a_b} U +$
    $\sqrt{2T\eta}\, \xi_{a_b}$
 7:         refresh $f_p, r_p$ after the last inner step
 8:     **end for**
 9:     ARD update $\rho$: $\alpha_{\text{post}} = \alpha_0 + \frac{B}{2}$, $\beta_{\text{post}} = \beta_0 + \frac{1}{2} \sum_b \|w_b\|^2$, $\rho \leftarrow (1 - \lambda)\rho + \lambda\, \alpha_{\text{post}}/\beta_{\text{post}}$
10: **end for**

---

**Empirical calculation of $m_S$ and generalization error** Let the teacher be a single Walsh mode $\chi_S$, so $y(\boldsymbol{x}) = \chi_S(\boldsymbol{x})$. On a held-out set $\{x_\mu\}_{\mu=1}^{P_{\text{eval}}}$ define the vector $c \in \mathbb{R}^{P_{\text{eval}}}$ by $c_\mu = \chi_S(\boldsymbol{x}_\mu)$, the empirical Gram scalar

$$ g = \frac{1}{P_{\text{eval}}} c^\top c, \tag{158} $$

and the (scalar) empirical overlap

$$ m_S = \frac{1}{P_{\text{eval}}} \sum_{\mu=1}^{P_{\text{eval}}} \chi_S(\boldsymbol{x}_\mu) f(x_\mu) = \frac{1}{P_{\text{eval}}} c^\top f. \tag{159} $$

Let $\bar{f}^2 = \frac{1}{P_{\text{eval}}} \sum_\mu f(\boldsymbol{x}_\mu)^2$. Then the empirical test MSE decomposes as

$$ \widehat{\mathcal{E}}_{\text{test}} = \frac{1}{2P_{\text{eval}}} \sum_{\mu=1}^{P_{\text{eval}}} \big(f(\boldsymbol{x}_\mu) - \chi_S(\boldsymbol{x}_\mu)\big)^2 = \underbrace{\tfrac{1}{2}(1 - m_S)^2}_{\text{mode term}} + \underbrace{\tfrac{1}{2}\big(\bar{f}^2 - 2\, m_S^2 + g\, m_S^2\big)}_{\text{noise / orthogonal term}} \tag{160} $$

$$ = \frac{1}{2}\big(\bar{f}^2 - 2\, m_S + g\big), \tag{161} $$

where the second line is the direct empirical expression. When the Walsh basis is orthonormal on the evaluation set ($g = 1$), this reduces to $\widehat{\mathcal{E}}_{\text{test}} = \frac{1}{2}(\bar{f}^2 - 2m_S + 1)$.

# F TRAINING DETAILS (HYPERPARAMETERS)

Here, we present the hyperparameters for SGLD and MF-ARD for the different figures. The hyperparameters for Figure 5 are specified below.

- Data in Figure 1 is a slice of Figure 5 for $\kappa = 5 \cdot 10^{-3}$.

- Data in Figure 3 is a slice of Figure 5 for $\kappa = 5 \cdot 10^{-3}$.

- Figure 4 are averages over 3 trained models from the data of Figure 5 for $\kappa = 5 \cdot 10^{-3}$.

| Hyperparameter | SGLD |
|---|---|
| $d$ (input dimension) | 35 |
| $P$ (train set sizes) | $\{10, 100, 500, 750, 1000, 2133, 3666, 5000, 7500, 10000\}$ |
| $\kappa$ values | $\{5{\times}10^{-4}, 10^{-3}, 5{\times}10^{-3}, 7.5{\times}10^{-3}, 10^{-2}, 5{\times}10^{-2}, 10^{-1}\}$ |
| $E$ (experiments / config) | 3 |
| teacher set S | $\{0, 1, 2, 3\}$ |
| data distribution | $X \in \{-1, +1\}^d,\ y = \prod_{j \in S} X_j$ |
| activation | ReLU |
| $N$ (hidden units) | 512 |
| $\gamma$ (scaling exponent) | 0.5 |
| $g_w,\ g_a$ (prior variances) | 1.0, 1.0 |
| initialization | $w \sim \mathcal{N}(0, g_w/d),\ a \sim \mathcal{N}(0, g_a)$ |
| temperature $T$ | $2\,\kappa^2$ |
| epochs (max) | 7,500,000 |
| batch size | full-batch |
| loss | mean MSE |
| learning rate $\eta$ (final) | $5{\times}10^{-4}$ |
| start LR $\eta_{\text{start}}$ | $5{\times}10^{-3}$ |
| LR scheduler | polynomial (power 2): $\eta_{\text{start}} \to \eta$ over $2{\times}10^6$ steps |

Table 1: Algorithm outlined in Section A.1. Hyperparamters for Figure 5 **a)**.

| Hyperparameter | MF-ARD |
|---|---|
| $d$ (input dimension) | 35 |
| $P$ (train set sizes) | $\{10, 100, 500, 750, 1000, 2133, 3666, 5000, 7500, 10000\}$ |
| $\kappa$ values | $\{5{\times}10^{-4}, 10^{-3}, 5{\times}10^{-3}, 7.5{\times}10^{-3}, 10^{-2}, 5{\times}10^{-2}, 10^{-1}\}$ |
| $E$ (experiments / config) | 3 |
| teacher set S | $\{0, 1, 2, 3\}$ |
| data distribution | $X \in \{-1, +1\}^d,\ y = \prod_{j \in S} X_j$ |
| activation | ReLU |
| $B$ (particles) | 512 |
| $N$ | 512 |
| $\gamma$ | 0.5 |
| $\sigma_a,\ \sigma_w$ | 1.0, 1.0 |
| initialization | $w \sim \mathcal{N}(0, \sigma_w^2/d),\ a \sim \mathcal{N}(0, \sigma_a^2)$ |
| outer steps | 7,500,000 |
| learning rate scheduler | poly-2 decay: $5 \times 10^{-3} \to 5{\times}10^{-4}$ over $2{\times}10^6$ steps |
| SGLD inner steps $K$ | $K_0{=}12\ \to\ K_{\min}{=}2$ (decay over $6{\times}10^5$ steps) |
| temperature $T$ | $2\kappa^2$ |
| ARD | on: $\alpha_0{=}4.0,\ \beta_0{=}4/35$, EMA 0.5, $\rho \in [0, 10^{18}]$ |

Table 2: Algorithm outlined in Section E.1. Hyperparamters for Figure 5 **b)** with ARD disabled and **c)** with ARD enabled.

For the single index model we introduced a bias vector in the architecture.

| Hyperparameter | SGLD (Hermite single-index) |
|---|---|
| $d$ (input dimension) | 18 |
| $P$ (train set sizes) | $\{75{,}000, 50{,}000, 25{,}000, 10{,}000, 5{,}000, 1{,}000, 100, 50\}$ |
| $\kappa$ values | $\{10^{-5}, 10^{-4}, 10^{-3} 10^{-2}, 10^{-1}\}$ |
| $E$ (experiments / config) | 4 |
| teacher type | single-index Hermite |
| Hermite degree $p$ | 4 |
| support size $k$ | 2 (random per experiment) |
| data distribution | $X \sim \mathcal{N}(0, I_d)$ |
| teacher $w$ | $w_i = \frac{1}{\sqrt{k}}$ on support, else 0 |
| labels | $y = \mathrm{He}_p(Xw)$ |
| activation | ReLU |
| $N$ | 1024 |
| $\gamma$ | 0.5 |
| $\sigma_a,\ \sigma_w,\ \sigma_b$ | 1.0, 0.5, 1.0 |
| initialization | $w \sim \mathcal{N}(0, \sigma_w^2/d),\ a \sim \mathcal{N}(0, \sigma_a^2),\ b \sim \mathcal{N}(0, \sigma_b^2)$ |
| temperature $T$ | $2\,\kappa^2$ |
| epochs (max) | 4,000,000 |
| batch size | full-batch |
| loss | mean MSE |
| learning rate $\eta$ (final) | $5 \times 10^{-4}$ |
| start LR $\eta_{\text{start}}$ | $2 \times 10^{-3}$ |
| LR scheduler | polynomial (power 2): $\eta_{\text{start}} \to \eta$ over $2 \times 10^6$ steps |

Table 3: Algorithm outlined in Section A.1. Hyperparamters for Figure 7 **a)**.

| Hyperparameter | MF-ARD (Hermite single-index) |
|---|---|
| $d$ (input dimension) | 18 |
| $P$ (train set sizes) | $\{75{,}000, 50{,}000, 25{,}000, 10{,}000, 5{,}000, 1{,}000, 100, 50\}$ |
| $\kappa$ values | $\{10^{-5}, 10^{-4}, 10^{-3} 10^{-2}, 10^{-1}\}$ |
| $E$ (experiments / config) | 4 |
| teacher type | single-index Hermite |
| Hermite degree $p$ | 4 |
| support size $k$ | 2 (random per experiment) |
| data distribution | $X \sim \mathcal{N}(0, I_d)$ |
| teacher $w$ | $w_i = \frac{1}{\sqrt{k}}$ on support, else 0 |
| labels | $y = \mathrm{He}_p(Xw)$ |
| activation | ReLU |
| $B$ (particles) | 1024 |
| $N$ | 1024 |
| $\gamma$ | 0.5 |
| $\sigma_a,\ \sigma_w,\ \sigma_b$ | 1.0, 0.5, 1.0 |
| initialization | $w \sim \mathcal{N}(0, \sigma_w^2/d),\ a \sim \mathcal{N}(0, \sigma_a^2),\ b \sim \mathcal{N}(0, \sigma_b^2)$ |
| outer steps | 4,000,000 |
| learning rate scheduler | poly-2 decay: $2 \times 10^{-3} \to 5 \times 10^{-4}$ over $2.5 \times 10^6$ steps |
| SGLD inner steps $K$ | $K_0 = 12\ \to\ K_{\min} = 2$ (decay over $6 \times 10^5$ steps) |
| temperature $T$ | $2\kappa^2$ |
| ARD | on: $\alpha_0 = 0.1,\ \beta_0 = \alpha_0/d = 0.1/18$, EMA 0.5, $\rho \in [0, 10^{18}]$ |

Table 4: Algorithm outlined in Section E.1. Hyperparamters for Figure 7 **b)** with ARD enabled.