# Rebuttal: A simple mean field model of feature learning

November 18, 2025

## 1 Additional figures

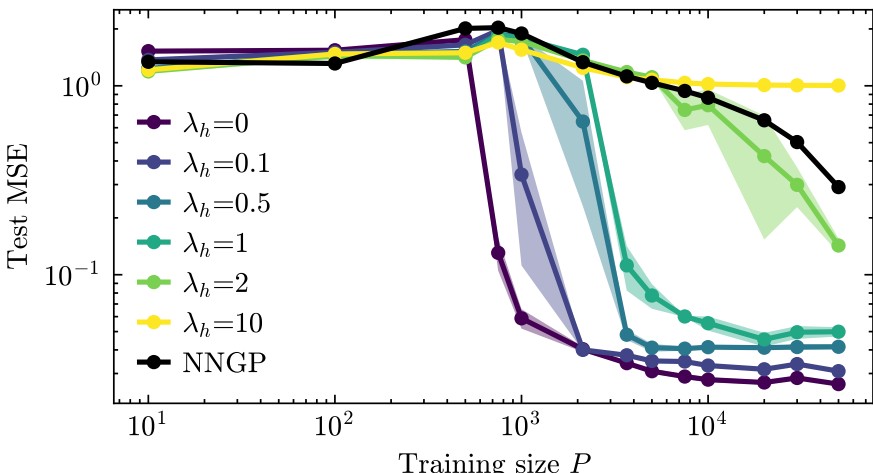

Figure 1: Generalisation error vs training set size on $k$-sparse parity with $(k = 4, d = 35)$ in dependence of the homogeneous regularization $\lambda_h$ for a ReLU network with $N = 512$. The network is trained with the following loss function $\ell_{\text{total}} = \ell_{\text{MSE}} + \lambda_h \sum_{j=1}^{N} \left[ \frac{1}{d} \sum_{i=1}^{d} \left( |w_{ij}| - \left( \frac{1}{d} \sum_{k=1}^{d} |w_{kj}| \right) \right)^2 \right]$. The regularization penalizes the variance of the coordinates of a neuron $\text{Var}(|\boldsymbol{w}_i|)$. I.e., it exactly acts against the IFS we see in real networks learning $k$-sparse parity. We see that a higher regularization $\lambda_h$ leads to the network not being able to learn the target anymore. This is strong independent evidence that IFS is necessary to learn low-dimensional targets from isotropic data.

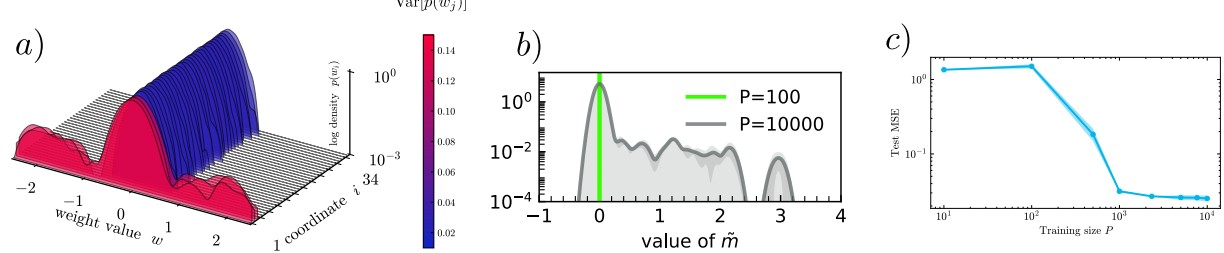

Figure 2: **a)** Repeat of Fig. 4 a) from the paper with sigmoid. We see that for sigmoid, similar to ReLU the first $j = 1, 2, 3$ coordinate marginal distributions in one neuron $p(w_j)$ develop heavy tails (we use $k = 3$ here as sigmoid cannot learn even parity). This again demonstrates the coordinate level phase transition. **b)** Repeat of Fig. 4 b). Sigmoid shows a similar specialization distribution $p(\tilde{m})$ to ReLU. **c)** Also the learning curve, i.e. the generalisation error vs training set size shows the same phase-transition type learning as for ReLU.

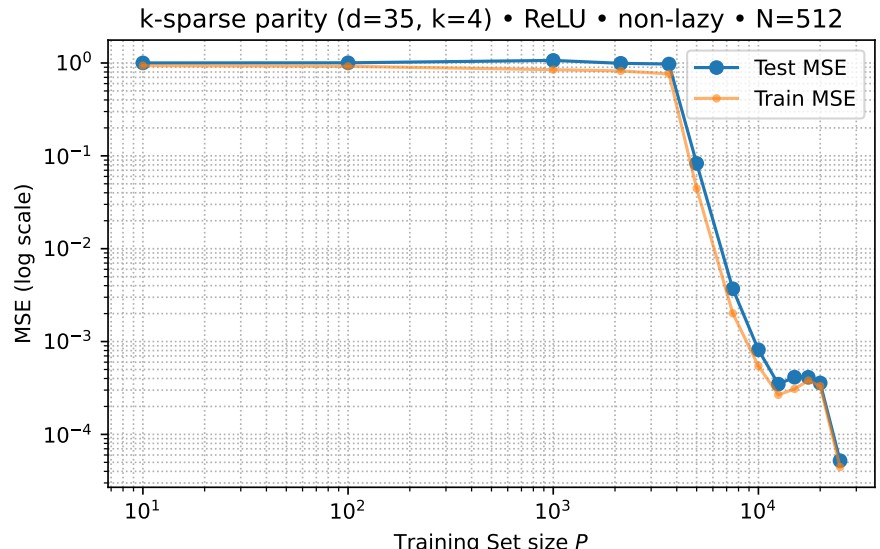

Figure 3: We use the following hyperparameters for the script: `python mala_ksparse.py -d 35 -k 4 -P_grid "10,100,1000,2133,3666,5000,7500,10000,12500,15000,17500,20000,25000" -P_test 5000 -N 512 -L 1 -phi relu -nonlazy 1 -fixed_norm 1 -temp 1e-5 -draws 10000 -tune 2000 -chains 4 -step_size 0.07 -skip_steps 1 -precision 32 -out_dir results_sparse_parity`. The result is quite sensitive to the step size. However, even for larger step sizes the drop in generalisation error never goes beyond $P^* = 5000$ while becoming very unstable for larger $P$. Our Fig. 1 a) in the paper shows a drop at approximately $P^* = 750$.

## 2 Proof of equivalence of MF theories

Our paper's single-neuron mean-field negative log posterior is

$$-\ln p_{\mathrm{MF}}(\boldsymbol{w}, a, \{m_A\}) = \frac{a^2}{2\sigma_a^2} + \frac{d}{2\sigma_w^2}|\boldsymbol{w}|^2 + \frac{a^2}{2\kappa^2 N^{2\gamma}}\Sigma(\boldsymbol{w}) - \frac{a}{\kappa^2 N^\gamma}\left(J_{\mathcal{Y}}(\boldsymbol{w}) - \sum_A m_A J_A(\boldsymbol{w})\right) \quad (1)$$

or after integrating out $a$:

$$-\ln p_{\mathrm{MF}}(\boldsymbol{w}, \{m_A\}) = \frac{1}{2}\ln\left(\frac{1}{\sigma_a^2} + \frac{\Sigma(\boldsymbol{w})}{N^{2\gamma}\kappa^2}\right) - \frac{(J_{\mathcal{Y}}(\mathbf{w}) - \sum_A m_A J_A(\mathbf{w}))^2}{2N^{2\gamma}\kappa^4\left(\frac{1}{\sigma_a^2} + \frac{\Sigma(\boldsymbol{w})}{N^{2\gamma}\kappa^2}\right)} + \frac{d}{2\sigma_w^2}\|\boldsymbol{w}\|^2 \quad (2)$$

with

$$\Sigma(\boldsymbol{w}) = \mathbb{E}_{\boldsymbol{x}}[\phi(\boldsymbol{w}^\top\boldsymbol{x})^2], \qquad J_{\mathcal{Y}}(w) = \mathbb{E}_{\boldsymbol{x}}[\phi(\boldsymbol{w}^\top\boldsymbol{x})y(\boldsymbol{x})], \qquad J_A(w) = \mathbb{E}_{\boldsymbol{x}}[\phi(\boldsymbol{w}^\top x)\chi_A(\boldsymbol{x})].$$

where the mean prediction $\langle f \rangle$ is $\langle f(\boldsymbol{x})\rangle = \sum_A m_A \chi_A(\boldsymbol{x})$ (see eq. 7 in the paper)

**Theorem 2.1.** *eq.* (2) *is equivalent to Rubin 2024's action*

$$S[w] = \frac{|w|^2}{2\sigma_w^2} - \frac{\sigma_a^2}{2N}\sum_{\mu,\nu}\bar{t}(\mathbf{x}_\mu)^T\bar{t}(\mathbf{x}_\nu)\underbrace{\phi(\mathbf{w}\cdot\mathbf{x}_\mu)\phi(\mathbf{w}\cdot\mathbf{x}_\nu)}_{:=\sigma_a^{-2}\bar{Q}_{\mu\nu}}, \quad (3)$$

*once,*

- *one neglects the $\frac{1}{2}\ln(...)$-term.*

- *one neglects the self-energy $\Sigma$-term.*

- *one re-expresses the self-consistency equation in the discrepancy field $\bar{t}(\boldsymbol{x}) := \frac{y(\boldsymbol{x}) - \langle f(\boldsymbol{x})\rangle}{\kappa^2}$.*

- *Notation-wise, there is the following identification $\kappa^2 = \sigma^2$ and $\bar{f} = \langle f \rangle$*

*Proof.* **1. Rewriting the discrepancy field**
It holds that

$$J_{\mathcal{Y}}(\boldsymbol{w}) - \sum_A m_A J_A(\boldsymbol{w}) = \sum_\mu\left[\phi(\boldsymbol{w}^\top\boldsymbol{x}_\mu)(y(\boldsymbol{x}_\mu) - \langle f(\boldsymbol{x}_\mu)\rangle)\right] \quad (4)$$

$$= \kappa^2\sum_\mu[\phi(\boldsymbol{w}^\top\boldsymbol{x}_\mu)\bar{t}(\boldsymbol{x}_\mu)]. \quad (5)$$

Hence

$$(J_{\mathcal{Y}}(\boldsymbol{w}) - \sum_A m_A J_A(\boldsymbol{w}))^2 = \kappa^4\sum_{\mu,\nu}[\phi(\boldsymbol{w}^\top\boldsymbol{x}_\mu)\bar{t}(\boldsymbol{x}_\mu)\phi(\boldsymbol{w}^\top\boldsymbol{x}_\nu)\bar{t}(\boldsymbol{x}_\nu)] \quad (6)$$

Now, we set

$$\frac{(J_{\mathcal{Y}}(\mathbf{w}) - \sum_A m_A J_A(\mathbf{w}))^2}{2N^{2\gamma}\kappa^4\left(\frac{1}{\sigma_a^2} + \frac{\Sigma(\boldsymbol{w})}{N^{2\gamma}\kappa^2}\right)} \approx \frac{(J_{\mathcal{Y}}(\mathbf{w}) - \sum_A m_A J_A(\mathbf{w}))^2}{2N^{2\gamma}\kappa^4\left(\frac{1}{\sigma_a^2}\right)} \quad (7)$$

plugging eq. (6) in gives

$$\frac{(J_{\mathcal{Y}}(\mathbf{w}) - \sum_A m_A J_A(\mathbf{w}))^2}{2N^{2\gamma}\kappa^4 \left(\frac{1}{\sigma_a^2}\right)} = \frac{\sigma_a^2 \sum_{\mu,\nu}[\phi(\boldsymbol{w}^\top \boldsymbol{x}_\mu)\bar{t}(\boldsymbol{x}_\mu)\phi(\boldsymbol{w}^\top \boldsymbol{x}_\nu)\bar{t}(\boldsymbol{x}_\nu)]}{2N^{2\gamma}} \tag{8}$$

Dropping the $\frac{1}{2}\ln(...)$-term gives the final result.

**2. The self-consistency equation**

We start from the self-consistency equation for $m_A$ and the posterior mean $\mu(\boldsymbol{w})$.

$$m_A = N^{1-\gamma} \langle \mu(\boldsymbol{w}) J_A(\boldsymbol{w}) \rangle_{p(\boldsymbol{w}|\{m_B\})} \tag{9}$$

$$\mu(\boldsymbol{w}) = \frac{\frac{1}{\kappa^2 N^\gamma}\left(J_{\mathcal{Y}}(\boldsymbol{w}) - \sum_B m_B J_B(\boldsymbol{w})\right)}{\frac{1}{\sigma_a^2} + \frac{\Sigma(\boldsymbol{w})}{N^{2\gamma}\kappa^2}} \tag{10}$$

Here we have defined the denominator as $A(\boldsymbol{w}) := \frac{1}{\sigma_a^2} + \frac{\Sigma(\boldsymbol{w})}{N^{2\gamma}\kappa^2}$.

**a) Reconstruct the mean prediction $\langle f \rangle$:** The mean prediction $\langle f(\boldsymbol{x}_\mu) \rangle$ is reconstructed from the mean-field parameters $m_A$ and the basis functions $\chi_A(\boldsymbol{x})$ (which we assume form an orthonormal basis on the data points, i.e., $\sum_A \chi_A(\boldsymbol{x}_\mu)\chi_A(\boldsymbol{x}_\nu) = \delta_{\mu\nu}$).

$$\langle f(\boldsymbol{x}_\mu) \rangle := \sum_A m_A \chi_A(\boldsymbol{x}_\mu) = \sum_A \left(N^{1-\gamma} \langle \mu(\boldsymbol{w}) J_A(\boldsymbol{w}) \rangle\right) \chi_A(\boldsymbol{x}_\mu)$$

We swap the order of summation and expectation:

$$\langle f(\boldsymbol{x}_\mu) \rangle = N^{1-\gamma} \left\langle \mu(\boldsymbol{w}) \left(\sum_A J_A(\boldsymbol{w})\chi_A(\boldsymbol{x}_\mu)\right) \right\rangle$$

**b) Simplify the inner sum:** We expand $J_A(\boldsymbol{w})$ (using empirical data sums instead of expectations): $J_A(\boldsymbol{w}) = \sum_\nu \phi(\boldsymbol{w}^\top \boldsymbol{x}_\nu)\chi_A(\boldsymbol{x}_\nu)$.

$$\sum_A J_A(\boldsymbol{w})\chi_A(\boldsymbol{x}_\mu) = \sum_A \left(\sum_\nu \phi(\boldsymbol{w}^\top \boldsymbol{x}_\nu)\chi_A(\boldsymbol{x}_\nu)\right)\chi_A(\boldsymbol{x}_\mu)$$

$$= \sum_\nu \phi(\boldsymbol{w}^\top \boldsymbol{x}_\nu)\left(\sum_A \chi_A(\boldsymbol{x}_\nu)\chi_A(\boldsymbol{x}_\mu)\right)$$

$$= \sum_\nu \phi(\boldsymbol{w}^\top \boldsymbol{x}_\nu)\delta_{\mu\nu} \quad \text{(using orthonormality)}$$

$$= \phi(\boldsymbol{w}^\top \boldsymbol{x}_\mu)$$

**c) Substitute back and insert $\mu(\boldsymbol{w})$:** Plugging this result back into the equation for $\langle f(\boldsymbol{x}_\mu) \rangle$ gives a simpler form:

$$\langle f(\boldsymbol{x}_\mu) \rangle = N^{1-\gamma} \left\langle \mu(\boldsymbol{w})\phi(\boldsymbol{w}^\top \boldsymbol{x}_\mu) \right\rangle \tag{11}$$

Now, we insert the full expression for $\mu(\boldsymbol{w})$:

$$\langle f(\boldsymbol{x}_\mu) \rangle = N^{1-\gamma} \left\langle \left(\frac{\frac{1}{\kappa^2 N^\gamma}\left(J_{\mathcal{Y}}(\boldsymbol{w}) - \sum_B m_B J_B(\boldsymbol{w})\right)}{A(\boldsymbol{w})}\right)\phi(\boldsymbol{w}^\top \boldsymbol{x}_\mu) \right\rangle$$

Using the definition of the discrepancy field $\bar{t}$ from eq. (6):

$$\langle f(\boldsymbol{x}_\mu)\rangle = N^{1-\gamma}\left\langle\left(\frac{\frac{1}{\kappa^2 N^\gamma}\left(\kappa^2\sum_\nu \phi(\boldsymbol{w}^\top\boldsymbol{x}_\nu)\bar{t}(\boldsymbol{x}_\nu)\right)}{A(\boldsymbol{w})}\right)\phi(\boldsymbol{w}^\top\boldsymbol{x}_\mu)\right\rangle$$

$$= N^{1-\gamma}\left\langle\frac{1}{A(\boldsymbol{w})}\left(\frac{1}{N^\gamma}\sum_\nu \phi(\boldsymbol{w}^\top\boldsymbol{x}_\nu)\bar{t}(\boldsymbol{x}_\nu)\right)\phi(\boldsymbol{w}^\top\boldsymbol{x}_\mu)\right\rangle$$

**d) Rearrange and define the Kernel Q:** We group the constants and swap the sum over $\nu$ with the $\boldsymbol{w}$-expectation:

$$\langle f(\boldsymbol{x}_\mu)\rangle = N^{1-2\gamma}\left\langle\frac{\phi(\boldsymbol{w}^\top\boldsymbol{x}_\mu)}{A(\boldsymbol{w})}\sum_\nu \phi(\boldsymbol{w}^\top\boldsymbol{x}_\nu)\bar{t}(\boldsymbol{x}_\nu)\right\rangle$$

$$= \sum_\nu\left(\underbrace{\left\langle\frac{N^{1-2\gamma}}{A(\boldsymbol{w})}\phi(\boldsymbol{w}^\top\boldsymbol{x}_\mu)\phi(\boldsymbol{w}^\top\boldsymbol{x}_\nu)\right\rangle}_{:=Q_{\mu\nu}}\right)\bar{t}(\boldsymbol{x}_\nu)$$

This defines the exact kernel matrix $\mathbf{Q}$ with components $Q_{\mu\nu}$. To get the kernel from Paper 2, we apply the stated approximation: $A(\boldsymbol{w})\approx 1/\sigma_a^2$.
Applying these approximations to the kernel definition:

$$Q_{\mu\nu}\approx\left\langle\frac{1}{(1/)}\phi(\boldsymbol{w}^\top\boldsymbol{x}_\mu)\phi(\boldsymbol{w}^\top\boldsymbol{x}_\nu)\right\rangle = \left\langle\phi(\boldsymbol{w}^\top\boldsymbol{x}_\mu)\phi(\boldsymbol{w}^\top\boldsymbol{x}_\nu)\right\rangle$$

This is precisely the kernel definition $Q_{\mu\nu}=\sigma_a^2\langle\phi(\boldsymbol{w}\cdot\boldsymbol{x}_\mu)\phi(\boldsymbol{w}\cdot\boldsymbol{x}_\nu)\rangle_{S[\boldsymbol{w}]}$.

**e) Solve the fixed-point equation:** Using our kernel $\mathbf{Q}$, the self-consistency equation is:

$$\langle f(\boldsymbol{x}_\mu)\rangle = \sum_\nu Q_{\mu\nu}\bar{t}(\boldsymbol{x}_\nu)$$

In vector notation, this is $\langle\boldsymbol{f}\rangle = \mathbf{Q}\bar{\mathbf{t}}$. Now, we substitute the definition of $\bar{\mathbf{t}}$:

$$\bar{\mathbf{t}} = \frac{1}{\kappa^2}(\boldsymbol{y}-\langle\boldsymbol{f}\rangle)$$

This gives the final equation for $\langle\boldsymbol{f}\rangle$:

$$\langle\boldsymbol{f}\rangle = \mathbf{Q}\left(\frac{1}{\kappa^2}(\boldsymbol{y}-\langle\boldsymbol{f}\rangle)\right)$$
$$\kappa^2\langle\boldsymbol{f}\rangle = \mathbf{Q}(\boldsymbol{y}-\langle\boldsymbol{f}\rangle)$$
$$\kappa^2\langle\boldsymbol{f}\rangle = \mathbf{Q}\boldsymbol{y}-\mathbf{Q}\langle\boldsymbol{f}\rangle$$
$$\kappa^2\langle\boldsymbol{f}\rangle+\mathbf{Q}\langle\boldsymbol{f}\rangle = \mathbf{Q}\boldsymbol{y}$$
$$(\kappa^2\mathbf{I}+\mathbf{Q})\langle\boldsymbol{f}\rangle = \mathbf{Q}\boldsymbol{y}$$
$$\langle\boldsymbol{f}\rangle = (\mathbf{Q}+\kappa^2\mathbf{I})^{-1}\mathbf{Q}\boldsymbol{y}$$

Since $\mathbf{Q}$ and $(\mathbf{Q}+\kappa^2\mathbf{I})$ commute, this is identical to:

$$\boxed{\langle\boldsymbol{f}\rangle = \mathbf{Q}(\mathbf{Q}+\kappa^2\mathbf{I})^{-1}\boldsymbol{y}} \tag{12}$$

Lastly, we identify $\kappa^2\leftrightarrow\sigma^2$ for the notation from Rubin 2024.

$\square$

# 3 Code for Fig. 3

```python
"""
Learning curve for k-sparse parity (d=35, k=4) with a one-hidden-layer ReLU NN
in the non-lazy regime, using JAX MALA sampling
Saves a results .npz and plots.

This version:
- No json_and_hash (no dataset signature issues).
- --precision {32,64} (default 32) to reduce memory.
- Makes sampling keep ~`draws` results, not ~10x more.
- Excludes warmup samples from prediction.
- Computes predictive means in chunks to avoid OOM.
"""

import argparse
import os
import re
from time import time
from datetime import datetime
from functools import partial

import numpy as np
import matplotlib.pyplot as plt

import jax
from jax import random, config, jit
import jax.numpy as jnp

# Your codebase bits
from nonlazy.functions_jax import choose_phi as choose_phi_jax
import nonlazy.mala_jax as mala

# ---------------------------
# Dataset: k-sparse parity
# ---------------------------

def make_sparse_parity_dataset(rng: np.random.Generator, d: int, k: int,
                               P: int, support=None):
    """
    X in {-1,+1}^d i.i.d. Rademacher. A fixed hidden support S (|S|=k).
    Label y = product_{i in S} X_i in {-1,+1}.
    Returns X (P,d), y (P,), and the support S used.
    """
    if support is None:
        support = rng.choice(d, size=k, replace=False)
    X = rng.integers(0, 2, size=(P, d), dtype=np.int8) * 2 - 1   # ±1
    y = np.prod(X[:, support], axis=1).astype(np.float32)        # ±1
    return X.astype(np.float32), y, support

# ---------------------------
# Model & sampler (based on your code 3)
# ---------------------------

def build_network_fns(N_in, N, L, phi, nonlazy: bool, fixed_norm: bool, dtype):
    """
    Returns (neural_network, neural_network_cs) matching your code 3 interface.
```

```python
58          """
59          assert L == 1, "This script is for a single hidden layer (L=1)."
60
61          @jit
62          def neural_network(input, weight_list):
63              # weight_list[0] shape: (N, N_in)
64              h = jnp.dot(weight_list[0], input.T) / jnp.sqrt(N_in)    # (N, P)
65              act = phi(h)
66              if fixed_norm:
67                  return jnp.dot(weight_list[1], act) / N              # (P,)
68              else:
69                  return jnp.dot(weight_list[1], act) / jnp.sqrt(N)
70
71          @jit
72          def neural_network_cs(input, weight_lists):
73              # weight_lists shapes per chain/sample: (..., N, N_in), (..., N)
74              h = jnp.einsum("...mn,pn->...mp", weight_lists[0], input) / jnp.sqrt(N_in)
75              act = phi(h)
76              if fixed_norm:
77                  return jnp.einsum("...m,...mp->...p", weight_lists[1], act) / N
78              else:
79                  return jnp.einsum("...m,...mp->...p", weight_lists[1], act) / jnp.sqrt(N)
80
81          return neural_network, neural_network_cs
82
83
84  def make_logprob_fn(N_in, N, L, sig, sig_scale,
85                      nonlazy: bool, fixed_norm: bool, temp: float,
86                      neural_network):
87          """
88          Builds the posterior log-density (per chain).
89          """
90          assert L == 1
91
92          def logprob_network(weights, input, output, temp, sig_a_scaled):
93              # Unpack flat weights into [w_in, w_out]
94              weight_list = mala._reshape_weights(weights, N_in, N, L)
95
96              # Priors
97              logprob = -jnp.sum(weight_list[0] ** 2, axis=(0, 1)) / (2 * (sig * sig_scale[0]) ** 2)
98              logprob += -jnp.sum(weight_list[1] ** 2, axis=0) / (2 * sig_a_scaled**2)
99
100             # Likelihood: Normal with variance temp/N (nonlazy) or temp (lazy)
101             resid = output - neural_network(input=input, weight_list=weight_list)  # (P,)
102             if nonlazy:
103                 logprob += -(jnp.sum(resid ** 2, axis=0) * N) / (2 * temp)
104             else:
105                 logprob += -(jnp.sum(resid ** 2, axis=0)) / (2 * temp)
106             return logprob
107
108         return logprob_network
109
110
111 def init_weights_numpy(rng, N_in, N, sig, sig_scale, sig_a_scaled, dtype):
112         w_in = rng.normal(0, sig * sig_scale[0], (N, N_in)).astype(np.float32)
113         w_out = rng.normal(0, sig_a_scaled, (N,)).astype(np.float32)
114         flat = np.concatenate([w_in.reshape(-1), w_out.reshape(-1)]).astype(np.float32)
115         return jnp.array(flat, dtype=dtype)
116
117
```

```python
118  def _parse_pgrid(s_or_list):
119      """
120      Accepts:
121        - a single string: "10 100 1000" or "10,100,1000" or mixed
122        - a list of strings: ["10", "100", "1000"] (from nargs="+")
123      """
124      if isinstance(s_or_list, list):
125          s = " ".join(s_or_list)
126      else:
127          s = str(s_or_list)
128      return [int(x) for x in re.split(r"[,\s]+", s.strip()) if x]
129
130
131  def predict_mean_in_chunks(X, samples_cs, neural_network_cs, chunk_draws=64):
132      """
133      Compute posterior predictive mean E[f(X;weights)] by chunking draws
134      to keep memory bounded. samples_cs = [w_in, w_out] with shape
135      [chains, draws, ...] each.
136      """
137      C = samples_cs[0].shape[0]
138      D = samples_cs[0].shape[1]
139      total = C * D
140      acc = np.zeros(X.shape[0], dtype=np.float64)
141      for start in range(0, D, chunk_draws):
142          end = min(D, start + chunk_draws)
143          sub = [samples_cs[0][:, start:end, ...], samples_cs[1][:, start:end, ...]]
144          preds = neural_network_cs(X, sub)                    # (C, end-start, P)
145          acc += np.sum(np.array(preds, dtype=np.float64), axis=(0, 1))
146      return acc / total
147
148
149  def run_one_fit(
150      *, key, rng, X_train, y_train, X_test, y_test,
151      d, k, N, L, phi, nonlazy, fixed_norm, temp, sig, sig_scale,
152      draws, tune, chains, step_size, skip_steps, precision
153  ):
154      """
155      Runs warmup + sampling (MALA), returns predictive means and metrics.
156      """
157      # Precision / dtype
158      if precision == 64:
159          dtype = jnp.float64
160      else:
161          dtype = jnp.float32
162
163      N_in = X_train.shape[1]
164      neural_network, neural_network_cs = build_network_fns(N_in, N, L, phi, nonlazy, fixed_norm,
     ↪  dtype)
165
166      # Non-lazy fixed-norm scaling for readout (L=1): sigma_a / sqrt(P_train)
167      P_train = X_train.shape[0]
168      if fixed_norm:
169          sig_a_scaled = sig * sig_scale[L] / np.sqrt(P_train)   # L=1 => sqrt(P)
170      else:
171          sig_a_scaled = sig * sig_scale[L]
172
173      # Build base logprob and single-arg wrapper (weights only)
174      base_logprob = make_logprob_fn(N_in, N, L, sig, sig_scale, nonlazy, fixed_norm, temp,
     ↪  neural_network)
```

```
175        logprob = partial(base_logprob, input=X_train, output=y_train, temp=temp,
        ↪  sig_a_scaled=sig_a_scaled)
176        logprob = jit(logprob)
177
178        # Initial weights per chain
179        init_w = jnp.array([init_weights_numpy(rng, N_in, N, sig, sig_scale, sig_a_scaled, dtype) for _
        ↪  in range(chains)])
180
181        # Effective step size (matches your code 3)
182        eff_step = (temp * step_size / N) if nonlazy else (temp * step_size)
183
184        # We keep approx. `draws` posterior samples (not 10× more).
185        # We also keep warmups small and DO NOT use warmup samples for prediction.
186        micro_per_result = max(1, int(round(1.0 / step_size)))              # e.g., 10 for
        ↪  step_size=0.1
187        gd_results       = max(1, int(round(0.4 * tune)))                  # fewer stored GD-like
        ↪  results
188        warm_results     = max(1, int(round(0.2 * tune)))                  # fewer stored warmup
        ↪  results
189        samp_results     = max(1, int(draws))                              # ~draws stored samples
190
191        gd_args = dict(
192            step_size=eff_step,
193            metropolis=False,
194            noise=True,
195            logprob=logprob,
196            skip_steps=micro_per_result,
197            steps=gd_results,
198        )
199        warmup_args = dict(
200            step_size=eff_step,
201            metropolis=True,
202            noise=True,
203            logprob=logprob,
204            skip_steps=micro_per_result,
205            steps=warm_results,
206        )
207        sampling_args = dict(
208            step_size=eff_step,
209            metropolis=True,
210            noise=True,
211            logprob=logprob,
212            skip_steps=max(1, int(round(skip_steps / step_size))),  # decimation between kept results
213            steps=samp_results,
214        )
215
216        # Warmup 1 (gd-like)
217        key, subkey = random.split(key)
218        warm1, lp1, acc1 = mala.langevin_pmap(init_w, subkey, chains, **gd_args)
219
220        # Warmup 2 (short Metropolis)
221        key, subkey = random.split(key)
222        warm2, lp2, acc2 = mala.langevin_pmap(warm1[:, -1, :], subkey, chains, **warmup_args)
223
224        # Sampling
225        key, subkey = random.split(key)
226        posterior_samples, lp3, acc3 = mala.langevin_pmap(warm2[:, -1, :], subkey, chains,
        ↪  **sampling_args)
227
228        # Keep ONLY posterior samples for prediction (exclude warmups)
```

```python
229         samples_cs = mala.reshape_weights_cs(posterior_samples, N_in, N, L)  # list: [w_in, w_out]
230
231         # Predictive mean in chunks to keep memory bounded
232         mu_train = predict_mean_in_chunks(X_train, samples_cs, neural_network_cs, chunk_draws=64)
233         mu_test  = predict_mean_in_chunks(X_test,  samples_cs, neural_network_cs, chunk_draws=64)
234
235         # Metrics
236         mse_train = float(np.mean((mu_train - y_train) ** 2))
237         mse_test  = float(np.mean((mu_test  - y_test)  ** 2))
238         acc_train = float(np.mean(np.sign(mu_train) == y_train))
239         acc_test  = float(np.mean(np.sign(mu_test)  == y_test))
240
241         summary = {
242             "acc_train": acc_train,
243             "acc_test": acc_test,
244             "mse_train": mse_train,
245             "mse_test": mse_test,
246             "acceptance_rate_warmup": float(np.mean(np.array(acc2)[:, -1])),
247             "acceptance_rate_sampling": float(np.mean(np.array(acc3)[:, -1])),
248         }
249         return mu_train, mu_test, summary
250
251
252 def main():
253     parser = argparse.ArgumentParser()
254     # Task/data
255     parser.add_argument("--d", type=int, default=35)
256     parser.add_argument("--k", type=int, default=4)
257     parser.add_argument("--P_test", type=int, default=5000)
258     # Accept both: --P_grid 10 100 1000  OR  --P_grid "10,100,1000"
259     parser.add_argument("--P_grid", nargs="+", type=str, default=["32,64,128,256,512,1024"])
260
261     # Network/sampling
262     parser.add_argument("--N", type=int, default=512)         # width
263     parser.add_argument("--L", type=int, default=1)           # fixed to 1 here
264     parser.add_argument("--nonlazy", type=int, default=1)
265     parser.add_argument("--fixed_norm", type=int, default=1)
266     parser.add_argument("--phi", type=str, default="relu")
267     parser.add_argument("--sig", type=float, default=1.0)
268     parser.add_argument("--sig_scale", nargs="+", type=float, default=[1.0, 1.0])
269     parser.add_argument("--temp", type=float, default=1e-4)
270
271     parser.add_argument("--draws", type=int, default=300)    # smaller default to start
272     parser.add_argument("--tune", type=int, default=600)     # smaller warmup storage
273     parser.add_argument("--chains", type=int, default=2)
274     parser.add_argument("--step_size", type=float, default=0.1)
275     parser.add_argument("--skip_steps", type=int, default=1)
276     parser.add_argument("--precision", type=int, choices=[32, 64], default=32)
277
278     parser.add_argument("--seed", type=int, default=12345)
279     parser.add_argument("--out_dir", type=str, default="results_sparse_parity")
280     args = parser.parse_args()
281
282     # Precision must be set BEFORE any JAX array creation
283     config.update("jax_enable_x64", args.precision == 64)
284
285     # Parse P_grid (handles spaces or commas)
286     P_grid = _parse_pgrid(args.P_grid)
287
288     # Validate/prepare
```

```
289        L = args.L
290        assert L == 1, "This script is specialized to L=1."
291        nonlazy = bool(args.nonlazy)
292        fixed_norm = bool(args.fixed_norm)
293        assert nonlazy and fixed_norm, "For non-lazy ReLU learning curve, set --nonlazy 1 --fixed_norm
           ↪    1."
294
295        # Sig scale vector length = L+1
296        sig_scale = args.sig_scale
297        if len(sig_scale) == 1:
298            sig_scale = sig_scale * (L + 1)
299        assert len(sig_scale) == L + 1, f"sig_scale must have length L+1={L+1}."
300
301        # Devices vs. chains: pmap requires axis size <= #devices
302        n_dev = jax.device_count()
303        chains_eff = min(args.chains, n_dev if n_dev > 0 else 1)
304        if chains_eff < args.chains:
305            print(f"[warn] Reducing chains from {args.chains} to {chains_eff} to match available
           ↪    devices ({n_dev}).")
306
307        # PRNGs
308        rng = np.random.default_rng(args.seed)
309        key = random.PRNGKey(args.seed + 1)
310
311        # Activation
312        phi = choose_phi_jax(args.phi)   # 'relu' -> jnp.maximum(0, x)
313
314        # Fixed sparse support for all runs
315        support = rng.choice(args.d, size=args.k, replace=False)
316
317        # Prepare output dirs
318        os.makedirs(args.out_dir, exist_ok=True)
319        figs_dir = os.path.join(args.out_dir, "figs")
320        os.makedirs(figs_dir, exist_ok=True)
321        data_dir = os.path.join(args.out_dir, "data")
322        os.makedirs(data_dir, exist_ok=True)
323
324        # Manual tag
325        tag = (
326            f"ksparse_parity_d{args.d}_k{args.k}_N{args.N}_"
327            f"{args.phi}_nonlazy{int(nonlazy)}_fixed{int(fixed_norm)}_p{args.precision}_"
328            f"{datetime.now().strftime('%Y%m%d_%H%M%S')}"
329        )
330
331        # Test set (fixed across P)
332        X_test, y_test, _ = make_sparse_parity_dataset(rng, args.d, args.k, args.P_test,
           ↪    support=support)
333
334        # Sweep P_train
335        P_list, acc_test_list, acc_train_list, mse_test_list, mse_train_list = [], [], [], [], []
336        for P in P_grid:
337            now = datetime.now().time().strftime("%H:%M:%S")
338            print(f"\n[{now}] === Running P_train = {P} ===")
339            X_train, y_train, _ = make_sparse_parity_dataset(rng, args.d, args.k, P, support=support)
340
341            tic = time()
342            _, _, summary = run_one_fit(
343                key=key, rng=rng,
344                X_train=X_train, y_train=y_train,
345                X_test=X_test, y_test=y_test,
```

```
346                d=args.d, k=args.k, N=args.N, L=args.L, phi=phi,
347                nonlazy=nonlazy, fixed_norm=fixed_norm, temp=args.temp,
348                sig=args.sig, sig_scale=sig_scale,
349                draws=args.draws, tune=args.tune, chains=chains_eff,
350                step_size=args.step_size, skip_steps=args.skip_steps,
351                precision=args.precision,
352            )
353            toc = time()
354            print(f"  done in {toc - tic:.1f}s | "
355                  f"acc_test={summary['acc_test']:.3f}, mse_test={summary['mse_test']:.4f}, "
356                  f"acc_train={summary['acc_train']:.3f}, "
                  ↪  ar_samp={summary['acceptance_rate_sampling']:.3f}")

357
358            P_list.append(P)
359            acc_test_list.append(summary["acc_test"])
360            acc_train_list.append(summary["acc_train"])
361            mse_test_list.append(summary["mse_test"])
362            mse_train_list.append(summary["mse_train"])

363
364        # Sort by P just in case
365        order = np.argsort(np.array(P_list))
366        P_arr        = np.array(P_list)[order]
367        acc_test_arr  = np.array(acc_test_list)[order]
368        acc_train_arr = np.array(acc_train_list)[order]
369        mse_test_arr  = np.array(mse_test_list)[order]
370        mse_train_arr = np.array(mse_train_list)[order]

371
372        # Save meta + results
373        meta = dict(
374            dataset="ksparse_parity",
375            d=args.d, k=args.k, support=support.tolist(),
376            P_grid=P_arr.tolist(), P_test=args.P_test,
377            N=args.N, L=args.L,
378            nonlazy=nonlazy, fixed_norm=fixed_norm, phi=args.phi,
379            sig=args.sig, sig_scale=sig_scale, temp=args.temp,
380            draws=args.draws, tune=args.tune, chains=chains_eff,
381            step_size=args.step_size, skip_steps=args.skip_steps, seed=args.seed,
382            precision=args.precision,
383        )
384        np.savez(
385            os.path.join(data_dir, f"lc_{tag}.npz"),
386            P=P_arr, acc_test=acc_test_arr, acc_train=acc_train_arr,
387            mse_test=mse_test_arr, mse_train=mse_train_arr,
388            meta=meta,
389        )

390
391        # Plot learning curve (test accuracy)
392        plt.figure(figsize=(6,4))
393        plt.plot(P_arr, acc_test_arr, marker="o", label="Test accuracy")
394        plt.plot(P_arr, acc_train_arr, marker=".", alpha=0.6, label="Train accuracy")
395        plt.xscale("log")
396        plt.ylim(0.0, 1.05)
397        plt.xlabel("Training Set size $P$")
398        plt.ylabel("Accuracy")
399        plt.title(f"k-sparse parity (d={args.d}, k={args.k}) ● ReLU ● non-lazy ● N={args.N}")
400        plt.grid(True, which="both", ls=":")
401        plt.legend()
402        plt.tight_layout()
403        plt.savefig(os.path.join(figs_dir, f"learning_curve_acc_{tag}.png"), dpi=150)
404        plt.savefig(os.path.join(figs_dir, f"learning_curve_acc_{tag}.pdf"))
```

```
405        plt.close()
406
407        # Also plot MSE
408        plt.figure(figsize=(6,4))
409        plt.plot(P_arr, mse_test_arr, marker="o", label="Test MSE")
410        plt.plot(P_arr, mse_train_arr, marker=".", alpha=0.6, label="Train MSE")
411        plt.xscale("log")
412        plt.yscale("log")
413        plt.xlabel("Training Set size $P$")
414        plt.ylabel("MSE (log scale)")
415        plt.title(f"k-sparse parity (d={args.d}, k={args.k}) • ReLU • non-lazy • N={args.N}")
416        plt.grid(True, which="both", ls=":")
417        plt.legend()
418        plt.tight_layout()
419        plt.savefig(os.path.join(figs_dir, f"learning_curve_mse_{tag}.png"), dpi=150)
420        plt.savefig(os.path.join(figs_dir, f"learning_curve_mse_{tag}.pdf"))
421        plt.close()
422
423        print(f"\nSaved results to: {os.path.join(data_dir, f'lc_{tag}.npz')}")
424        print(f"Saved plots to:   {os.path.join(figs_dir, f'learning_curve_acc_{tag}.png')} and .pdf")
425
426  if __name__ == "__main__":
427        main()
```