# OpenReview forum: "A simple mean field model of feature learning"
_ICLR.cc/2026/Conference — Submitted to ICLR 2026_

### Official Review · Reviewer_mzGy · 2025-10-21

**Soundness:** 2
**Presentation:** 2
**Contribution:** 2
**Rating:** 4
**Confidence:** 4

**Summary:**

This paper develops a tractable mean-field (MF) theory to understand feature learning in two-layer neural networks trained with SGLD. The authors identify a two-stage feature learning process: (1) a phase transition where networks align with target functions, and (2) self-reinforcing input feature selection (IFS). This provides an extension over the recently developed MF theory. Experiments on k-sparse parity and single-index models demonstrate quantitative agreement of the theory with SGLD-trained networks.

**Strengths:**

1.	Analytical Tractability and Completeness: The paper provides a self-consistent Mean-Field framework for the Bayesian posterior of a two-layer network. This level of analytical derivation, together with the appropriate approximation, leads to closed-form solutions, which is a valuable result.
2.	Explicit Verification: The theoretical predictions are explicitly validated against SGLD simulations on the teacher-student setup, demonstrating a strong match between the Mean-Field approximation and the numerical results, particularly regarding the phase transition phenomenon.

**Weaknesses:**

1.	Limited Novelty: The main results, including the phase transition and the sparsification of neurons, are not genuinely novel. Furthermore, the phenomena of feature learning and the splitting of network behavior to different activation function were thoroughly explored by Van-Meegen-Sompolinsky Nat. Comm 2025 in earlier, comprehensive studies, which the current work fails to cite and compare there results to. The current paper does not provide key new insights beyond these existing frameworks, especially regarding the challenging question of extending these ideas to deeper neural networks.

2.	Narrow Scope and Lack of Generalization: The analysis is restricted to a single, synthetic teacher-student task (Hermite polynomials) and demonstrates results limited to this toy setting. The paper fails to provide any insights or validation on real-world data, which limits the scope and relevance of the conclusions compared to related works.

3.	The Introduction is insufficient, failing to clearly articulate the existing literature and the specific gap the paper aims to fill, especially given the extensive prior work on mean-field feature learning.

**Questions:**

1.	Contextualization and Novelty: please properly cite provide work and provide detail comparison highlighting the novelly of the current work, and how your results go beyond the established phase diagrams and insights provided in the paper by Meegen-Sompolinsky.

2.	Introduction and MF Theory Definition: The Introduction needs complete rewriting to clearly state the problem and the specific theoretical gap.

3. Line 107: Please remove the statement in Line 107 regarding the comparison between SGLD noise and SGD "batch noise," as this is misleading and conceptually inaccurate.

4.	Theorem Clarity (Theorem 4.1, Eq. (16)):
        (a)  Equation (16) uses both asymptotic notation ($\asymp$) and asymptotic bounds ($\Theta$). Please use only one type of notation for consistency.

        (b)  Please explicitly state in the main text whether the theorem result is specifically derived for the parity model or holds more generally.

       (c)  Please explicitly specify all assumptions required for the theorem to hold, including the assumptions on $\chi_S$ and any extra assumptions currently hidden in the Appendix.

5.	Generalization of $\phi$: The generalization results focus heavily on the overlap $\phi$ with the teacher function. All results both theoretical and numerical are stated for RELU. Can the authors provide more explicit results or discussion on other activation functions, and how do the phase transition and generalization error change? Is the same prior valid for all activation functions? How does it compare with existing MF theories.

6.	Can you provide examples where the sparse priors would be inappropriate?
7.    Extension to depth: Can the ARD-MF idea apply layer-wise in deeper networks? Do you have any preliminary evidence?

Van Meegen, A., & Sompolinsky, H. (2025). Coding schemes in neural networks learning classification tasks. Nature Communications, 16(1), 3354.

---

> ### Author Response · Authors · 2025-11-18
> **Rebuttal to reviewer mzGy**
>
> We thank the referee for her/his generally positive evaluation and supportive response.
>
> # Weaknesses
>
> - **1. Limited Novelty:** We kindly ask the reviewer to read our general comment above about the Van-Meegen-Sompolinsky  paper (vMS2024) and hope this clarifies the novelty of our work.  While we have not yet extended to deeper networks, you have to start somewhere, and we are currently working on different architectures and datasets.
> - **2. Narrow Scope and Lack of Generalisation:**  Our analysis focuses on $k$-sparse parity and single-index models. These serve as important theoretical model systems for understanding how  NNs learn low-dimensional target functions from high-dimensional, isotropic input distributions, scenarios where kernel methods provably fail to achieve polynomial sample complexity. However, standard image datasets like MNIST and CIFAR-10 do  satisfy criteria (a) and (b) outlined in our general comments (part (1)), and on these datasets, the performance gap between NNs and kernel methods is modest ([1],[2]), and feature learning is harder to study there.
> - **Weakness 3:**  Again, we want to refer the reviewer to the general comment.
>  Our starting point was to understand how NNs achieve better sample complexity than their kernel counterparts (e.g. NNGP) through feature learning. As outlined in the comment, an especially interesting setting is isotropic high-dimensional data with low dimensional targets becasue kernels are not able to learn these datasets with polynomial sample complexity. Our paper aims to explain how NNs can discover the low-dimensional target in the high-dimensional isotropic data. We will reframe the introduction accordingly.
>
>
> # Questions
>
> - **Q1 Contextualization and Novelty:**  See general comment.
>
> - **Q2 Introduction and MF Theory Definition:** See weakness 3.
>
> - **Q3 Line 107:** Thanks, we will remove this.
>
> - **Q4 Theorem Clarity (Theorem 4.1, Eq. (16)):**  **a)** Thanks, we will remove the $\Theta$ notation. **b)** Yes, right now the Theorem only holds for $k$-sparse parity as we mainly focused on this setting. It has all the important properties (isotropic data $X$, low dimensional target $Y$). However, we assume  the Theorem can be easily extended to the single index model, as the Theorem does not rely on any special property of $k$-sparse parity, besides the $\varepsilon$-symmetry breaking which is still true for a single index model as long as $k\ll d$. The $\sqrt{\frac{1}{dk}}$ vs $\sqrt{\frac{1}{k}}$ difference should remain, however the Theorem does not show two constants called $D_k,C_k$. These constants will change, as they explicitly depend on expectation values over the target-model overlap. We are happy to work out the algebra for the final version. **c)** Assumptions: 1) hidden layer as described in eq. 3. 2) Teacher is a single parity basis $y(\mathbf{x})=\chi_S(\mathbf{x})$ with $S \subseteq \{1,...,d\}$. 3) $P \rightarrow \infty$ limit 4) $\phi=$ ReLU (for now) 5) $\varepsilon$-symmetry breaking 5) $\beta_0=d/ \sigma_w^2$ as in our empirical trainings 6) bounded input data.
>
> - **Q5 Generalization of  $\phi$:** Yes, please see our general comment where we show IFS also happens for $\phi=$sigmoid (or directly Fig. 2 in the new supplementary material). We are happy to also add other activation functions but we hope showing IFS for sigmoid is enough for the proof of concept.  IFS is how NNs discover the subspace the targets $Y$ live in. This effect is largely independent of the activation function. We can extend Theorem 4.1 to other activation functions like sigmoid. The only important assumptions are that the activation functions have bounded first and second derivatives. Then, we can expand the central quantities like  $\Sigma(\alpha) = \mathbb{E}[\phi(\alpha s)^2] \sim c_2 \alpha^2 + O(\alpha^4)$ and $J_Y(\alpha) = \mathbb{E}[\phi(\alpha s) y(x)] \sim c_m \alpha^m + O(\alpha^{m+2}),$ with $s=\sum_{j \in S} x_j$. $m$ is the lowest order at which the coupling with the teacher does not vanish by symmetry. E.g. this shows sigmoid cannot learn even parity. However, for odd parity the  $\sqrt{\frac{1}{dk}}$ vs $\sqrt{\frac{1}{k}}$ difference in scaling will hold but again the constants $C_k,D_k$ will change.
>
> - **Q6:** Yes, whenever the target is not low-dimensional or when the target is very smooth as argued in this [3].
>
> - **Q7 Extension to depth:** We think that the IFS mechanism is an essential part in discovering the low dimensional target manifold $Y$ when the data  $X$ is high-dimensional and isotropic and that this mechanism should be at play for deeper networks as well.. We assume that this happens in the early layers of deep NNs. The higher layers then built upon the discovered low-dimensional manifold.  We are currently working on extending our formalism to deeper NNs.
>
> [1]:https://arxiv.org/pdf/2003.02237
> [2]: https://arxiv.org/pdf/2507.19680
> [3]: https://arxiv.org/abs/2206.12314

---

### Official Review · Reviewer_dnTH · 2025-10-29

**Soundness:** 3
**Presentation:** 3
**Contribution:** 2
**Rating:** 2
**Confidence:** 4

**Summary:**

This paper presents a theoretical analysis of FL in two-layer non-linear networks trained with SGLD. The authors argue that standard MF approaches to this kind of problem, while correctly predicting the onset of FL as a phase transition, fail to capture the significant generalization improvements that follow. They attribute this failure to a FL mechanism they term Input Feature Selection (IFS). Relying on simple intuitive reasoning, the authors argue that SGLD provides a self-reinforcing dynamic where the network amplifies weights corresponding to relevant input dimensions, leading to neuronal sparsification. The authors show that such a process cannot be described by standard MF approach. To remedy this, they propose a minimal extension, MF-ARD (Automatic Relevance Determination), which incorporates a learnable, coordinate-wise variance into the weight prior to the MF derivation. This model is shown to quantitatively reproduce the learning curves of SGLD-trained ReLU networks on sparse-parity tasks.

**Strengths:**

**Clarity**: The paper is exceptionally well-written and structured. The theoretical difficulty hierarchy of SGLD $\rightarrow$ MF $\rightarrow$ NNGP provides a clear and effective pedagogical framework for situating the paper's contributions and understanding the different levels of theoretical approximation. The core argument is easy to follow, and the visualizations are very helpful.

**Quality**: The empirical validation for the specific setting under consideration is of high quality. The quantitative agreement between the MF-ARD theory and SGLD experiments, as shown in Figures 1 and 5, is impressive and demonstrates that the proposed model accurately captures the learning dynamics in this regime.

**Originality & Significance**: The use of the IFS mechanism as a simple extension to the common MF approaches discussed in the paper is a sound and valuable scientific contribution. The authors have successfully predicted an emergent mechanism of FL, which has a simple intuitive explanation.

**Weaknesses:**

Despite its strengths, the paper suffers from several significant weaknesses related to its framing, the generality of its claims, and the novelty of its contribution. My primary concerns are detailed below. Other than these critical points, I think that this is a good paper, and I would be happy to raise my rating if the concerns here are appropriately addressed in the revised text.

**Critical reference omission**: The paper's analysis is critically hampered by its failure to engage with the concurrent work of Van Meegen & Sompolinsky (2024) (VM&S), which provides a general framework for understanding learned representations in the feature-learning regime. Crucially, VM&S. demonstrate that the weight distribution is determined by the neuronal nonlinearity. Specifically, they show that ReLU networks produce sparse distributions as in this work, while other nonlinearities like sigmoidal, produce qualitatively different distributions.

**Overstated novelties**:
(a) It seems that the omitted reference may have led the authors to misinterpret their central finding. The paper identifies sparsification (driven by IFS) as the key mechanism that standard MF theory fails to capture, whereas following VM&S, this seems to be only an outcome of the ReLU activation function used in this work. Considering the previous point, the scope of this paper may not be quite as wide as implied by the authors.
(b) A central claim of the authors is that they developed an interpretable MF theory, where other works involve “... complex theoretical machinery [that] often obscures the core mechanisms driving FL”.  This is not an accurate representation of existing works, where the “machinery” is solving an equation in one (Li & Sompolinsky, (2021); Pacelli et al. (2023; Ringel et al. (2025)) or two (Rubin et al. (2024)) variables. Moreover, existing frameworks produce simple, intuitive, and interpretable descriptions of FL, and make accurate predictions in various regimes. For instance, the works of Li & Sompolinsky (2021) and Pacelli et al. (2023) offer a simple rescaling explanation, Fischer et al. (2024) find interpretable structures emerging in kernels, and Rubin et al. (2024) show that weights learn to prefer relevant directions. The authors even claim that "Our key contribution is to show that incorporating this IFS mechanism requires only a minimal,
principled modification to the MF theory". By this reasoning, the interpretability of their framework cannot be a significant improvement over MF approaches.

**Questions:**

1. As far as I can tell, the MF theory provided here is identical to the one appearing in Seroussi et al. (2023), Rubin et al. (2025), and Ringel et al. (2025). However, the authors claim that one of the main contributions of this work is "Interpretable MF theory", which seems unjustified. In this context, I have the following questions/suggestions, and I would be happy to raise my score if they were to be answered as well:
(a) Could the authors explicitly relate their derived mean-field equations to the MF formulations in other recent static theories (e.g., Seroussi et al., 2023; Fischer et al., 2024; Rubin et al., 2024)? Or at least some of them?
(b) If, as I suspect, these result in the same distribution (possibly up to negligible factors of $1/N^2$), then positioning "interpretable FL theory" as a primary contribution is an overstatement and should be clarified in the text.
(c) Otherwise, the explicit difference between existing approaches should be highlighted.

2. The MF-ARD posterior for a single neuron (Eq. 14) has a very similar form to the plain MF posterior (Eq. 6), with the main difference being the introduction of $d$ new order parameters, the precisions $\rho_j$. Could the authors expand on in what sense does this modification constitute a more interpretable theory of feature learning as posited in the introduction?

3. Could the authors clarify the scaling of the noise parameter $\kappa$ with the network width $N$? In some mean-field treatments, $\kappa$ also acquires a width-dependent scaling, such as $\kappa \sim \mathcal{O}(N^{\gamma-1})$. Is such a scaling assumed here, and if not, how does its absence affect the results?

4. What are the limitations\assumptions on the MF approximation? This should be detailed explicitly in the main text.

5. The paper claims that the "homogeneity assumption" of the MF model prevents it from capturing the effect of specializing neurons. However, MF theory is seemingly capable of capturing distributions where only a small number of neurons specialize through multimodal Gaussian distributions. Essentially, taking a distribution where there is a $1/N$ probability of being in a certain nonzero mode is equivalent to saying that one neuron is drawn from a distribution that is some nonzero Gaussian, and the remaining neurons are drawn from the other modes of the distribution. How does this picture differ from the picture provided here? Contrasting this specialization with multi-modal distributions in the paper itself would be very helpful.

6. The paper's “related works” section implies that many static MF theories are restricted to the proportional limit where $P \propto N$, where $P,N$ are sample size and network width respectively. This is not a general requirement, as shown in other works (e.g., Ringel et al. (2025) - not cited in this paper). There, other scalings of $P$ were considered, specifically taking $N\propto d$ (where $d$ is the input dimension), but $P$ scales as $d^{3/4}$. Is taking $N\propto d\mapsto\infty$ what the authors were refering to in the statement: “either in double asymptotic limits of data and width”? If so, this should be clarified in the main text.

---

> ### Author Response · Authors · 2025-11-18
> **Rebuttal to reviewer dnTH**
>
> We thank the referee for the thoughtful comments.
>
> # Weaknesses
>
> #### **Critical reference omission**
> We kindly ask the reviewer to read our general comment about vMS2024 above. This paper analyses the neuron-level coding: how many out of $N$ neurons $\{\mathbf{w}_1,...,\mathbf{w}_N\}$ correlate with the target function. Our MF theory has only one neuron $\mathbf{w}=[w_1,...,w_d]$ and is about a phase transition in the coordinate space of this one neuron (i.e., how many of the $d$ coordinates pick up a strong target correlation). These two mechanisms are compatible: ReLU can induce neuron‑level sparsity (vMS2024), and simultaneously, within each active neuron, the coordinate marginals can develop heavy‑tailed anisotropy (due to our IFS mechanism). To further strengthen this point, we show in Fig. 2 in the new supplementary material that for *sigmoid* activation functions, the same IFS/ coordinate sparsification happens, even though on the neuron-level the prediction in  vMS2024 is that there is no neuron-level sparsification.  This clearly shows that our predictions  are about a different effect from the ones studied in  vMS2024.
> #### **Overstated novelties**
> **(a):**  We hope the discussion above  egarding vMS2024, as well as the additional experiments, fully clear up this point.
>
> **(b)**: We agree that "more interpretable" is a vague term.  We will remove this claim from the introduction. As explained in the response to your question 2 (see below), to our knowledge this paper is the first one to use an  MF framework together with the  IFS mechanism  to explain how NNs learn low-dimensional targets $Y$ from high-dimensional isotropic data $X$. We will focus the introduction more clearly on this key point.
>
> # Questions
>
>  - **Q1:** **(a)** Our simple MF theory, when neglecting ARD, is very close to the papers you cite.   Please see the new supplementary material section 2 for a detailed derivation of how simple MF relates to the MF theory in "Grokking as a First Order Phase Transition in Two Layer Networks, Rubin 2024". The main difference is that we projected on a function space basis and in this way we have a self-consistency equation for the  $m_A$-couplings instead of a self consistency equation in the kernel and discrepancy field. Mathematically, the two theories coincide, up to a $1/2 \ln (...)$ factor that our theory did not drop (but which is irrelevant for large $N$).  **(b)** As discussed above, in the final version we will drop the claims about interpretable ML theory, and instead focus in the introduction on how NNs learn low-dimensional signals from isotropic data without putting strong emphasis on the interpretability, as this is a vague term.
>  **(c)** Yes, we will focus on how MF-ARD explains how NNs find signals in isotropic data.
>
>  - **Q2:**  Standard MF theory cannot predict the correct generalisation behaviour. By adding a minimal extension with MF-ARD, we provide a theory that  is  interpretable in terms of the new order parameters $\rho_j$ which
>  are directly measurable relevance coefficients for input dimensions, predicting anisotropy in $p(w_j)$ (coordinate level!) that we observe in real trained NNs trained on isotropic data with low-dimensional labels.
>
>  - **Q3:** We are aware of the scaling. However, in our empirical plots we fixed the width $N$ and we did a sweep across $\kappa$ and $P$, hence, the scaling of $\kappa$ with $N$ would only add an overall offset to the plot, not any change in the actual behavior. If wanted, we can add a plot of generalisation error vs $N$ as well, where we would add the $\kappa \sim O(N^{1-\gamma})$ scaling.
>
>  - **Q4:** We neglect  an $O(1/N)$ error due to our cavity assumption of replacing $\sum_{i\neq i'}a_i\phi(w_i^\top \mathbf{x})$ with $\langle f \rangle$ (see eq. 5). This leads to a factorization across neurons (no neuron–neuron correlations, all neurons share the same single‑neuron posterior) as in Rubin2024. In all empirical experiments, we do not take a $P \rightarrow \infty $ limit and simulate the finite $P$ effects by averaging over multiple random simulations with randomly drawn subsets of the data of size $P$. In Theorem 4.1 we use the $P \rightarrow \infty$ limit.
>
>  - **Q5:**.  We hope our overall comment above clarifies this question.
>
>  - **Q6:** We will rewrite the related work section and add Ringel et al. 2025 as well, and remove the claim that MF theories are restricted to the proportional limit.

---

### Official Review · Reviewer_s426 · 2025-10-30

**Soundness:** 2
**Presentation:** 2
**Contribution:** 1
**Rating:** 2
**Confidence:** 4

**Summary:**

The paper proposes a *Mean-Field Automatic Relevance Determination* (MF-ARD) framework for two-layer Bayesian neural networks trained with SGLD. The goal is to model the “specialization transition” in fully connected networks at the interpolation regime, where just a finite subset of neurons align with the target function. Since standard mean-field theory (exact at infinite width $N\to \infty$ when sample $P=\mathcal{O}(1)$) yields a fully factorized posterior, the authors introduce a coordinate-dependent Gaussian prior with Gamma-distributed precisions (inverse variances controlling per-weight decay strength), enabling anisotropic shrinkage across weight coordinates. They derive fixed-point equations for these precisions and show, through comparisons with SGLD simulations, that the resulting model resemble the qualitative emergence of neuron specialization and sparse, task-aligned representations observed in finite-width networks.

**Strengths:**

The paper - like many recent works at the intersection of statistical physics and machine learning - addresses a relevant theoretical question: how mean-field theory can be extended beyond the infinite-width, small-data limit to capture internal feature specialization and neuron-neuron correlations in finite-width networks. The proposed MF-ARD formulation is conceptually clear and analytically tractable. The manuscript is clearly written, and the authors provide numerical comparisons between their theoretical predictions and SGLD simulations, showing qualitative agreement.

**Weaknesses:**

**Theoretical inconsistency between MF-ARD and SGLD dynamics.**

This paper is meant to describe the stationary distribution of SGLD dynamics for two-layer neural networks through a mean-field approximation and **uniform** weight decay, but the proposed MF-ARD formulation fundamentally changes the underlying Bayesian model. By introducing coordinate-dependent Gaussian priors with Gamma-distributed precisions $\rho_j$ the authors define a *different* posterior from the isotropic one actually sampled by SGLD. As a consequence, I don’t see how their “MF-ARD theory’’ can be interpreted as a faithful large $P$ limit of the true SGLD dynamics: they are effectively comparing two different models.

**Imposed rather than spontaneous specialization transition in the theory.**

Since the MF-ARD model introduces anisotropy explicitly through the prior rather than deriving it as a spontaneous effect of finite-width fluctuations (i.e., $\mathcal{O}(P/N)$ corrections to mean-field theory), the “specialization transition’’ is imposed by construction and has nothing to do with the original posterior of Eq.(4) they intend to study.

**Unrealistic neuron-wise sparsification.**

The transition described by the theory corresponds to a *hard sparsification*, where only a finite number of neurons acquire non-zero overlap $m_s>0$ with the teacher while the rest remain near initialization. This is fundamentally different to what happens in finite width neural networks during feature learning. In realistic dynamics, all neurons move substantially, but their updates become strongly correlated, so that the learned representation effectively spans a low-dimensional subspace.

**Restrictive assumptions on data and teacher.**

The theoretical analysis relies on a sparse teacher model, where the number of nonzero coefficients $m_A$ directly determines the number of “relevant” features or basis functions contributing to the target. In practice, this means the teacher vector is assumed to have support only on a small subset of input dimensions ($k$-sparse vector). While this setup is analytically convenient, it is not representative of realistic data or high-dimensional learning tasks. In real datasets, the relevant subspace is *not known a priori* and typically spans a continuum of directions with a decaying or heavy-tailed spectrum of features.  Given that the theoretical setup strongly relies on having $k$ finite, I don't see how it can be extended to practical scenarios.

**Theory describes a numerically unstable training setup.**

The proposed theory corresponds to an impractical actual training of a neural network with SGLD. Indeed, running SGLD or SGD with neuron-dependent (or coordinate-dependent) weight decay would require maintaining and updating a distinct regularization coefficient for each neuron or parameter, effectively introducing $\mathcal{O}(N) additional hyperparameters. This would break permutation symmetry across neurons, complicates optimization dynamics, and would make training highly unstable.

In summary, the paper starts from a standard SGLD framework but then shifts to analyzing a modified Bayesian model (MF-ARD) whose dynamics and posterior differ from those of the system it aims to describe. The resulting formulation is not a faithful large $P$ limit of a mean-field theory, nor does it capture the genuine feature-learning phenomenology observed in finite-width networks. For these reasons, I believe the paper requires major conceptual revision before it could be considered for publication. Its current scope and assumptions are limited to shallow networks and restricted to sparse targets, and the theoretical insights it provides are not clearly connected to modern learning dynamics in realistic settings.

**Questions:**

1.  Why did the authors choose to introduce anisotropy explicitly through the ARD prior rather than studying finite-width ($1/N$ or $P/N$) corrections to the isotropic mean-field theory that might be still tractable?
These corrections are known to produce neuron correlations and spontaneous anisotropy without changing the generative model.

2. Can the authors clarify how they are taking the asymptotic limit (of $P$ data and $N$ width)?
Of course these limits do not commute in practice, and to get a mean-field theory one usually takes $N\to \infty$ at fixed $P$, but Eqs. (6) and (7) have a population average.

3. Fig.4 (b) shows that the theoretical model at small $P$ over-estimates the anisotropy of hidden neurons. Could the authors comment on that?

4. In Fig. 1a, the test MSE stays flat even with increased samples $P$ before dropping sharply. Why does learning only start at such large $P$? Could the authors comment on what would happen at larger width?

Minor annotations:
- Lines 44-45: kernel renormalization theories are in the asymptotic limits of data and width so the sentence and citation is imprecise.
- Eq. (4) is missing a $+$ sign (typo).
- Line 160: fully factorized posterior for mean field theory is not an approximation, it’s exact in the limit $N\to \infty$.
- Line 249: $m_s$ mentioned before it has been introduced

**Details Of Ethics Concerns:**

None.

---

> ### Author Response · Authors · 2025-11-18
> **Rebuttal to reviewer s426 - Part 1**
>
> We want to thank the reviewer for the helpful comments.
>
> # Weaknesses
>
> #### **Theoretical inconsistency between MF-ARD and SGLD dynamics:**
> Please see our response to Question 2  for reviewer 1F6B. Briefly: SGLD will generate an anisotropic posterior when it interacts with the k-sparse parity data. The MF-ARD theory is an attempt to model the  SGLD and its interaction with the data. The SGLD equations are not modified.
>
> #### **Imposed rather than spontaneous specialisation transition in the theory**
> The ARD prior does not break symmetry by construction. At initialization (or at the MF fixed point with $m_A=0 \ \forall A$), the model is exactly isotropic in expectation:
>     $$\mathbb{E}[\alpha_0/\beta_0]=d/\sigma_w^2, \mathbb{E}[w_j^2]=\sigma_w^2/d \ \Rightarrow \rho_j=const. \forall j $$
> The only symmetry‑breaking term is the data coupling $J_{\mathcal{Y}}(\mathbf{w})$. Below the critical  sample size $P^*$, the MF fixed point remains at $m_A=0$ and all $\rho_j=const.$ remain equal, as long as the weights $w_j$ do not become strongly anisotropic. However, as soon as we cross the phase transition the  self‑consistent update for $\rho_j$ amplifies data‑induced differences in coordinate variances, as it should.
>
> #### **Unrealistic neuron-wise sparsification**
> We kindly ask the reviewer to read our general comment above regarding the misconception that our theory is about neuron sparsification. It is about IFS i.e., only some *coordinates* of a neuron pick up a strong correlation with the target. $m_S$ is the overlap of a single neuron with the target, our MF theory is on a single neuron level, hence there is no sparsification across different neurons.
>
> #### **Restrictive assumptions on data and teacher**
> Our theory makes no restrictive assumptions about the data. The function  $\langle f \rangle = \sum_A m_A \chi_A(\mathbf{x})$ can be expanded in any basis $\{\chi_A\}$.  However, we specifically choose $k$-sparse parity as well as a single index model because these datasets represent a setting where kernels fail to learn (in polynomial sample complexity) the target because the data $X$ is isotropic but the target $Y=f^*(X)$ has variance only on a small dimensional subspace (see general comment above point (1)). It is true that there are many other possible datasets to try, but this is, we believe, an interesting one for understanding feature learning because of large differences between NN and kernel performance.  Future work will look at other datasets.
>
> #### **Theory describes a numerically unstable training setup**
> We do not propose to train networks with per‑coordinate weight decay. The networks we compare against are trained with standard isotropic SGLD. The $\rho_j$ in MF‑ARD are variational order parameters of a static posterior approximation, not hyperparameters used during SGLD. No extra hyperparameters are introduced in training, and permutation symmetry across neurons is not broken. ARD is used to show that MF with a reinforcing symmetry breaking on *coordinate*-level (ARD) is enough to give correct generalisation predictions.

---

> ### Author Response · Authors · 2025-11-18
> **Rebuttal to reviewer s426 - Part 2**
>
> # Questions
> - **Q1:** Thank you for this question. We ask the reviewer to read the new section 4 and 4.1 where we derive ARD from linear response theory.
> - **Q2:** We do not need to take the $P \rightarrow \infty$ limit.  For all numeric predictions (learning curves, phase plots) we replace all expectations by empirical averages over the finite training set (and average over several datasets). This is exactly the replica over datasets approach which lets MF and MF‑ARD inherit finite‑sample effects. Only in Theorem 4.1 do we take the $P \rightarrow \infty$ limit for tractability. Considering the limit in $N$, the replacement in eq. 2 introduces a $O(1/N)$ error. We are happy to include a longer derivation in the final version having space for an additional page.
>            \item Just to be clear, Fig. 4b shows the anisotropy of the coordinates, not that of hidden neurons.  Nevertheless, MF-ARD slightly overestimates the anisotropy. We think this behaviour is caused by the ARD approximation missing cross‑neuron correlations. In real SGLD the shared residual creates negative feedback across neurons.
> - **Q3:** Explaining *why* a NN needs a critical sample set size to learn the target function is the central aim of the paper.  We argue that below the critical sample size, the NN does not discover  the  right signal in the isotropic data and no feature-learning occurs.  However, once the signal is strong enough, it induces a phase transition in the coordinates. This basic feature-learning effect can already be seen at the simple MF level, but to fully explain the good generalisation we need to include a second  IFS self reinforcement mechanism, which we capture in our MF-ARD approach.
>
> - **Q4:** What is the effect of larger width?: As discussed in the \textbf{No FL mechanism-paragraph} in section 2.3 of the paper,  there is no FL mechanism in the $N  \rightarrow \infty$ limit because the NN cannot pick up the anisotropy anymore on a coordinate level. This is explained analytically in that section, and illustrated for the infinite-width NNGP limit in Fig. 1a).  If needed, we are happy to include a plot with growing $N$ in the final submission.

---

### Official Review · Reviewer_1F6B · 2025-10-31

**Soundness:** 2
**Presentation:** 4
**Contribution:** 2
**Rating:** 4
**Confidence:** 4

**Summary:**

Summary

The manuscripts proposes a novel form of a heuristically
defined mean-field theory for feature learning in single
hidden layer networks in a setting of Bayesian inference.
The aim is to describe feature learning beyond previous works
of kernel adaptation and kernel renormalization.

The main novelty is the proposal of a heterogeneous prior distribution
on the first layer's weights. Deriving mean-field equations for a set of
order parameters, including the variances of the weights' priors, allows
the interpetation of the latter as a measure of relevance of input
features and describes settings where individual neurons specialize
on distinct input features.
Numerically trained networks with Langevin gradient descent with
weight decay show good agreement with the theory.

Soundness
Most of the manuscript is sound, in particular the first part of the
manuscript, until including section 3, which is equivalent to earlier
work on feature learning in a setting of Bayesian inference. The authors
nicely explain the mean-field approach also in intuitive terms, yet provide
all detail in the appendix.

The novel part starting in section 4. Here the main puzzling point to me
is that the heterogeneous variances and the hyper-prior for the variances
of individual neurons' weights are introduced ad-hoc. This is in contrast to
the first part, where the startionary distribution (Eq. (2)) is a direct
consequence of the applied Langevin training dynamics.

Presentation
The presentation of the work is very good and provides excellent explanations
that can also be followed intuitively, even for readers who do not follow
all mathematical details.

Contribution
The main contribution the work is to provide a simple ad-hoc (not strictly
derived from first principles) mean-field theory that explains specialization
of individual neurons to explain feature learning.
A point that needs to be clarified in the rebuttal is the novelty.

In particular in the light of previous work by van Meegen & Sompolinsky 2024
"Coding schemes in neural networks learning classification tasks" Nat Comm.
The latter work considers the same problem as the authors and derives a mean-field theory to explain the specialization of neurons on particular features.
The authors should, for example, look at Figure 8 of said paper. The theory
by van Meegen and Sompolinsky performs a mean-field approach where the readout
weights (variable "a" in the current manuscript) concentrate.

**Strengths:**

I very much like the form of presentation chosen by the authors, summarizing
main equations in boxes and providing explanations and intuition for the appearing order parameters. Also the authors throughout provide careful comparisons between theory and numerics.

**Weaknesses:**

The missing mention of van Meegen & Sompolinksy 2024 as a tightly related
work casts some doubts about the novelty and contribution. In case this
point can be clarified in the rebuttal I am happy to raise my score.
(see section contribution in summary box).

**Questions:**

Given my concern on the novelty (see "Contribution"), my first concrete
question to the authors is to check whether the previous theory by van
Meegen & Sompolinsky is able to explain their numerical findings or
whether there is still a gap between feature learning theory and
numerics that their theory is able to close.

My second question is more conceptual. I understand the heuristic motivation
of the approach. What I fail to see is why the proposed log likelihood (Eq. 14)
is the log of the stationary distribution of the Langevin training in (Eq. 1).
To me this seems like the attempt to capture heuristically the observations
from numerics rather than deriving this result from first principles.

---

> ### Author Response · Authors · 2025-11-18
> **Rebuttal to reviewer 1F6B**
>
> We thank the reviewer for the positive feedback on the presentation of the paper and appreciate pointing out the missing citation.
>
> # Weaknesses
>
> We kindly ask the reviewer to read our general comment above as it should clearly separate our work from vMS2024.
>
> # Questions
>
> ### **Q1 - Novelty and vMS2024**
> Please see the comment on the weakness above.
>
> ### **Q2**
> The stationary distribution of SGLD is given by Eq. 2 in the paper:
>
> $$-\ln p_{GD} (\mathbf{\theta} |\mathcal{D}) \propto \sum_{l=1}^L \frac{1}{2 \sigma_i^2}\sum_{i=1}^{N_l} | \theta_i |^2 + \frac{1}{2 \kappa^2 P} \sum_{\mu=1}^P (f_{\theta}(x_\mu) - y_\mu)^2$$
>
> However, because all $N$ neurons interact, this posterior is intractable. Standard mean-field theory simplifies the posterior by assuming neurons are i.i.d., and while this approximation predicts the coordinate-level symmetry breaking seen in settings with isotropic data and low-dimensional targets (such as the standard example of $k$-sparse parity), it fails to explain the neural network's performance gains. To address this, we introduce MF-ARD, which utilizes a small set of new order parameters, the precisions $\rho_j$, to capture the mechanism of feature learning in these subspaces.
>
> *Crucially, we do not change the SGLD sampling equations.* The precisions $\rho_j$ are introduced strictly as new order parameters within a mean-field theory aimed at constructing a more faithful *variational approximation* to the full, intractable SGLD posterior. The ARD prior is an attempt to capture the self-reinforcing input feature selection effect induced by the interaction of Gradient Descent and the data, a well-known phenomenon (see, e.g., [1], [2]).
>
> While we acknowledge that the ARD prior is not yet fully derived from first principles, the excellent empirical predictions suggest that our combination of symmetry breaking and self-reinforcement captures something fundamental about feature learning in this context. We note that in statistical physics, using a heuristic approach to create new order parameters with the aim of capturing more complex phenomena is standard practice. Nevertheless, we are currently exploring a 2-population model of dormant versus active neurons (based on [3]) as a potential first-principles derivation and will include a discussion of this approach in the final version.
>
>
>
> [1]: https://arxiv.org/pdf/2410.04264?
> [2]: https://arxiv.org/pdf/2502.21009
> [3]: https://arxiv.org/pdf/2506.06489

---

> > ### Comment · Reviewer_1F6B · 2025-11-24
> >
> > I acknowledge the work stat went into the rebuttal.
> > My first main point of criticism, that the theory is not derived from first principles, is still not addressed. Since the authors seem to have found a way around that, I would encourage them to work this out in more detail (and likely resubmit to one
> > of the other venues).
> > The second point, the difference between van Meegen & Sompolinksy 2024 and their work has been explained nicely as the difference in specialization between input (this work) and output (vM & S24).
> > Still, I would like to keep my score, mainly because of point 1 (the derivation of the theory from first principles).

---

> > > ### Author Response · Authors · 2025-11-26
> > > **Answer to rebuttal**
> > >
> > > - We thank the reviewer for the response. We would like to point out that the reviewer specifically wrote: "The missing mention of van Meegen & Sompolinsky (2024) as a tightly related work casts some doubts about the novelty and contribution. In case this point can be clarified in the rebuttal I am happy to raise my score." The reviewer now acknowledges that we have clarified this point, but the score has not been raised.
> > >
> > > - Independent of this, we would like the reviewer to look at the new version of the PDF. We derive ARD from a linear response ansatz in the SGLD dynamics, and we hope this clarifies the reviewer's remaining concerns (section 4 and 4.1).

---

> > > ### Comment · Reviewer_1F6B · 2025-11-26
> > >
> > > The authors are right, that I mentioned as my main concern the missing link to van Meegen & Sompolinksy 2024. They have indeed very clearly explained the difference.
> > >
> > > Even more importantly, in the new revision they in addition provide a justification for their theory from first principles.
> > > I was misled by their response "While we acknowledge that the ARD prior is not yet fully derived from first principles, ..." that the theory is heuristic.
> > > I like their idea to employ linear response theory and using one-loop fluctuation corrections on the level of weights
> > > and had missed this new addition to the manuscript at my earlier inspection. This makes the work much stronger, so I am happy to raise my score to 6.

---

### Author Response · Authors · 2025-11-18
**General comment on how our paper differs from van Megen and Sompolinsky 2024**

We thank the reviewers for drawing our attention to the paper by van Megen and Sompolinsky (2024, henceforth vMS2024). In the revised version, we now cite this paper and include a brief discussion of its relation to our own findings in a new *Appendix C.2.* However, our paper differs from vMS2024 as the precise meaning of sparsification differs:

**Consider the following example:** Let the first layer’s weight matrix be of size $d \times N$. The sparsification of vMS2024 is sparsification among the  $N$ neurons. In our work, because MF theory does not distinguish between $N$ neurons, the sparsification is among $d$ input coordinates of a *single neuron*.

In more detail:

- **vMS2024:** Sparsification is across neurons - in a network with $N$ neurons, only $O(1)$ neurons strongly correlate with the target.

- **Our work:** Sparsification is within **input** coordinates of a *single neuron*. $\mathbf{w}=[w_1, \ldots,w_d]$ - only  $k$ (out of $d$ dimensions) correlate with the target. Our mean-field (MF) theory analyzes *a single neuron*, and thus is unable to predict any cross-neuron sparsification (see Fig. 2, Fig. 4(a)).

This fundamental difference in approach has several important consequences, which we discuss below:

 - **(1) Our paper focuses on input feature selection (IFS):** We focused on specific datasets like $k$-sparse parity or single index models where only a few input dimensions are relevant to the target because kernels have trouble learning these tasks. Prior work [1], [2] showed that kernels match NN sample complexity when (loosely speaking) two conditions hold:
    -    (a) The data $X$ is anisotropic, meaning $E[XX^\top]$ has $d_0 \ll d$ large eigenvalues spanning a subspace $V$ and $d-d_0$ small eigenvalues.
    - (b) The target variance focuses on the exact same subspace $V$, i.e., $Y=f^* (X) \approx f^*(P_V X)$, with projection $P_V$.


When either condition fails, NNs achieve (oftentimes exponentially) better sample complexity. Our paper shows that NNs can succeed on these datasets by having an anisotropic phase transition in the *input coordinates* (Fig. 3 and 4), *not* the sparsification between $N$ neurons as studied in vMS2024.


- **(2) Sparsification on the inputs is needed to explain why NNs outperform kernels:**  To further strengthen the evidence that the anisotropic phase transition in coordinates is a central mechanism explaining how NNs outperform kernels,  we performed  experiments training   on $k$-sparse parity with an additional homogeneity regularisation for coordinates within a neuron:
$$\ell_{\text{total}} = \ell_{\text{MSE}} + \lambda_h \sum_{j=1}^{N} \left[ \frac{1}{d} \sum_{i=1}^{d} \left( |w_{ij}| - \left( \frac{1}{d} \sum_{k=1}^{d} |w_{kj}| \right) \right)^2 \right]$$
Crucially, we don't penalise for sparsification at the neuron level— it could be that only $O(1)$ out of $N$ neurons can pick up the target as in vMS2024.  We observe that with increasing homogeneity (higher $\lambda_h$), the NN learning curves begin to behave like NNGP curves, see Fig. 1 in the new **supplementary material** (with a more thorough discussion). This provides strong independent evidence that a phase transition in the *coordinates* is a key factor enabling NNs to outperform kernels when learning from isotropic data.


- **(3) Sigmoid and ReLU give the same IFS phenomenon:** In Fig. 2 in the new **supplementary material** we repeat our experiments with the sigmoid activation, i.e., we repeat Fig. 1,4 from the main paper. With sigmoid, IFS happens in the same qualitative way as with ReLU. This is not in contradiction to vMS2024 (they predict ReLU $\rightarrow$ sparse coding, sigmoid $\rightarrow$ redundant coding) as our IFS mechanism works on the level of  coordinates, *not* neurons.


- **(4) We explore limitations of standard MF theory:** We emphasise that our paper shows that for target functions with isotropic inputs but low-dimensional labels, standard MF theory — including that of vMS2024 — fails to predict learning curves.  Using the code from the vMS2024 paper, we find that while their MF theory does reproduce a rapid reduction in the error for large enough $P$, the predicted transition occurs about an order of magnitude later than the true phase transition for the $k$-sparse parity task (see Fig. 3 and shared code snippet in the new **supplementary material**). The critical insight, which goes beyond vMS2024, is that IFS is necessary (as shown in Theorem 4.1 in the paper and the experiments above).

We hope this clarifies how our use of “sparsification” differs from that in vMS2024.


[1]: https://proceedings.neurips.cc/paper_files/paper/2020/file/a9df2255ad642b923d95503b9a7958d8-Paper.pdf
[2]: https://arxiv.org/pdf/2507.19680

---

### Meta-Review · Area_Chair_41u5 · 2026-01-09

**Summary:**

This paper presents a mean-field analysis of feature learning in two-layer neural networks trained with SGLD, and identifies an input feature selection mechanism that facilitates feature learning. The reviewers raised the following concerns.

* Reviewers mzGy, dnTH, and s426 have concerns about the limited theoretical scope, such as sparsity being a property of the ReLU activation. The authors clarified that feature-level sparsification is not a unique feature of the chosen activation and provided empirical evidence.

* Reviewers mzGy, dnTH, and 1F6B mentioned overlap with Van Meegen and Sompolinsky 2025. The authors addressed this concern by emphasizing that their studied feature-level sparsification is fundamentally different from the neuron-level sparsity studied in prior works.

* Reviewers s426 and 1F6B questioned whether the ARD prior alters the SGLD algorithm. The authors clarified that while the prior is not derived from first principles, it empirically captures feature learning behavior better than traditional mean-field approaches.

Overall, the authors' response partly addressed the reviewers' major concerns. However, the area chair believes that the contribution of this submission needs to be better positioned in the feature learning theory literature, i.e., what the advantage and new insights of this formulation are in the learning of low-dimensional teacher models, which has been extensively studied.
On a side note, the listed references do not show sample complexity separation between fixed kernels and feature learning for the original $k$-parity problem (with no data modification). The complexity of both approaches is likely $n = d^{\Theta(k)}$ unless you assume axis-aligned initialization.

**Reviewer Concerns:**

See above.

**Reviewer Scores:**

The authors clarified the overlap with Van Meegen and Sompolinsky 2025, hence addressing one of the main concerns of reviewers 1F6B, dnTH, and mzGy — two of these reviewers have updated their ratings to borderline accept. Reviewer s426's concerns are also partly addressed. Therefore, the area chair believes that after the rebuttal period, the submission is placed right at the borderline.

---

### Decision · Program_Chairs · 2026-01-26

Reject